# Guided phase transition for mitigating voltage hysteresis of iron fluoride positive electrodes in lithium-ion batteries

Hyoi Jo[1,7], Minjeong Gong[2,7], Se Young Kim [3], Dong-Hwa Seo [2,8] ✉ & Sung-Kyun Jung [1,4,5,6,8] ✉

Despite the high capacity attained by conversion-reaction-based metal-fluoride positive materials in lithium-ion batteries through multiple electron storage, the large voltage hysteresis and low structural reversibility constrain their use. Herein, we propose guided phase transitions for designing conversion-type positive materials that undergo minimal structural changes upon lithium-ion storage. This approach reduces the compositional inhomogeneity, a culprit of the voltage hysteresis, while providing high structural reversibility. The thermodynamically stable rhombohedral $FeF_3$ involves irreversible phase transitions accompanied by significant structural rearrangement during lithiation. In contrast, the metastable tetragonal $FeF_3$, electrochemically derived from a $LiF$-$FeF_2$ composite, undergoes facile and reversible phase transitions by maintaining structural integrity, enabled by conversion reactions between structurally analogous phases. Our study provides valuable insights into the importance of avoiding irreversible reaction pathways and deliberately guiding them to minimize structural changes in the crystal lattice, which is critical for designing positive materials with high structural reversibility.

Lithium-ion batteries (LIBs) have been implemented in core energy-storage technology in the value chain of sustainable energy production for a wide range of applications from portable electronic devices to electric vehicles[1–4]. Intercalation chemistry has achieved great success in positive materials for the industrial application of LIBs through lithium and electron storage with angstrom-scale reaction in the same open-framework crystal structure. By leveraging intercalation chemistry, numerous promising positive material candidates have been developed, including NMC (nickel manganese cobalt), LCO (lithium cobalt oxide), LMO (lithium manganese oxide), LFP (lithium iron phosphate), and DRX (Disordered Rock salt)[5–14]. However, these intercalation-based materials have limited capacity due to the finite number of interstitial sites in the host lattice, which restricts further enhancements of the specific energy.

Conversion reaction chemistry, which can store multiple electrons without the constraints of an open framework crystal structure, presents a viable option to overcome the limitations in specific energy. This approach offers significantly higher specific capacity than existing intercalation materials[15–18] by decoupling lithium and electron storage through the formation of lithium compounds and transition metals. Despite these advantages, pervasive issues of large voltage hysteresis and low structural reversibility[19–24], widely

[1]Institute for Battery Research Innovation, Seoul National University, Seoul, Republic of Korea. [2]Department of Materials Science and Engineering, Korea Advanced Institute of Science and Technology (KAIST), Daejeon, Republic of Korea. [3]Energy Storage Research Center, Korea Institute of Science and Technology (KIST), Seoul, Republic of Korea. [4]Department of Materials Science and Engineering, College of Engineering, Seoul National University, Seoul, Republic of Korea. [5]School of Transdisciplinary Innovations, Seoul National University, Seoul, Republic of Korea. [6]Research Institute of Advanced Materials, Seoul National University, Seoul, Republic of Korea. [7]These authors contributed equally: Hyoi Jo, Minjeong Gong. [8]These authors jointly supervised this work: Dong-Hwa Seo, Sung-Kyun Jung. ✉e-mail: dseo@kaist.ac.kr; naecard@snu.ac.kr

observed in conversion-reaction materials, remain the greatest challenges.

The voltage hysteresis and low structural reversibility are understood to stem from compositional inhomogeneity caused by structural reconfigurations and phase displacement[19–21,25] and are closely related to both reaction kinetics and mechanism. Therefore, strategies such as designing the composites with conductive materials[26–30] or reducing the particle size to the nano-size level[19,21,26,31] have been proposed to overcome the limited reaction kinetics due to the low electronic conductivity and sluggish mass transport from long-range diffusion. Despite these efforts, the compositional inhomogeneity and voltage hysteresis have not been completely resolved, which implies that they may also have reaction pathway origins. According to recent studies on the reaction mechanisms of $FeF_3$, which is a representative conversion positive material due to its high theoretical specific energy (1922 Wh kg$^{-1}$), high voltage, and cost-competitive[17,18,32,33], intermediate multi-phases with different chemical compositions are irreversibly formed during the first discharge process, with each phase following different reaction pathways upon charge and discharge[21]. This process deepens the compositional inhomogeneity during repeated cycling, ultimately leading to poor cyclability.

Compositional inhomogeneity inevitably induces different reaction pathways in general for chemical and electrochemical reactions. In previous studies on sodium-based positive materials, for the solid-state synthesis of $Na_{0.7}CoO_2$, an initial phase different from the global composition ($NaCoO_2$) forms preferentially during synthesis[34]. This preferential formation is caused by the local minimum of Gibbs free energy depending on the local chemical composition. Another study reported that the synthesis method can affect local compositional variations, leading to the formation of thermodynamically metastable phases depending on the synthesis approach[35]. These findings suggest that in situations with significant compositional variation and spatial separation, different phases can grow in distinct regions, each following different reaction pathways. As structural reformation observed during charge–discharge cycles in the (re)conversion reaction can be regarded as a type of electrochemical synthesis process[36–39], it is necessary to suppress thermodynamically induced compositional inhomogeneity to fundamentally address the voltage hysteresis and low structural reversibility.

To mitigate the compositional inhomogeneity from a reaction pathway perspective, it is essential to evade phase-displacement reactions accompanying long-range diffusion. In this respect, nanocomposite cathodes composed of lithium compounds and transition-metal compounds have successfully guided reversible reaction routes with minimal diffusion while maintaining the mother structure or anion framework of transition-metal compounds. For example, the LiF-FeO composite guided the formation of a new cubic FeOF host structure electrochemically while retaining the cubic FeO structure, distinct from the conventional rutile FeOF structure[40]. In the LiF-MnO system, the incorporation of the F anion induces a reversible phase transition from the rock-salt to spinel-like structure, which proceeds with short diffusion of Mn ions from the octahedral site to the face-shared tetrahedral site[41]. Both combinations avoid drastic structural changes even during charge and discharge reactions compared with typical conversion reactions. Based on these previous results[40,41], it can be reasonably predicted that tetragonal $FeF_3$[42] can be formed from nanocomposites of $LiF$-$FeF_2$ by maintaining the structural similarity with tetragonal $FeF_2$. Moreover, this nanocomposite can overcome kinetic limitations by ensuring efficient lithium and electron transport through nanosized particles and carbon composites. Thus, effectively utilizing the nanocomposite strategy can provide a comprehensive solution for designing advanced conversion-type positive materials with mitigated compositional inhomogeneity and voltage hysteresis.

Herein, we report on the design of tetragonal $FeF_3$ (T-$FeF_3$) derived from a LiF-$FeF_2$ nanocomposite to address the issue of large voltage hysteresis and low structural reversibility observed in conventional rhombohedral $FeF_3$ (R-$FeF_3$). We reveal that the crystalline structure of the tetragonal $FeF_2$ in the LiF-$FeF_2$ nanocomposite successfully guides the phase transition towards the formation of metastable T-$FeF_3$ rather than the thermodynamically stable R-$FeF_3$. The induced T-$FeF_3$ exhibits structural similarity to the discharged $FeF_2$, facilitating facile phase transitions, including insertion and conversion reactions. These phase transitions reduce compositional inhomogeneity, resulting in low voltage hysteresis and high structural and electrochemical reversibility. As a result, T-$FeF_3$ maintained 72% of its initial capacity after 300 cycles at a specific current of 50 mA g$^{-1}$, significantly outperforming R-$FeF_3$, which retained only 50% of its capacity. Moreover, the high structural reversibility of T-$FeF_3$ was maintained even after the formation of LiF and Fe metal phases under the deep discharge of a conversion reaction. Our research underscores that a guided phase transition that can maintain structural similarity can open a new reaction route that can evade the reaction pathway accompanying long-range diffusion, which can mitigate the pervasive issues of large voltage hysteresis and low structural reversibility in conversion-reaction chemistry.

## Results and discussion
### T-$FeF_3$ guided from LiF-$FeF_2$ nanocomposite

The LiF-$FeF_2$ nanocomposite was prepared by mechanical ball milling of LiF and $FeF_2$ following previously reported procedures[43]. Rietveld refinement confirmed the formation of the LiF-$FeF_2$ nanocomposite, revealing phase fractions of 55.34% for LiF (s.g. *Fm-3m*) and 44.66% for $FeF_2$ (s.g. *P4₂/mnm*) (Fig. 1a and Supplementary Table S1). A slight excess of the Li source is employed to enhance electrochemical capacity by increasing accessibility of the fluorination source, LiF, to $FeF_2$[43,44] (Supplementary Fig. S1). The transmission electron microscopy (TEM) of Fig. 1b shows well-mixed nanodomains of 5–10 nm in size, with $FeF_2$ and LiF represented by white and green, respectively. The azimuthal integration of the FFT patterns from the TEM images also indexed LiF and $FeF_2$ (Supplementary Fig. S2), further verifying the successful formation of the LiF-$FeF_2$ nanocomposite.

First, the voltage profile of the LiF-$FeF_2$ nanocomposite was examined at 25 °C within a voltage range of 4.8–2.0 V. Figure 1c shows the charge-discharge profiles of the LiF-$FeF_2$ for the initial 10 cycles, compared with the 10th cycle profile of rhombohedral $FeF_3$ (s.g. *R-3c*, R-$FeF_3$), prepared by ball-milling with carbon to mitigate kinetic limitations (Supplementary Fig. S3). During cycling, the 4 V plateau in the LiF-$FeF_2$ nanocomposite, indicated by the red-shaded area, gradually evolved, with the average discharge voltage increasing from 3.02 V (1st cycle) to 3.15 V (10th cycle). By the 10th cycle, the discharge capacity is 189.5 mAh g$^{-1}$, corresponding to the insertion of 0.85 Li$^+$ (theoretical capacity of 223.75 mAh g$^{-1}$ for single-electron transfer). In contrast, such electrochemical features were absent in R-$FeF_3$, as clearly seen in the dQ/dV analysis. Given the reversible redox reaction of iron involving fluorination, which is confirmed by X-ray absorption spectroscopy (Supplementary Figs. S4a and S5), it is notable that the redox reaction around the 4 V in the LiF-$FeF_2$ nanocomposite represents a higher voltage for the Fe$^{2+}$/Fe$^{3+}$ redox couple compared to LiFeSO$_4$F with triplite (3.9 V) and tavorite structure (3.6 V)[45–47]. The origin of high redox potential will be discussed later, but it implies structural evolution during electrochemical cycling, distinct from R-$FeF_3$.

To investigate the structural evolution of LiF-$FeF_2$ during cycling, ex situ XRD was performed after the 1st, 5th, and 10th cycles (Fig. 1d). New peaks at 34.8°, 40°, and 66.7° (indicated by arrows) in the charged state gradually became more pronounced with cycling. However, the newly formed diffraction pattern in the charged state does not match the R-$FeF_3$ pattern. This contrasts with previous studies suggesting that LiF-$FeF_2$ forms a rhombohedral-like $FeF_3$ structure upon charging[43,48]. Instead, the evolved structure is rather consistent with the tetragonal $FeF_2$ diffraction pattern, implying the formation of a

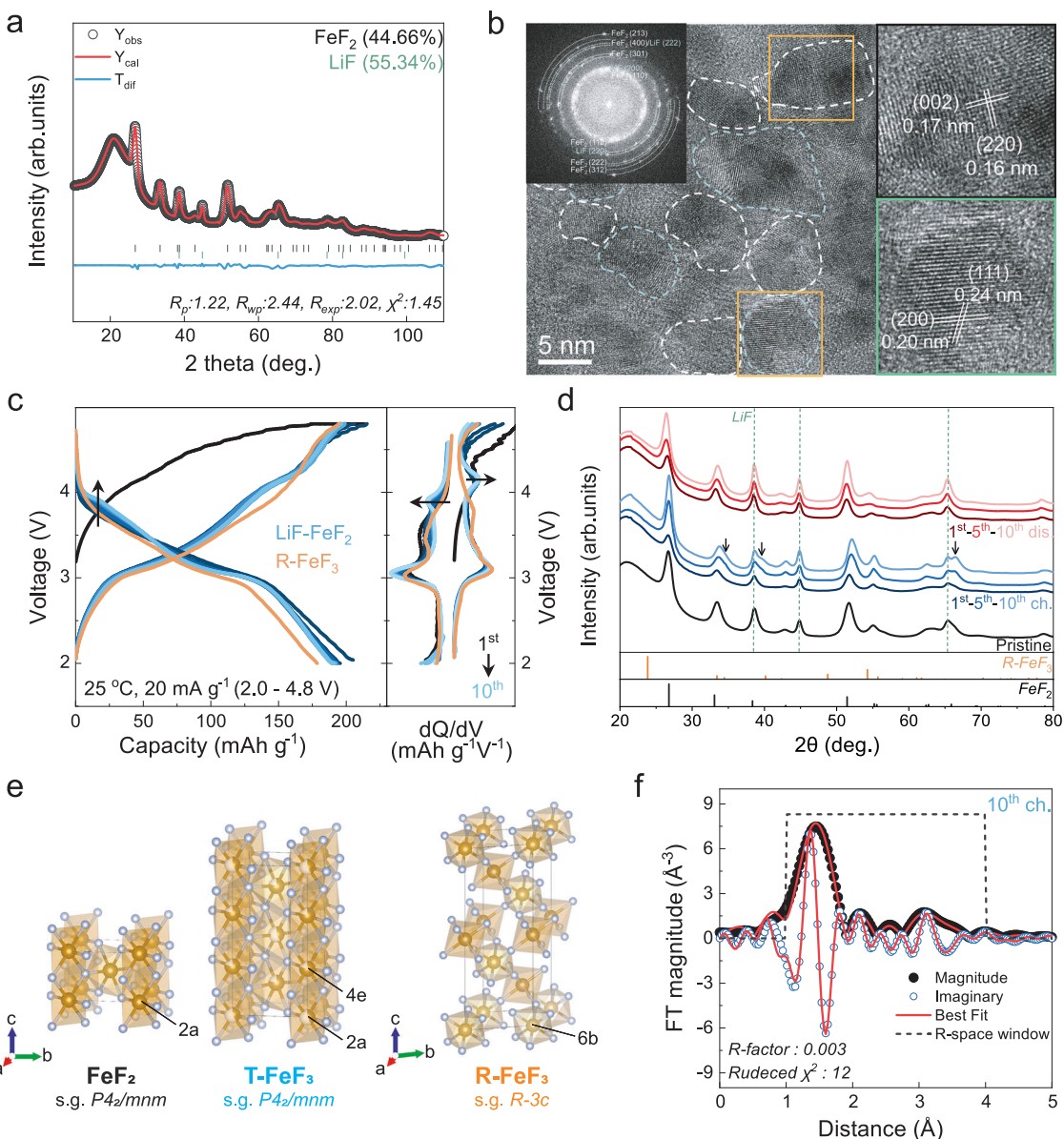

**Fig. 1 | Formation of T-FeF₃ phase guided by LiF-FeF₂ nanocomposite. a** Rietveld refinement of the X-ray diffraction (XRD) data (λ = 1.5406 Å) of the LiF-FeF₂. **b** High-resolution transmission electron microscope (TEM) image of LiF-FeF₂ in a pristine state. Each domain is outlined with a dotted line. The inset shows the fast Fourier transform (FFT) pattern of LiF-FeF₂. White and green represent FeF₂ and LiF, respectively. **c** Electrochemical profile of LiF-FeF₂ nanocomposite at 25 °C and 20 mA g⁻¹ specific current. Blue depicts the evolving voltage profile of LiF-FeF₂ up to the 10th cycle, while yellow represents the 10th cycle profile of R-FeF₃. The right is the differential analysis of the voltage profile. **d** Ex situ XRD patterns of LiF-FeF₂ electrodes at charged/discharged states after the 1st, 5th, and 10th cycles measured at 25 °C and current density of 20 mA g⁻¹. Red and blue are discharge and charge states, respectively. **e** Crystal structures of FeF₂, T-FeF₃ (determined through X-ray diffraction of the 10th charge state), and R-FeF₃. Brown and silver balls indicate Fe and F ions, respectively. **f** Fourier transformed magnitude (black), imaginary part (blue), and best fit (red) using the T-FeF₃ model for the charged electrode.

tetragonal FeF₃ phase. This discrepancy with the previous study may be due to the local probe analyses, such as XAS, which might not definitely represent the overall structure[43,48]. When comparing the structural evolution between LiF-FeF₂ nanocomposite and R-FeF₃ (Supplementary Fig. S7), it is observed that charge/discharge progresses while maintaining the tetragonal and rhombohedral phases, respectively. This implies the formation of a tetragonal FeF₃ phase.

Based on the structural evolution of LiF-FeF₂, which maintains the tetragonal FeF₂ structure, the feasibility of tetragonal FeF₃ formation was further verified with local structural analysis. Figure 1e presents the crystal structures of FeF₂, tetragonal FeF₃ (T-FeF₃)[42], and R-FeF₃. T-FeF₃ consists of FeF₆ octahedra that are edge-sharing along the *c*-axis and corner-sharing in the *ab* plane, similar to the anion framework of

FeF₂ but with reduced Fe occupancy, leading to a modified unit cell. In contrast, R-FeF₃ has a structure based solely on corner-sharing, structurally distinct from T-FeF₃. To verify the formation of T-FeF₃ at the local-environment level, extended X-ray absorption fine structure (EXAFS) fitting was performed in the 10th charged state (Fig. 1f). The reduced $\chi^2$ and R-factor were lower for the tetragonal phase than for the rhombohedral phase, indicating better structural agreement with the tetragonal phase (Supplementary Fig. S8). This result is attributed to the shorter Fe–Fe bond distance of T-FeF₃ (3.16 and 3.69 Å) with its edge-sharing framework, compared to R-FeF₃ (~3.7 Å), which only has a corner-sharing framework of iron octahedra (Supplementary Fig. S4b and Supplementary Note 3). Moreover, for the pair distribution function (PDF) analysis, the 10th charge state was more consistent with

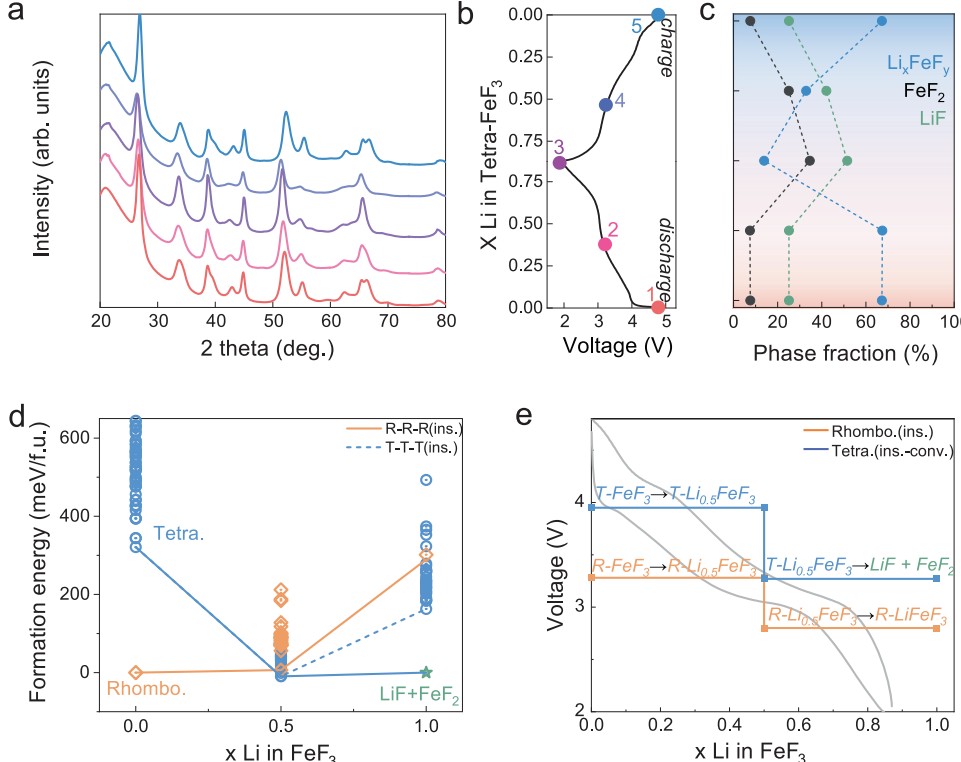

**Fig. 2 | Intercalation and conversion reaction of T-FeF₃. a** Ex situ XRD patterns of T-FeF$_3$ nanocomposite at different lithiation states in the wide voltage range (WV, 4.8V–2.0 V). **b** Voltage profile for the 10$^{th}$ cycle depending on lithiation state in the WV range, measured at 25 °C and a current density of 20 mA g$^{-1}$. **c** Phase fraction at different lithiation states determined by XRD Rietveld refinement. **d** The formation energy of T-FeF$_3$ and R-FeF$_3$ at different states of lithiation. **e** Experimentally measured voltage profile and DFT calculated reaction voltage for T-FeF$_3$ at different states of lithiation.

T-FeF$_3$ than with R-FeF$_3$ (Supplementary Fig. S9). Conclusively, Rietveld refinement of the XRD pattern of the electrode in the 10$^{th}$ charged state (Supplementary Fig. S10 and Supplementary Note 4) revealed a match with the T-FeF$_3$ phase, including residual LiF and FeF$_2$. These results confirm that the LiF-FeF$_2$ nanocomposite successfully leads to the gradual formation of T-FeF$_3$ while maintaining structural similarity to the mother structure (FeF$_2$). The efficient formation of T-FeF$_3$ is closely governed by the interfacial contact between LiF and FeF$_2$, which facilitates the guided phase transition during cycling (Supplementary Note 5) Notably, tetragonal FeF$_3$ derived from LiF-FeF$_2$ nanocomposites offers practical advantages in full-cell manufacturing[43] compared to the previously reported tetragonal FeF$_3$ phase is formed via the delithiation of Li$_{0.5}$FeF$_3$, particularly due to the safety concerns and chemical instability associated with metallic lithium and lithium-containing negative electrodes[4].

**Reaction mechanism of T-FeF₃**

Reversible and sequential intercalation and conversion reactions were confirmed for T-FeF$_3$ during charge and discharge. First, to investigate the reaction mechanism of T-FeF$_3$, ex situ XRD was performed across various voltage ranges. The voltage regions, including each redox reaction around 4 V and 3 V, were categorized as the wide voltage range (WV, 4.8–2.0 V), the upper voltage range (UV, 4.8–3.4 V), and the lower voltage range (LV, 3.4–2.0 V). Figure 2a shows the ex situ XRD patterns during the charged and discharged process, including the charged state at 4.8 V (red), half-discharged state at 3.4 V (light red), discharged state at 2 V (purple), and recharged state at 4.8 V (blue). Additionally, Fig. 2b shows the voltage profile and state of charge used for structural analysis, and Fig. 2c displays the phase fractions at each state obtained via Rietveld refinement (Supplementary Fig. S13). As shown in Fig. 2a and Supplementary Fig. S14, the ex situ XRD data of T-FeF$_3$ indicate that the diffraction pattern largely retains the

diffraction patterns of *P4₂/mnm* structure throughout the charge-discharge process, while certain peaks exhibit shifts and new peaks gradually emerge (Supplementary Note 6). At points 1 and 2 (within the UV range), only a small expansion (0.03 Å, a 0.3% increase) in the c/3 lattice parameter of the tetragonal phase (Li$_x$FeF$_y$, x < 0.5, and 0.5 < y < 3) was observed, with no noticeable change in phase fraction or occupancy. This gradual shift in the XRD peaks, without a significant change in phase fraction, suggests that the structural evolution in this region is primarily driven by lattice parameter changes rather than a phase transformation, indicating that Li$^+$ insertion occurs within the host structure of T-FeF$_3$ in the UV region rather than triggering a phase transition (Supplementary Fig. S15 and Fig. 2c). However, as shown in Supplementary Fig. S15, during further discharge from point 2 to point 3 (within the LV range), the *a* and *c*/3 lattice parameters of the tetragonal phase increased by 0.8% and 1.6%, respectively, and become similar to those of FeF$_2$. Furthermore, the ratio of Fe to F significantly decreased from 1:2.96 (point 2) to 1:2.59 (point 3). At point 3, the similarity of the lattice parameters and Fe–F ratio of the tetragonal phase and FeF$_2$ as well as the phase increase of LiF and FeF$_2$ with the consumption of the tetragonal phase indicates that the conversion reaction of the tetragonal phase to FeF$_2$ occurs near 3 V.

This result is also consistently verified by the PDF analysis (Supplementary Fig. S16). The PDF patterns at points 1 and 2 were nearly identical, implying that the host structure was maintained. In addition, a shift in the overall pattern and distinct peaks at 2.85 and 4.95 Å, corresponding to LiF, were observed at point 3, indicating structural changes involving LiF formation in the LV region. During the charging process, Li extraction from the lithiated phase and a reconversion reaction occur simultaneously with LiF splitting (points 4 and 5). Given the recovery of the amount of T-FeF$_3$ phase to its initial state after the reconversion reaction, the sequential intercalation and conversion reaction appear to be highly reversible. The reversibility of the reaction

mechanism of the UV and LV regimes was verified through electrochemical-cycle evaluation across various voltage ranges after the initial 10 cycles to form T-FeF$_3$ (Supplementary Fig. S17). At the 100$^{th}$ cycle, the electrode operated by only intercalation within the UV range exhibited a capacity retention of 81%, whereas the electrode that underwent the conversion reaction in the LV regime exhibited a lower capacity retention of 69%. The capacity decrease in the LV regime appears to stem from the voltage being too low to split LiF[41,44,49–51] (Supplementary Note 7) that is necessarily required for the reversible reconversion reaction.

**DFT calculations for the reaction pathways of T-FeF$_3$ and R-FeF$_3$**
Using the DFT calculation, the reaction mechanism of T-FeF$_3$ was further verified by comparing the reaction voltage with experimental data and R-FeF$_3$. Li$_{0.5}$FeF$_3$ (s.g. $P4_2/mnm$) based on a previous report[42] and FeF$_3$ (s.g. $R\bar{3}$-$c$) from Materials Project[52] were used as host structures for lithium intercalation into tetragonal and rhombohedral structures, respectively (Supplementary Data 1–4). Based on previous reports[21,42], Li/Fe disordering in tetragonal structures and stacking faults in rhombohedral structures were also considered. In addition, the phase diagram of the Li–Fe–F system was constructed to evaluate the conversion reaction (Supplementary Fig. S16).

Figure 2d shows the formation energies of Li$_x$FeF$_3$ calculated using DFT. The rhombohedral and tetragonal structures are the most stable at the fully delithiated ($x = 0$) and lithiated ($x = 1$) states, respectively. Upon considering the conversion reaction, it was found that the conversion into LiF and FeF$_2$ is energetically more stable than maintaining the tetragonal LiFeF$_3$ structure at the lithiated state. Notably at $x = 0.5$, the energy differences among various structures are quite small compared to those at fully delithiated and lithiated states. According to Pauling's third rule[53], the structures with edge-sharing or especially face-sharing cationic octahedra are less stable than those with only corner-sharing due to longer cationic distances of corner-sharing Fe octahedra, reducing repulsion between them, a trend confirmed in our calculations (Supplementary Fig. S20). As a result, in the fully delithiated state, the rhombohedral structure, which features exclusively corner-sharing Fe octahedra, is significantly more stable than the tetragonal structure with some edge-sharing connections, as shown in Fig. S21. In addition, in the fully lithiated state, the tetragonal phase, lacking face-sharing Fe and Li octahedra, is more stable than other configurations (Supplementary Fig. S22a–d). Additionally, the structural characteristics of FeF$_2$, which lacks face-sharing octahedra (Supplementary Fig. S22e) and features longer distances between cations than the tetragonal LiFeF$_3$ structure, enhance its stability. This contributes to the lower formation energy of LiF and FeF$_2$ than for one of LiFeF$_3$ structure, leading to energetically favorable decomposition of LiFeF$_3$ into LiF and FeF$_2$.

Based on the formation energy of Li$_x$FeF$_3$ ($0 \leq x \leq 1$) in Fig. 2d, the voltage profiles for the lithiation and delithiation reaction of tetragonal and rhombohedral Li$_x$FeF$_3$ structures are calculated as shown in Fig. 2e and Supplementary Fig. 20. The anion orderings of the tetragonal and rhombohedral structures of Li$_x$FeF$_3$ are distinctly different (Supplementary Figs. S21 and S22). If the energy barrier for the phase transition to a more stable polymorph is high, the metastable phase can be kinetically stabilized, making the transition to a more stable polymorph less likely to occur[54]. Thus, assuming the host anion framework is preserved during cycling, the voltage profiles of tetragonal and rhombohedral structures are calculated based on the topotactic reaction.

In the high-voltage region of Li$_x$FeF$_3$ ($0 \leq x \leq 0.5$), the calculated voltage of the tetragonal structure is 3.64 V for the reaction from ordered tetragonal FeF$_3$ to disordered tetragonal Li$_{0.5}$FeF$_3$. Based on a previous report[42], the disordering between Li and Fe sites occurs in the tetragonal phase after cycling. When considering this disordering, the voltage between disordered tetragonal FeF$_3$ and Li$_{0.5}$FeF$_3$ increases to 3.95 V, which is similar to our experimental results (Fig. 1c and Fig. 2e).

At this point, the energy difference between ordered and disordered Li$_{0.5}$FeF$_3$ is quite small (9.31 meV/atom), indicating that there are no site preferences of Li and Fe at 25 °C (Supplementary Fig. S24, Supplementary Note 8 and Supplementary Data 5). For the rhombohedral structure, the calculated voltage is 3.27 V for the reaction from rhombohedral FeF$_3$ to Li$_{0.5}$FeF$_3$ with stacking faults.

In the low-voltage region ($0.5 \leq x \leq 1$), the reaction mechanisms of tetragonal and rhombohedral structures are different. In the tetragonal structure, a conversion reaction occurs from the disordered tetragonal Li$_{0.5}$FeF$_3$ to LiF and FeF$_2$ with a reaction voltage of 3.27 V. In contrast, the reaction voltage of the rhombohedral structure is 2.72 V through the insertion reaction between the rhombohedral Li$_{0.5}$FeF$_3$ and LiFeF$_3$. This difference originates from the structure of thermodynamically stable FeF$_2$. At a fully lithiated state ($x = 1$), LiF and FeF$_2$, as the conversion reaction products, are the most stable, and thus, the decomposition is energetically more favorable compared to the insertion reaction in both structures.

The structural characteristics of the tetragonal and rhombohedral forms further elucidate these distinct reaction pathways. For the topotactic reaction to occur, the host anion frameworks must be maintained during the reversible reaction. In the case of the tetragonal structure, its anion framework is the same as that of FeF$_2$, only with slight differences in the occupancy and ordering of Fe between tetragonal Li$_x$FeF$_3$ and FeF$_2$. Therefore, the phase transition between tetragonal Li$_x$FeF$_3$ and FeF$_2$ may occur through the reordering of Fe ions accompanied by the formation or splitting of LiF. In contrast, the anion framework of the rhombohedral structure is distinctly different from that of FeF$_2$. Thus, even though the conversion-reaction products, LiF and FeF$_2$, are thermodynamically more stable than rhombohedral LiFeF$_3$, the kinetic barrier to decomposition might be too high to overcome at room temperature. As a result, the topotactic intercalation reaction path is kinetically more favorable than the conversion reaction in the rhombohedral structure during charge and discharge. However, because the conversion reaction is thermodynamically preferred, LiF and FeF$_2$ can also be formed from rhombohedral structures after long-term cycling[21] (Supplementary Fig. S25 and Supplementary Note 9).

**Comparison of voltage hysteresis and compositional inhomogeneity in T-FeF$_3$ and R-FeF$_3$**
To evaluate the effect of maintaining the structural integrity during intercalation and conversion reaction for T-FeF$_3$ on voltage hysteresis and compositional inhomogeneity, we first compared the voltage hysteresis in T-FeF$_3$ and R-FeF$_3$ with galvanostatic intermittent titration technique (GITT) analysis (Fig. 3a). Charge/discharge measurements were performed at 11.2 mAh g$^{-1}$ (corresponding to 0.05 e$^-$ per formula unit) with a current of 20 mA g$^{-1}$, and each relaxation step was maintained for 3 h until the voltage decay rate (dV/dt) dropped below -0.01 mV s$^{-1}$, a criterion commonly used to approximate quasi-equilibrium (Supplementary Fig. S26a). All the analyses were performed after 10 cycles to ensure the evolution of T-FeF$_3$ from LiF-FeF$_2$. Figure 3b shows the voltage gap between the relaxed voltages during charge and discharge, which corresponds to reaction pathway-dependent kinetic hysteresis arising from phase-transition and bond-breaking barriers[55]. Figure 3c presents the extent of voltage change during relaxation, which reflects conventional kinetic polarization related to ion/electron transport. The reaction pathway-dependent kinetic hysteresis was smaller for T-FeF$_3$ than for R-FeF$_3$, and this trend remained consistent even after extended relaxation for 48 h (Supplementary Fig. S26b–d), suggesting that the major voltage gap originates not from transient transport polarization but from slow structural transformations such as phase transition and bond breaking/reformation. This voltage difference is prominent at the end of charge or discharge. Both T-FeF$_3$ and R-FeF$_3$ exhibit larger voltage hysteresis during the charging process than during discharge. For T-FeF$_3$, this is mainly

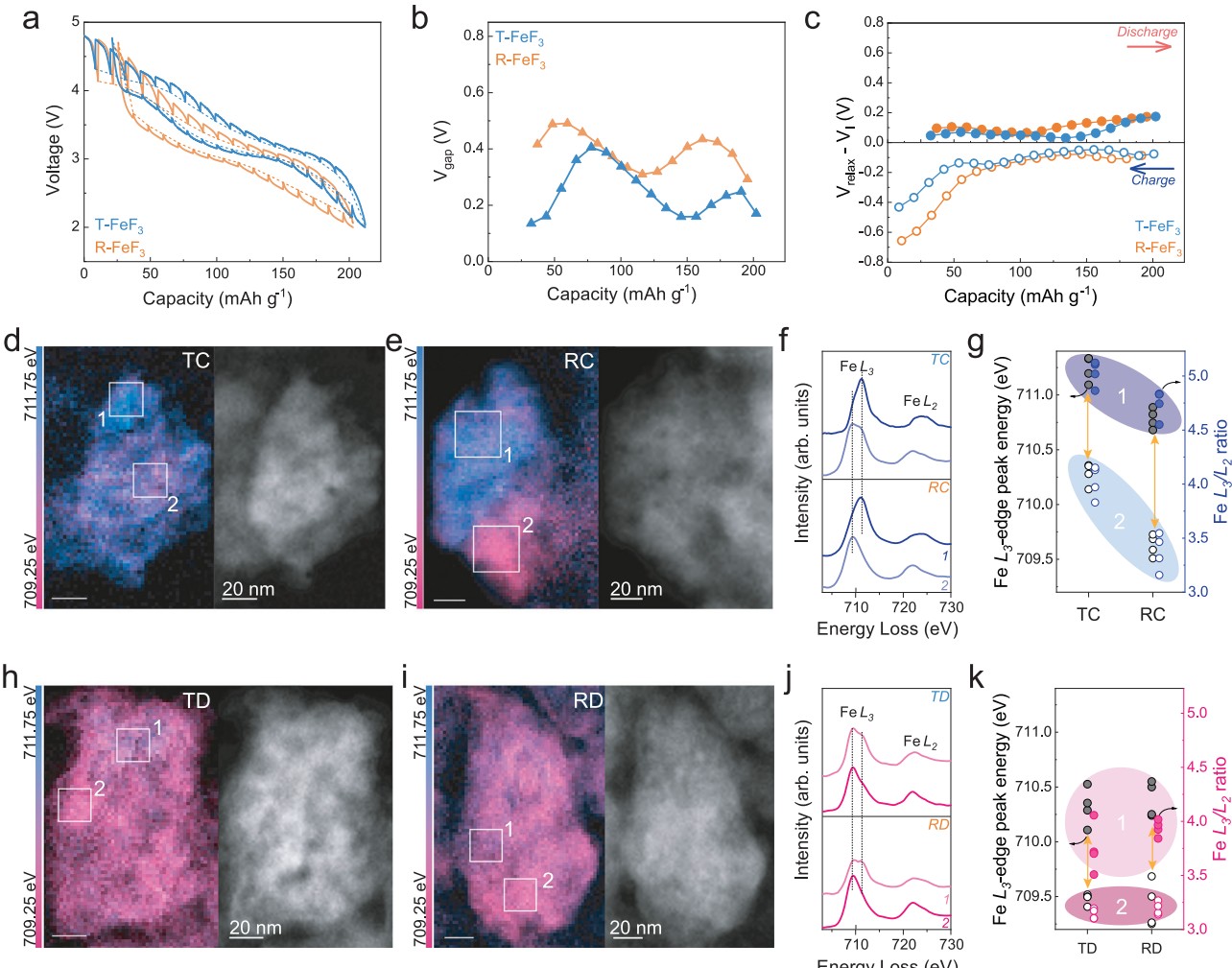

**Fig. 3 | Mitigated voltage hysteresis and compositional inhomogeneity of T-FeF₃ compared to R-FeF₃. a** Galvanostatic intermittent titration technique (GITT) profiles of T-FeF$_3$ and R-FeF$_3$ after 10$^{th}$ cycle. The cells were allowed to relax for 3 h after every 11.2 mAh g$^{-1}$ (corresponding to 0.05 e$^-$/formula unit) discharging/charging at 20 mA g$^{-1}$ at 25 °C. **b** Voltage difference (V$_{gap}$ = V$_{relax, charge}$ − V$_{relax, discharge}$) between charge and discharge steps after the 3 h relaxation at the same state of lithiation of T-FeF$_3$ and R-FeF$_3$. **c** Voltage changes after the 3 h relaxation at different states of discharge and charge at T-FeF$_3$ and R-FeF$_3$. **d, e, h, i,** a scanning TEM (STEM)-electron energy loss spectroscopy (EELS) images of T-FeF$_3$ and R-FeF$_3$ in charged state and discharged state for the energy distribution of the Fe $L_3$-edge peak. The charge state of T-FeF$_3$ (TC) and R-FeF$_3$ (RC). The discharge state of T-FeF$_3$ (TD) and R-FeF$_3$ (RD). These are for the 10$^{th}$ cycle measured at a current density of 20 mA g$^{-1}$ at 25 °C. **f, j** EELS spectra of Fe $L_{3,2}$-edge for each region (1 and 2) in the charged state (TC and RC) and the discharged state (TD and RD). Regions 1 and 2 represent the most oxidized and reduced regions, respectively. **g, k** Fe $L_3$-edge peak energies and $L_3/L_2$ ratios observed in the most oxidized regions (closed symbols) and the most reduced regions (hollow symbols) for different particles (n = 4) at each TC, TD, RC, and RD. The distribution and spectra of the Fe $L_3$-edge peaks for these particles are shown in Supplementary Figs. S27 and S28.

attributed to LiF splitting that occurs during charging (Supplementary Note 7), while in the case of R-FeF$_3$, the increased hysteresis likely results from phase transitions involving long-range diffusion. However, this difference in kinetic hysteresis between T-FeF$_3$ and R-FeF$_3$ is relatively insignificant during both charge and discharge states. This is due to the improved reaction rate and mass transfer in both cases using carbon composites with nano-sized particles. Taken together, despite similar particle size and carbon content, these results indicate that the reduced hysteresis in T-FeF$_3$ compared to R-FeF$_3$ stems from differences in reaction pathways and the reversibility of phase transitions rather than from extrinsic kinetic limitations such as transport resistance[21] (Supplementary Fig. S12 and Supplementary Note 5).

To verify the origin of the low-voltage hysteresis of T-FeF$_3$ regarding compositional inhomogeneity, the distribution of the oxidation state of Fe during charge/discharge was evaluated and compared with that of R-FeF$_3$ using scanning TEM coupled with electron energy-loss spectroscopy (STEM–EELS) analysis. Figure 3d and e present a color map of the energy distribution of the Fe $L_3$-edge peak

(high: blue, low: pink) for charged T-FeF$_3$ (TC) and R-FeF$_3$ (RC), respectively. TC shows a spatially uniform energy distribution, whereas RC exhibits a relatively inhomogeneous distribution. Figure 3f displays the Fe $L_{3,2}$-edge EELS spectra for the local regions having the highest (1) and lowest (2) Fe $L_3$-edge peak energy in TC and RC. The spectra for region 1 are similar for both TC and RC. However, in region 2, TC is characterized by the copresence of peaks at both high (711.25 eV) and low energies (709.5 eV), indicating partial reduction, whereas only a peak at low energy is predominantly displayed for RC, indicating that it is almost fully reduced to Fe$^{2+}$. The larger deviation of the iron oxidation state for the charged state, as indicated by the Fe $L_3$-edge energy and $L_3/L_2$ intensity ratio, is commonly observed across various particles (Fig. 3g and Supplementary Fig. S27). TC has a Fe $L_3$-edge energy variation of 0.95 eV between the oxidized and reduced regions, whereas RC shows a larger energy variation of 1.16 eV, which is also consistently observed in the $L_3/L_2$ intensity ratio. The presence of compositional inhomogeneity was also observed even after the discharge; however, the difference of inhomogeneity between T-FeF$_3$ and

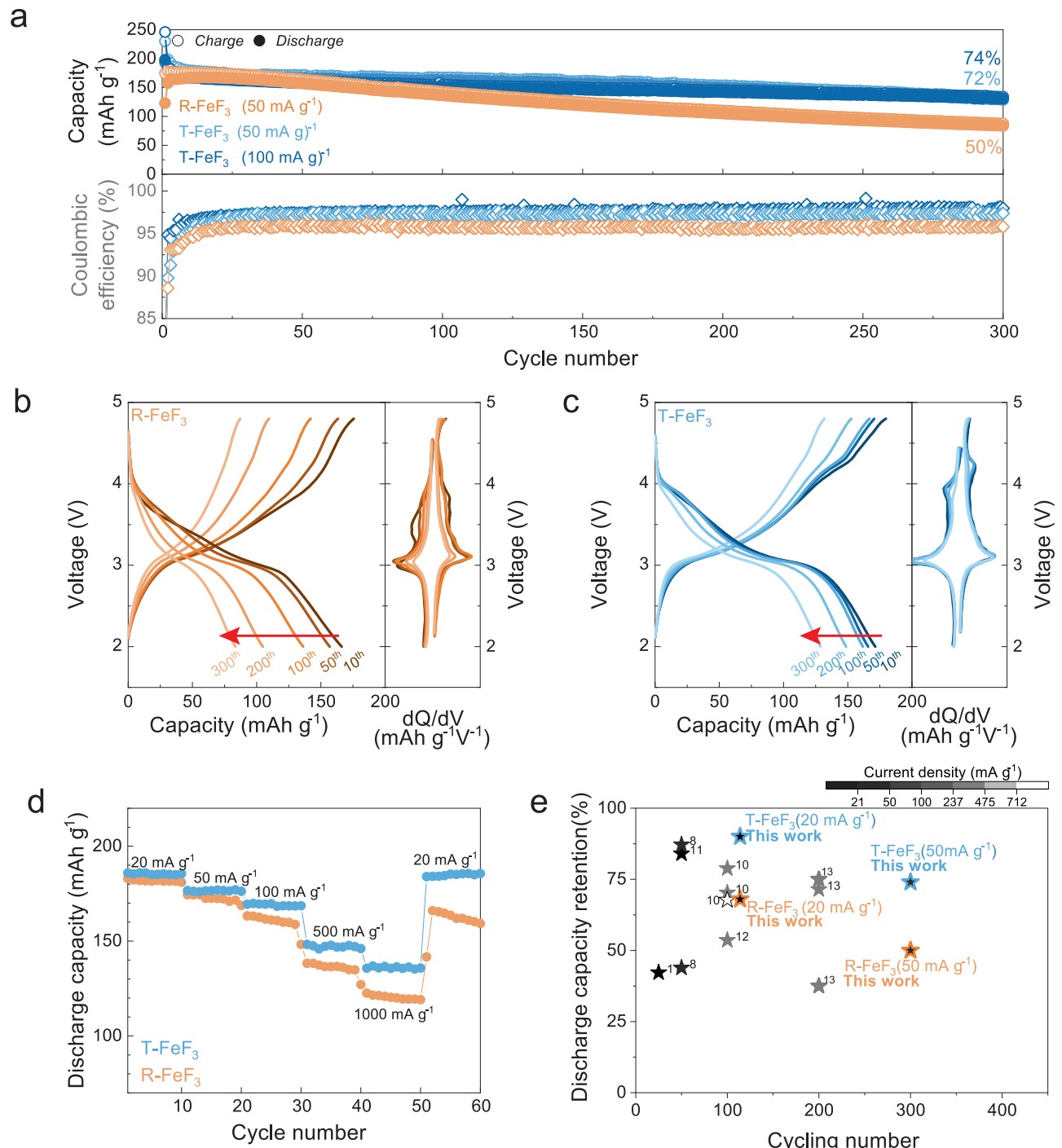

**Fig. 4 | Electrochemical performance of T-FeF₃. a** Cycle stability of T-FeF₃ and FeF₃, measured at 25 °C and a current density of 50 mA g⁻¹. **b, c** Electrochemical profile of T-FeF₃ and R-FeF₃ at various cycles. **d** Rate performance of LiF-FeF₂ and FeF₃. **e** Comparison of capacity retention of T-FeF₃ and iron fluoride materials mixed with carbon. The electrochemical stability of iron fluoride materials was evaluated in the 1-electron transfer range (Discharge cutoff voltage -2 V).

R-FeF₃ was less than that for the charged state. As shown in Fig. 3h and i, discharged T-FeF₃ (TD) and R-FeF₃ (RD) exhibit similar Fe $L_3$-edge peak energy distributions. Both TD and RD have Fe $L_{3,2}$-edge spectra reduced to $Fe^{2+}$ in region 2 and partially oxidized spectra in region 1 (Fig. 3j). The Fe $L_3$-edge energy and $L_3/L_2$ intensity ratio observed across various particles indicate that TD and RD have similar Fe oxidation state distributions (Fig. 3k and Supplementary Fig. S28). Therefore, the comprehensive results indicate that composition inhomogeneity is more pronounced at the charged state than the discharged state, which implies that the composition inhomogeneity is governed by the reversibility of the reconversion reaction rather than

the conversion reaction. This result is consistent with the larger value of mitigated voltage hysteresis at the charged state (0.28 V) than the discharged state (0.12 V) for T-FeF₃ compared to R-FeF₃ (Fig. 3b). Thus, mitigated compositional inhomogeneity of T-FeF₃ is expected to result in not only low-voltage hysteresis but also highly reversible cycle stability compared to R-FeF₃.

## Reversibility of T-FeF₃ accompanying intercalation and conversion reaction

The cycle stability of T-FeF₃ was evaluated and compared with that of R-FeF₃ to validate the reversibility of T-FeF₃ intercalation and

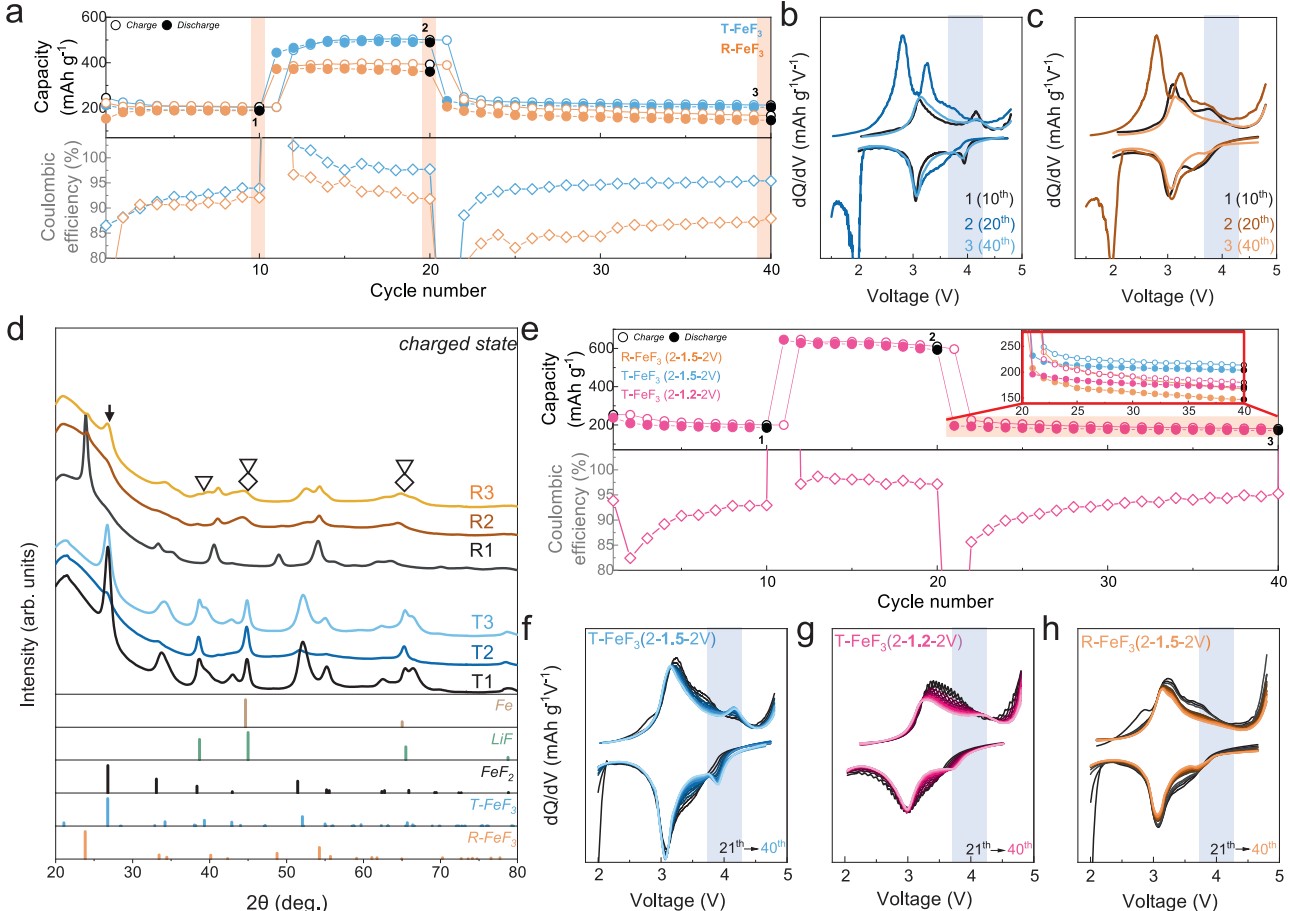

**Fig. 5 | Reversibility of T-FeF₃ even under deep discharge. a–c** The cycle ability and differential analysis of the voltage profile of T-FeF₃ and R-FeF₃ with repeated changing discharge cut-off voltage (2-1.5-2 V), measured at 25 °C and a current density of 20 mA g⁻¹. **d** The XRD pattern of charged states for T-FeF₃(T) and R-FeF₃(R) at each point in the cycle was measured at the changing cut-off voltage. Point 1 (10ᵗʰ cycle), point 2 (20ᵗʰ cycle), and point 3 (40ᵗʰ cycle). The tetragonal phase (arrows), LiF (inverted triangle), and Fe metal (diamond). **e** The cycle ability of T-FeF₃ with repeated changing discharge cut-off voltage (2-1.2-2 V). Changes of Differential analysis of the voltage profile of T-FeF₃ and R-FeF₃ from 20 to 40 cycles. **f, g** Blue and pink are the change conditions of 2-1.5-2 V and 2-1.2-2 V discharge cut-off voltage of T-FeF₃, respectively. These were measured at 25 °C and a current density of 20 mA g⁻¹. **h** Yellow is the change condition of 2-1.5-2 V discharge cut-off voltage of R-FeF₃.

conversion sequential reaction (Fig. 4a). R-FeF₃ exhibited continuous capacity decay, maintaining only 50% of its capacity after 300 cycles. In contrast, T-FeF₃ demonstrated improved capacity retention of 72% after 300 cycles (Fig. 4b and c). Even at a higher specific current of 100 mA g⁻¹, stable capacity retention of 74% was maintained (Fig. 4a). Figure 4d displays the rate performance of T-FeF₃ at various specific currents ranging from 20 to 1000 mA g⁻¹. At the high specific current of 1000 mA g⁻¹, R-FeF₃ showed a capacity of 120 mAh g⁻¹, whereas T-FeF₃ maintained an improved capacity of 136 mAh g⁻¹ (Supplementary Fig. S29). Moreover, R-FeF₃ exhibited capacity decay across all specific currents and a significant drop in capacity upon returning to 20 mA g⁻¹. However, T-FeF₃ displayed stable capacity retention overall, demonstrating its reversibility under various specific current conditions. This cycling performance of T-FeF₃ is attributed to the minimized structural evolution, with the analogous anion framework maintained despite undergoing both Li⁺ insertion and conversion reactions, which is closely linked to a recent report that maintaining the structural integrity of an amorphous structure after a conversion reaction can ensure structural and electrochemical reversibility[56]. Thus, minimal structural change leads to better capacity retention for T-FeF₃ compared to previously reported carbon-composited R-FeF₃ (Fig. 4e and Supplementary Fig. S30).

The reversibility of T-FeF₃ is maintained even at deep discharge of the conversion reaction to LiF and Fe metal. Figure 5a shows the capacity retention when the low cut-off voltage range is continuously varied back and forth from 2 to 1.5 V. The initial 10 cycles were preceded within a 4.8–2.0 V voltage range (point 1, 10ᵗʰ cycle) to form T-FeF₃. Then, the voltage range was changed to 4.8–1.5 V for 10 cycles to induce drastic structural evolution, forming the Fe metal phase (point 2, 20ᵗʰ cycle). Subsequently, the voltage range was recovered to 4.8–2.0 V for another 20 cycles (point 3, 40ᵗʰ cycle) to validate the reversibility. The characteristic 4 V redox feature of T-FeF₃ was absent in the differential curve, and the electrochemical profile is quite analogous to R-FeF₃ at point 2 (Fig. 5b). During deep discharge, Fe metal conversion occurs[19,21,31,57,58] (Supplementary Figs. S31–33 and Supplementary Note 10), which involves long-range diffusion and could exacerbate the compositional inhomogeneity[19,21]. As shown in Supplementary Fig. S34, when cycling under deep discharge conditions involving the conversion reaction to LiF and Fe, both R-FeF₃ and T-FeF₃ commonly experience capacity degradation. Interestingly, despite these harsh conditions (deep discharge), T-FeF₃ exhibits a reversible recovery of its characteristic 4 V redox process at Point 3. Consequently, the electrochemical profile of Point 3 closely resembles that observed at Point 1. This feature is repeatedly observed during cycling with a periodically altering cut-off voltage (Supplementary Fig. S35). In contrast, R-FeF₃ did not exhibit the 4 V redox feature at any point (Fig. 5c and Supplementary Fig. S36). This finding indicates that T-FeF₃ can be reversibly recovered even after

undergoing a conversion reaction involving severe structural evolution. Furthermore, note that the lithiated state after deep discharge appears to be the same as LiF and Fe metal for both T-FeF$_3$ and R-FeF$_3$, given the similarity of the electrochemical profile at point 2; however, it could in fact be different.

To reveal the structural reversibility of T-FeF$_3$ during the reconversion reaction even after deep discharging cycles, the charged-state structures of T-FeF$_3$ and R-FeF$_3$ at each point in Fig. 5a are analyzed (Fig. 5d). The labels T and R refer to T-FeF$_3$ and R-FeF$_3$, respectively, with the subsequent numbers indicating the corresponding points in Fig. 5a. At point 1, each structure of T-FeF$_3$ and R-FeF$_3$ was well maintained respectively (T1 and R1). After deep discharge, R-FeF$_3$ showed very broad diffraction peaks of LiF and Fe metal due to the nanosized domain caused by the previously observed phase displacement involving long-range diffusion (R2)[21,59,60]. At point R3, unlike at R1, not the rhombohedral phase but the tetragonal phase (arrows) was observed with LiF (inverted triangle) and Fe metal (diamond). This indicates that irreversible phase separation by long-range diffusion has occurred (Supplementary Note 10), and the tetragonal phase that has resulted from this process is difficult to activate due to the compositional inhomogeneity. On the other hand, the trace of the tetragonal phase was observed for T-FeF$_3$ even after deep discharge at T2 (26.7°, 33.4°, 51.8°, and 55.12°), and the T-FeF$_3$ phase reversibly recovered to its original state at point T3 while maintaining high crystallinity compared to R3. This result indicates that T-FeF$_3$ exhibits better structural reversibility than R-FeF$_3$. Even in electrochemical evaluation protocols starting with deep discharge, the formation of the 4 V redox feature was observed (Supplementary Fig. S36). Furthermore, despite starting with a deep discharge, the capacity retention rate was better than that of R-FeF$_3$ (Supplementary Fig. S38). Therefore, the reversibility of T-FeF$_3$ may be attributed to the seed of the remaining tetragonal phase even in the discharged state. To confirm this assumption, electrochemical evaluation was conducted in an over-deep discharge voltage range (4.8–1.2 V) to remove the seeds of the tetragonal phase (Supplementary Fig. S39). Figure 5e shows the capacity retention under this over-deep discharge condition. At point 3 in Fig. 5e, following the over-deep discharge, significant capacity decay and failure to clearly recover the 4 V redox peak were observed (Fig. 5f–h and Supplementary Fig. S40). These results imply that T-FeF$_3$ can achieve reversible structural recovery if the phase seed remains, whereas irreversible structural changes occur in the case of R-FeF$_3$, regardless of the presence of the phase seed due to the irreversible reactions accompanying the reaction mechanism.

## Discussion

Figure 6a illustrates the reaction mechanism of the LiF-FeF$_2$ nanocomposite as predicted from both experimental and computational studies.

Initial cycling: Phase transition into T-FeF$_3$

During initial cycling, the LiF-FeF$_2$ nanocomposite electrochemically forms T-FeF$_3$ via LiF splitting:

$$LiF + FeF_2 \rightarrow FeF_{3-\delta, tetra} + Li^+ \quad (1)$$

As the cycle progresses, a 4 V plateau gradually forms in the electrochemical profile (Fig. 1c). Correspondingly, new peaks at 34.8°, 40°, and 66.7° are also observed; however, none of the XRD patterns in the charged state significantly deviate from those of FeF$_2$ (Fig. 1d). This finding indicates the electrochemical formation of a tetragonal phase, which demonstrates that the LiF–FeF$_2$ adopts a tetragonal structure induced by the tetragonal structure of FeF$_2$ rather than the thermodynamically stable rhombohedral structure.

Points 1-2 (UV range): Intercalation reaction

As shown in Fig. 6, the formed T-FeF$_3$ undergoes an insertion reaction to produce Li$_x$FeF$_y$ (x < 0.5, 2.5 < y < 3):

$$FeF_{3-\delta, tetra} \rightarrow Li_xFeF_y \quad (2)$$

Despite a 0.3% increase in the c/3 lattice parameter within the UV range, the phase fraction and PDF patterns remain almost constant (Supplementary Figs. S15 and S16). Moreover, the capacity in the UV range after the initial 10 cycles for forming T-FeF$_3$ demonstrates capacity retention of 93% at the 100$^{th}$ cycle (Supplementary Fig. S17). These results suggest a highly reversible reaction through Li$^+$ insertion, which is further confirmed by DFT analysis. The Li ions are inserted into T-FeF$_3$ with Li/Fe site disordering, showing a high reaction voltage near 4 V, which is consistent with the experimental results (Fig. 2e).

Points 2–3 (LV range): Conversion reaction

Upon further discharge (points 2–3), Li$_x$FeF$_y$ (x < 0.5, 2.5 < y < 3) undergoes a conversion reaction to form LiF and FeF$_2$:

$$Li_xFeF_y \rightarrow LiF + FeF_2 \quad (3)$$

During further discharge from Point 2 to Point 3, the consumption of the lithiated Li$_x$FeF$_y$ phase results in the increase of the phase fraction of LiF and FeF$_2$ (Fig. 2c) and the formation of LiF peaks in the PDF pattern (Supplementary Fig. S16), indicating the occurrence of the conversion reaction in the LV range. Additionally, DFT calculations show that the decomposition products, LiF and FeF$_2$, are the most stable in the fully lithiated state (x = 1) (Fig. 2d). Thus, the phase transition occurs from Li$_x$FeF$_3$ to FeF$_2$ through the reordering of Fe ions with LiF formation.

Points 3–5: Reconversion reaction

During the charging process, T-FeF$_3$ is gradually formed via LiF splitting:

$$LiF + FeF_2 \rightarrow FeF_{3, tetra} + Li^+ \quad (4)$$

The gradual progression of the reconversion reaction during the charging process can be confirmed by the changes in the phase fraction refined from ex situ XRD (Fig. 2c). Furthermore, the reversibility in the LV region shows a capacity retention of 69%, unlike 93% in the UV region, due to the absence of a high-voltage operation necessary for LiF splitting[41,44,49–51] (Supplementary Fig. S17 and Supplementary Note 7). This result verifies that the charging process involves a reconversion reaction along with LiF splitting from Points 3 to 5.

By comparing the reaction mechanism between R-FeF$_3$[21] and T-FeF$_3$, both materials commonly show intercalation and conversion reactions (Fig. 6b). Although the final products of the conversion reaction (down to 2 V) are the same in both cases (Fig. 2a and Supplementary Fig. S41), the FeF$_2$ and LiF converted from R-FeF$_3$ are formed in small amounts in isolated regions, resulting in poor interfacial contact (Supplementary Fig. S42). Moreover, from a structural standpoint, T-FeF$_3$ exhibits much higher reversibility than R-FeF$_3$ due to its closer structural similarity to FeF$_2$. This difference is further validated by STEM–EELS analysis performed after 100 cycles. Since it was measured at 100 mA g$^{-1}$, both T-FeF$_3$ and R-FeF$_3$ show a mixed Fe$^{2+}$/Fe$^{3+}$ state in the charged state. However, R-FeF$_3$ displays a significantly greater presence of reduced Fe$^{2+}$ regions and lower Fe$^{3+}$ intensity even in the most oxidized areas, indicating more pronounced compositional inhomogeneity compared to T-FeF$_3$ (Supplementary Fig. S43). R-FeF$_3$ undergoes a structural change from a corner-sharing FeF$_6$ group structure (rcp) to an edge-sharing tetragonal phase (tcp) during charging and discharge[19,21,57,61] It is understood that the significant structural difference between the two structures causes an irreversible

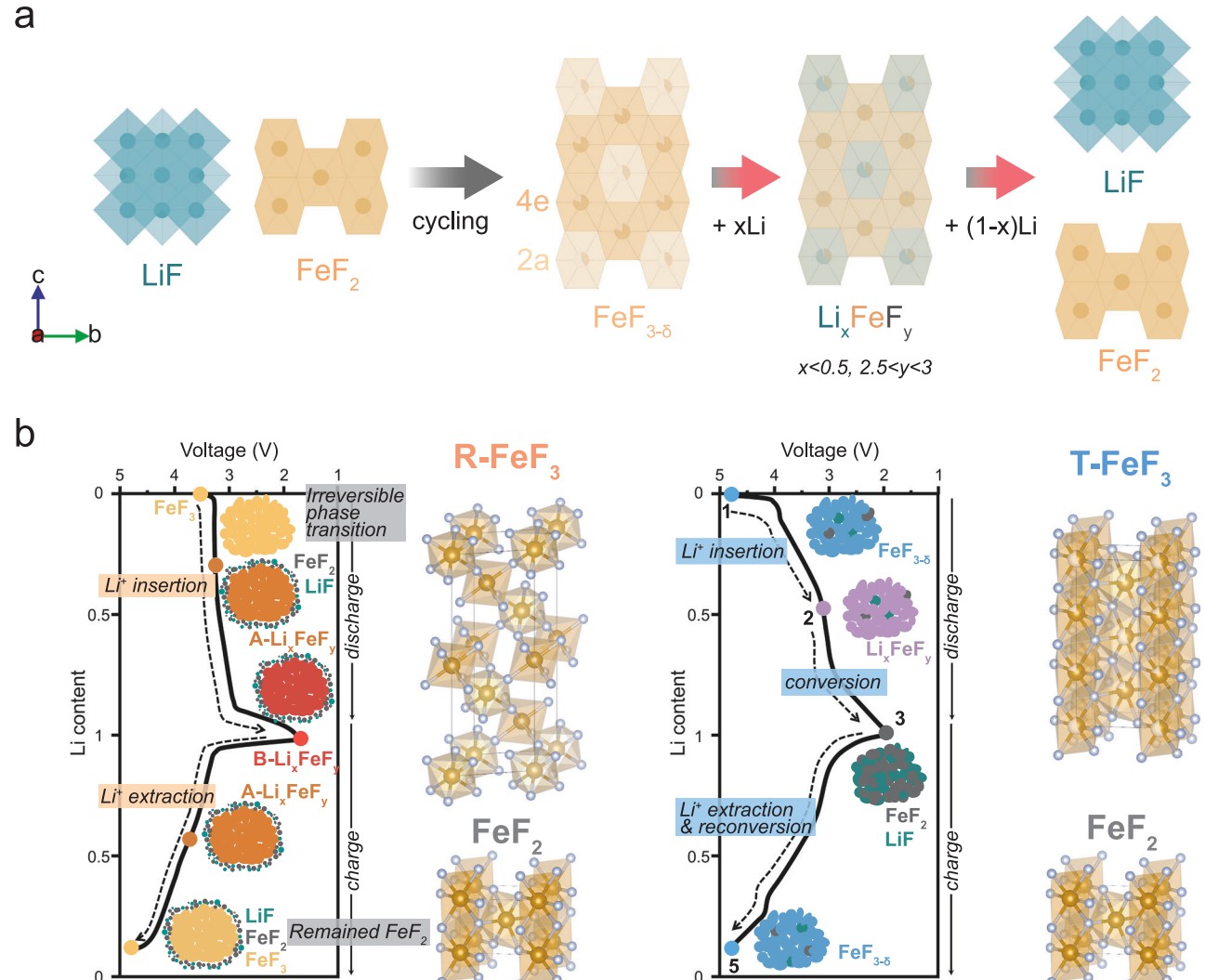

**Fig. 6 | Reaction mechanism of T-FeF₃ and comparison with R-FeF₃. a** Crystal structures of the LiF-FeF₂ nanocomposite, T-FeF₃, and the intermediate phase derived from experimental data and DFT calculations. **b** Schematic illustration comparing the reaction mechanisms of R-FeF₃ (reproduced from ref. 21 with permission from Springer Nature) and T-FeF₃. Brown and silver balls indicate Fe and F ions, respectively. The schematic is based on electrochemical profiles measured at 25 °C in the voltage range of 4.8–2.0 V with a current density of 20 mA g⁻¹.

phase transition, which leads to compositional inhomogeneity[21] (Supplementary Fig. S25 and Supplementary Note 9). This inherent structural mismatch and poor interfacial contact between electrochemically formed LiF and Fe species further hinder the reformation of the tetragonal FeF₃ phase upon recharging. Additionally, when LiF and Fe metal of the discharge state reconvert, the fcc(LiF)−tcp transition is preferred over the fcc(LiF)-rcp transition. This leads to promotion of the irreversible FeF₂ phase formation and worsens the compositional inhomogeneity compared to the initial state[21]. Moreover, the compositional inhomogeneity can lead to localized variations in the reaction kinetics, with slower kinetics in inhomogeneous regions resulting in greater hysteresis[19,20]. Therefore, despite proposed strategies such as nanosizing and compositing with conductive materials to kinetically suppress the voltage hysteresis, unresolved hysteresis is still observed[19,21,26−31]. In contrast, T-FeF₃ can easily transform into FeF₂ through metal migration within the same anion framework, involving relatively shorter diffusion compared to the rcp−tcp transition. Therefore, unlike R-FeF₃, T-FeF₃ shows high reversibility due to the structural similarity to the phase after the conversion reaction, resulting in mitigated compositional inhomogeneity and voltage hysteresis. In this context, evading the phase-displacement reaction

accompanying long-distance diffusion can not only mitigate compositional inhomogeneity and voltage hysteresis but also ensure electrochemical reversibility in the conversion reaction.

We emphasize the potential of tetragonal FeF₃ (T-FeF₃) derived from LiF-FeF₂ nanocomposites to address the issues of compositional inhomogeneity and voltage hysteresis in iron fluoride positive materials. LiF-FeF₂ nanocomposite successfully guided the phase transition into metastable T-FeF₃ while maintaining the structural framework of FeF₂. Due to the structural integrity, lower voltage hysteresis is achieved for T-FeF₃ under conversion reaction into FeF₂ with mitigated compositional inhomogeneity. This result starkly contrasts with that for R-FeF₃, which inevitably suffers from compositional inhomogeneity induced by irreversible phase transitions into FeF₂. As a result, although T-FeF₃ undergoes sequential insertion and conversion reactions, this material maintained 72% and 74% of its capacity at 50 and 100 mA g⁻¹, respectively, over 300 cycles. In addition, its energy efficiency improves from 81% for R-FeF₃ to 87% for T-FeF₃. Moreover, the reversibility of the T-FeF₃ phase recovery was further validated even after conversion into LiF and Fe metal phases if the seeds of the tetragonal phase remain. Our study suggests that harnessing the conversion reaction that maintains structural integrity can resolve the

chronic issues of large voltage hysteresis and low structural reversibility for conversion reaction electrode materials. Furthermore, our approach of using nanocomposites to design positive materials could offer a new direction as a model system for developing rechargeable batteries with high specific energy using conversion chemistry. Hereafter, further investigations, such as developing micron-scale particles and optimizing bulk synthesis methods, are needed to enable the practical application of designed materials.

## Methods

### Synthesis of $LiF$-$FeF_2$ nanocomposite and R-$FeF_3$

In an Ar-filled environment, a mixture of iron (II) fluoride (anhydrous, 98%, Alfa Aesar) and lithium fluoride (≥99.99% trace metals basis, Sigma-Aldrich) powders in a 1:1.2 molar ratio was sealed. The sealed powders were ball-milled using a Pulverisette 7 premium line high-energy ball mill at 500 rpm for 48 h in an 80 mL silicon nitride jar with silicon nitride balls at a ball-to-powder weight ratio of 20:1. To the ball-milled mixture, 20 wt.% of graphite powder (−200 mesh, 99.9995% metals basis, Alfa Aesar) was added and then ball-milled at 500 rpm for 12 h under the same milling conditions to synthesize the $LiF$-$FeF_2$ nanocomposite. R-$FeF_3$ was prepared by sealing iron (III) fluoride (anhydrous, 97% min, Alfa Aesar) and graphite powder in a 1:0.2 mass ratio in an Ar-charged environment and then high-energy ball milling at 500 rpm for 48 h using the same silicon nitride jar and balls with a ball-to-powder weight ratio of 20:1.

### Electrochemistry

The prepared active material, conductive agent (Super P), and binder (polyvinylidene fluoride, Solef 5130, Solvay) in a 7:2:1 weight ratio were dispersed in N-methyl-2-pyrrolidone (NMP, >99.5%, Sigma-Aldrich) using a planetary centrifugal mixer (AR-100, Thinky). The prepared slurry was cast onto Al foil (99.3% purity, 20 μm thickness) with a loading level of approximately 1.5 mg cm$^{-2}$, and a positive electrode with a diameter of 12.5 mm (puncher, P02M, Rohtec) was subsequently prepared. Coin cells (CR2032) were assembled in an Ar-filled glove box ($O_2$ < 0.1 ppm, $H_2O$ < 0.1 ppm). A 1 M $LiPF_6$ solution in a 1:1 volume ratio of ethylene carbonate and dimethyl carbonate (Enchem) was used as the electrolyte. To fully wet the separator and electrodes, 150 μL of the electrolyte was injected into each coin cell. Glass microfiber filters (Whatman GF/C, 1.2 μm pore size, 0.26 mm thick) served as the separators (19 mm), and Li-metal foil (0.1 mm thick, FMC) was used as the negative material. Electrochemical analyses were conducted at 25 °C using a battery testing system (WBCS 3000, WonATech). For electrochemical measurements in a fluorine-free environment, polyacrylonitrile (PAN, average $M_v$ 150000, Sigma-Aldrich) binder and 1 M $LiClO_4$ electrolyte (Enchem) were used instead of polyvinylidene fluoride binder and 1 M $LiPF_6$ electrolyte. GITT was conducted at 20 mA g$^{-1}$, with the cell allowed to relax for 3 h after each 11.2 mAh g$^{-1}$ (equivalent to 0.05 e$^-$ per formula unit) discharge/charge step. The rate performance was tested at specific currents of 20, 50, 100, 500, and 1000 mA g$^{-1}$. All the electrochemical evaluations were performed after 10 cycles at 20 mA g$^{-1}$ to ensure the formation of T-$FeF_3$ and were verified by repeating three times to obtain reasonable data.

### Computation

First principal calculations were performed using the density functional theory (DFT) as implemented in the Vienna Ab initio Simulation Package (VASP)[62]. The spin-polarized generalized gradient approximation (GGA) with the Perdew-Burke-Ernzerhof (PBE) functional[63] was applied. To correct the self-interaction error in iron 3 d states, we added the Hubbard-type U parameter (GGA + U) of 4.0 eV as employed in a previous computational study[21]. A plane-wave basis set was used with an energy cutoff of 520 eV, and 3 × 3 × 3 k-point grid on the

Gamma-centered mesh were used for the calculations. All structures were fully relaxed until the forces on each atom were below 0.05 eV.

All distinct Li-vacancy orderings for tetragonal and rhombohedral structures within the unit cell of $Li_xFeF_3$ (x = 0, 0.5, 1), including 16 formula units, were generated, and the 30 configurations with the lowest electrostatic energy at each Li content were conducted by GGA + U. For the tetragonal structure, the host structure was $Li_{0.5}FeF_3$ (s.g. $P4_2/mnm$), and Li/Fe disordering was also considered based on the previous report[42]. To consider the Li/Fe disordering in the tetragonal structure, the structures with Li-Fe orderings were generated using the same enumeration technique for Li-vacancy ordering, and the energies of these structures were calculated. The host structures for rhombohedral structures were $FeF_3$ (s.g. $R3-c$) from Materials Project[52] with stacking faulted structures that were suggested in a previous study[21]. To generate additionally Li inserted rhombohedral $FeF_3$ and tetragonal $Li_{0.5}FeF_3$ structure, the cation insertion algorithm was applied[53]. The charge densities of each host structure were calculated, and the local minima sites of them were suggested through this algorithm. In addition, the ground states for tetragonal and rhombohedral $Li_xFeF_3$ structures were known as antiferromagnetic state (AFM)[21], thus, all tetragonal and rhombohedral structures were calculated with antiferromangetic or weak ferromagnetic response. The voltage profiles were obtained from the DFT energies of the most stable configurations at each Li contents as[64]

$$V = -\frac{E(Li_{x2}FeF_3) - E(Li_{x1}FeF_3) - (x_2 - x_1)E(Li)}{(x_2 - x_1)F}$$

where $E(Li_xFeF_3)$ and $E(Li)$ are the DFT energy of the most stable $Li_xFeF_3$ structure and bcc Li metal, and F is the Faradaic constant.

For analysis of phase stability of the Li-Fe-F system, the structures in Li-Fe-F system were obtained from Materials Project[52] and formation energies of each calculated structure at their respective compositions were calculated based on the energy of pure elements such as Li, Fe, and F. The phase diagram was plotted based on the convex hull from formation energies.

### Characterization

For clear structural analysis, the XRD patterns of the samples were measured at the 6D UNIST-PAL beamline at the Pohang Accelerator Laboratory (PAL). The patterns were collected over a 2θ range of 10° to 110° with an X-ray beam wavelength of 1.5406 Å. For ex situ XRD pattern measurements, cells were disassembled in an Ar atmosphere at an average temperature of 24 ± 2 °C after reaching specific voltages, and the electrodes were retrieved and washed with diethyl carbonate. The cleaned electrodes were collected from the Al foil, sealed in capillaries, and stored in double vacuum packaging until measurement. All XRD patterns were recorded with an exposure time of approximately 2 min. Each sample was measured twice to obtain the XRD patterns, ensuring the removal of noise, outliers, and spikes caused by high-energy X-rays. Rietveld refinement of the XRD data was performed using FullProf software. Total scattering data for PDF analysis were obtained using a PANalytical Empyrean with Ag-Kα radiation source, a Rh Kβ filter, and a GaliPIX$^{3D}$ detector. The powder sample was loaded in a 0.4-mm glass capillary in an Ar-filled glovebox, and each PDF data set was collected for 36 h. The data reduction was performed using PDFGetX3 software[65], and real-space data were fitted using PDFgui[66]. The damping factor associated with this instrumental configuration was calibrated by refining the structure of a silicon sample as a reference.

The particle size of the samples was observed through scanning Electron microscope (SEM) images taken at 10 kV using a field-emission scanning electron microscope (Nova Nano SEM, FEI). To characterize the crystallographic features, transmission electron

microscopy (TEM) was employed. To perform ex situ analyses, the coin cells were disassembled after reaching specific voltages, and the recovered electrodes were washed with dimethyl carbonate (DMC). Samples separated from the aluminum current collector were dispersed in DMC using ultrasonic treatment and loaded onto TEM grids. The loaded TEM sample grids were vacuum sealed until measurement to minimize air exposure. All these processes were performed under an Ar atmosphere at an average temperature of $24 \pm 2\,°C$. TEM images and STEM-EELS measurements were conducted using a Cs-corrected JEM-ARM300F (JEOL). For elemental mapping in STEM-EELS, energy dispersion was set to 0.25 eV per channel at 160 kV. To minimize electron beam damage to the samples and stabilize the beam, a beam shower was performed for approximately 15 min before measurements.

X-ray Absorption Spectroscopy (XAS) was performed to observe the local structure and oxidation state of Fe. The Fe *K*-edge spectra were measured in transmission mode using a Si(111) double-crystal monochromator at the PAL 6D beamline. Energy calibration was carried out through a standard iron foil, and the reference was measured simultaneously. For ex situ XAS analysis, the measured coin cells were disassembled, and the electrodes were recovered and washed with dimethyl carbonate (DMC) in an Ar atmosphere at an average temperature of $24 \pm 2\,°C$. To prevent air exposure, the electrodes were sealed with Kapton tape and stored in double vacuum packaging until measurement. The XANES and EXAFS spectra were processed using Athena software, with Fourier transforms of the EXAFS spectra performed in the k-range of $3.0 \sim 11.5\,\text{Å}^{-1}$ weighted by $k^2$. EXAFS fitting was conducted using Artemis software. The F *K*-edge spectra were collected in total electron yield (TEY) mode at the 10D KIST bending magnet beamline of PLS-II under a base pressure of $1 \times 10^{-9}$ Torr. The spectra were normalized to the incident photon flux with an energy resolution of 0.1 eV.

## Data availability

All data supporting the findings of this study are available in the manuscript and its Supplementary Information. The atomic coordinates of the optimized computational structures are provided as plain text files in Supplementary Data 1–5, packaged in a single zip file. No custom code was used in this study. Source data are provided with this paper.

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

## Acknowledgments

This research was supported by the Nano & Material Technology Development Program through the National Research Foundation of Korea (NRF) funded by Ministry of Science and ICT (RS-2024-00406724 (S.-K. Jung) and RS-2024-00435493 (D.-H. Seo)). The computational work was supported by the Supercomputing Center/Korea Institute of Science and Technology Information with supercomputing resources, including technical support (KSC-2023-CHA-0024 (D.-H. Seo)). This work was supported by a National Research Foundation of Korea (NRF) grant funded by the Korean government (MSIT) (No. 2022R1C1C1006575 (S.-K. Jung) and 2023R1A2C2008242 (D.-H. Seo)).

## Author contributions

H. Jo and S.-K. Jung conceived the research design and were responsible for the manuscript preparation. M. Gong and D.-H. Seo performed the DFT calculations. S.Y. Kim performed the PDF measurements and analysis. All the authors contributed to the discussion of this work. S.-K. Jung supervised the project.

## Competing interests

The authors declare no competing interests.
