## [Peer Review File · Nature Communications]

Guided phase transition for mitigating voltage hysteresis of iron fluoride positive electrodes in lithium-ion batteries

Corresponding Author: Professor Sung-Kyun Jung

Version 0:

Reviewer comments:

Reviewer #1

(Remarks to the Author)

This manuscript reported a tetragonal FeF₃ electrochemically derived from LiF/FeF₂ nanocomposite that alleviates iron fluoride cathode's voltage hysteresis and improves its cycling stability. Using a combination of detailed DFT calculations and experimental characterizations, the authors reveal the structure evolutions of tetragonal FeF₃ and rhombohedral FeF₃, as well as the relationship between composition homogeneity and voltage polarization during the phase conversion process. Unfortunately, the reaction mechanisms of metastable tetragonal FeF₃ and rhombohedral FeF₃ have been reported in the previous literatures (Chem. Mater., 2021, 33, 868-880.; Nat. Mater., 2021, 20, 841-850.), and this work provides a supplement. Regarding the pristine materials, the mixtures of LiF and redox-active Fe-species (e.g., FeO, FeF₂, FeO) are obtained by mechanical ball milling, and it is a common method for preparing pre-lithiated iron fluorides. Besides, the optimized electrochemical performance (e.g., capacity and lifespan) is not impressive, even when compared to the iron fluoride cathodes confined with single-electron transfer. Overall, this manuscript cannot meet the requirements of novelty and impressive scientific discoveries for publication on Nature Communications. Here are a few comments that will hopefully help improve the manuscript:

- (1) Why can't LiF/FeF₂ formed by electrochemical lithiation form the tetragonal FeF₃ after recharging? Which one has the better electric/ionic contact, LiF/FeF₂ obtained by mechanical ball milling or LiF/FeF₂ formed by electrochemical lithiation of conventional FeF₃?
- (2) Considering that LiFeF₃ is thermodynamically metastable, why does further lithiation of Li_{0.5}FeF₃ not result in the phase separation for the case of rhombohedral FeF₃ in Figure 2e? The authors proposed that no LiF is extruded from R-FeF₃ even when one lithium-ion is inserted (Figure 6b). Can the authors verify this through solid physical characterization?
- (3) In Figure 3a-c, why are the voltage hysteresis and chemical inhomogeneity during the charge state more serious than those during the discharge state for both T-FeF₃ and R-FeF₃?
- (4) Could the authors give a clearer explanation of the statement "the alleviation of voltage hysteresis in T-FeF₃ compared to R-FeF₃ is due not to kinetics but to thermodynamic origins"? Won't the kinetic barriers (i.e., electron/ion diffusion) contribute to the compositional inhomogeneity of reaction products or the proposed "thermodynamic hysteresis" in this manuscript?
- (5) In Figure 5a-c, why does R-FeF₃ suffer a more obvious overcharge problem with a lower coulombic efficiency than T-FeF₃?
- (6) Most of the electrochemical studies in this manuscript limit the redox reactions to single-electron transfer for iron fluoride, which causes a low reversible capacity < 200 mAh g⁻¹. Can the guided tetragonal structure have a significant effect on the high-capacity FeF₃ relying on three-electron transfer reactions?

Reviewer #2

(Remarks to the Author)

In the present article, the authors prepared tetragonal FeF₃ (T-FeF₃) with the aim of reducing reaction inhomogeneity and stabilizing the structure. This material maintains structural stability during the reaction process and exhibits lower voltage hysteresis. The authors explored the reaction mechanism through a series of characterizations and DFT calculations, and the investigation of the reaction mechanism is really impressive. However, there are several questions that need to be addressed before the paper can be published.

1. For example, in the battery cycling tests, the authors selected a voltage range of 4.8–2.0 V. Generally speaking, voltage tests above 4.5 V are not particularly common in lithium-ion batteries. The authors are kindly asked to explain why the upper limit of the voltage range in this study was chosen as 4.8 V. Furthermore, since the electrolyte used in the experiments is a

- common standard electrolyte, could the electrolyte introduce any other issues during the testing process?
2. Please add the corresponding electrochemical performance of this work to Supplementary Table S5 for the convenience of comparison by the readers.
 3. The authors mentioned in the introduction the advantages of FeF_3 as a typical conversion-type cathode material, such as high energy density and low cost. The theoretical capacity of FeF_3 as a cathode material for lithium-ion batteries is 712 mAh/g. However, the actual discharge capacity of the material prepared by the authors under low current density testing (e.g., 20 mA/g in the rate capability test) is much lower than the theoretical capacity of FeF_3 . Please explain the reason for this discrepancy.
 4. In Supplementary Table S5, different studies have used different voltage ranges for the cycling tests. For FeF_3 , varying voltage ranges imply significant differences in the dominant reaction mechanism during the electrochemical process (1Li-3Li). One of the effects of this difference is the variation in cycling stability. Given the significant differences in testing conditions, is it rigorous to compare the performance under these conditions?
 5. What is the theoretical capacity of the metastable FeF_3 prepared by the authors within the voltage range used in the tests mentioned in the paper?
 6. The GITT test can also be used to assess the diffusion level of lithium ions. Is there any difference in the lithium ions diffusion levels between the different materials in the paper?
 7. If long-cycle tests with a voltage range similar to that in Supplementary Table S5 are conducted, the research results will be more convincing.
 8. Why the cycle stability of T- FeF_3 is 72% after 300 cycles? It is related to the size of FeF_2 ?
 9. If the time of ball-milling would affect the size distribution of FeF_2 and LiF and the final electrochemical performance?
 10. In ex-XRD of Fig. 2a, is there any description for this figure?
 11. "only a change (0.03 Å, 0.3% increase) in the $c/3$ lattice parameter of the tetragonal phase (Li_xFeF_y , $x < 0.5$, and $0.5 < y < 3$). It was observed without a change in the phase fraction or occupancy" However, from XRD analysis, this phenomenon can not be observed. The peaks at 34.8 and 40° are also hard to observe, even after 10 cycles.
 12. The scale bar in Fig. S22 Figure S23 should be given.
 14. For R- FeF_3 and T- FeF_3 samples, EELS should be provided after 300 cycles to support the conclusion in Fig. 6b.
 15. Please add the figure number here." (Supplementary Fig. S33). Fig. shows the capacity retention under this over-deep discharge condition."

Reviewer #3

(Remarks to the Author)

The manuscript presents a significant advancement in lithium-ion battery technology by introducing a guided phase transition approach for iron fluoride cathode materials. The transformation of a LiF- FeF_2 nanocomposite into metastable tetragonal FeF_3 (T- FeF_3) demonstrates reduced voltage hysteresis and enhanced structural reversibility. This is a substantial contribution to the field, as it addresses persistent issues in conversion-reaction-based cathode materials, including compositional inhomogeneity and poor cycle stability. The experimental data, supported by detailed DFT calculations, convincingly demonstrate the superiority of T- FeF_3 over rhombohedral FeF_3 (R- FeF_3) in terms of electrochemical performance and structural stability.

This work is of high significance to the field of energy storage and conversion. The proposed T- FeF_3 material, derived from a nanocomposite strategy, offers a novel and effective solution to mitigate voltage hysteresis and structural degradation, problems widely reported in conversion-reaction materials. The findings align with and extend prior studies, such as those focusing on $\text{Na}_{0.7}\text{CoO}_2$ and LiF-MnO systems, by introducing a new paradigm in the design of cathode materials. The comparison to the established literature is thorough, and the study builds on previous insights by providing a thermodynamic and kinetic framework for improving reversibility and cycling stability. The references cited are appropriate and comprehensive.

The conclusions drawn are well-supported by the experimental and computational results. The authors demonstrate the formation and stability of T- FeF_3 through XRD, TEM, and EXAFS analyses, as well as the material's enhanced performance through electrochemical testing. The DFT calculations further strengthen the argument, providing insights into the thermodynamic stability and reaction pathways of T- FeF_3 compared to R- FeF_3 . However, additional details on the reproducibility of the synthesis method and its scalability could strengthen the conclusions further.

The methodology is sound and meets the expected standards for research in this domain. The combination of experimental techniques and theoretical modeling is comprehensive and provides a holistic understanding of the material properties and performance. The methods for nanocomposite synthesis, electrochemical testing, and structural characterization are described in sufficient detail to allow replication. The use of advanced tools like STEM-EELS and GITT for analyzing compositional homogeneity and hysteresis adds rigor to the study.

The data analysis is robust, and the interpretations are logical and well-aligned with the results. The study effectively correlates structural properties with electrochemical performance, providing a clear narrative for the advantages of T- FeF_3 . However, the manuscript could benefit from additional discussion on potential limitations, such as the impact of prolonged cycling at high current densities or the effect of varying the nanocomposite composition.

While the manuscript is comprehensive, additional long-term cycling data at varying current densities and a comparison with other conversion-reaction cathode materials could provide further validation. Additionally, an exploration of the impact of impurities or variations in the precursor materials would strengthen the generalizability of the findings.

The authors are encouraged to provide more details on the scalability of the synthesis process and the potential industrial applications of T-FeF₃.

A clearer explanation of the limitations of the R-FeF₃ system in practical applications would help contextualize the advantages of T-FeF₃.

The figures are well-designed, but the captions could be more detailed to aid standalone interpretation.

Version 1:

Reviewer comments:

Reviewer #1

(Remarks to the Author)

I have carefully reviewed the responses regarding the issues. The authors believe that the “guided phase transition strategy” addresses voltage hysteresis and structural reversibility, which are critical challenges in iron fluoride cathodes. However, according to the response to comment 6, this proposed strategy only exhibits favorable effects on the intercalation reaction of iron fluoride cathodes, while demonstrating limited effectiveness for conversion reactions. Intercalation-type iron fluoride cathodes have been extensively studied (e.g., *Adv. Mater.*, 2019, 31, 1905146; *Nano energy*, 2023, 108, 108181) and have enabled high-rate (up to 100C) and long-life (up to 1000 cycles with a high capacity retention of 84.1%) single-electron transfer reactions, as well as high-loading cathodes (5.3 mg/cm²). Therefore, the electrochemical performance achieved in this study is deemed unsatisfactory.

On the other hand, voltage hysteresis and structural reversibility have not been the key challenges for iron fluorides limited to the intercalation reaction stage, but rather for their conversion reactions to deliver high-capacity and high-energy. From a long-term perspective, activating the conversion reaction could enhance the competitiveness of iron fluorides compared with currently commercial cathodes, especially in terms of capacity and energy density. Unfortunately, the proposed strategy showed a limited effect on the critical conversion reactions. Overall, this work provides a supplement to the study on the intercalation reaction of T-FeF₃ cathode, but fails to make impressive progress in the development of iron fluoride cathodes. Therefore, I still cannot recommend this manuscript for publication in *Nature Communications*. Additionally, several issues remain unresolved. Please refer to the comments below.

(1) Regarding comment 1, I hope the authors can demonstrate the differences in chemical distribution and electronic/ionic contact between the LiF-FeF₂ composites formed by mechanical ball milling and electrochemical reaction through solid physical characterization.

(2) Regarding comment 4, voltage hysteresis within a reversible cycle is inherently a kinetic phenomenon. This reference (*Nat. Chem.*, 2021, 13, 1070.) may help the authors to understand the origin of voltage hysteresis observed during GITT testing.

(3) Regarding comment 5, a distinct additional oxidation behavior can be observed around 4.8 V in the differential profile of the charge curve for R-FeF₃ (Fig. 5c). This is likely related to the oxidative decomposition of the electrolyte under high-voltage application, rather than the LiF separation proposed by the authors. This overcharge phenomenon becomes more severe when conversion reactions are involved for both T-FeF₃ and R-FeF₃ (Fig. R3), leading to their low coulombic efficiency. The high charge cutoff voltage generally induces undesired interfacial side reactions and deeper structural changes, thereby reducing the cycling life and coulombic/energy efficiency. Does the formation of T-FeF₃ require such a high charge cutoff voltage of 4.8 V?

Reviewer #2

(Remarks to the Author)

After reviewing the authors' responses, I am satisfied with their revisions. So I recommend this article can be published in *Nature Communications*.

Reviewer #3

(Remarks to the Author)

Excellent revision, nothing more to say, I accept its publication

Version 2:

Reviewer comments:

Reviewer #1

(Remarks to the Author)

The authors have addressed my main concerns. This manuscript can be accepted now.

Response to reviewers

Guided phase transition for mitigating voltage hysteresis of iron fluoride cathode materials in lithium-ion batteries

Hyoi Jo,^{‡a} Minjeong Gong^{‡ b}, Se Young Kim^c, Dong-Hwa Seo^{*b} and Sung-Kyun Jung^{*d,e,f}

Affiliation

a. Institute for Battery Research Innovation, Seoul National University, 1 Gwanak-ro, Gwanak-gu, Seoul 08826, Republic of Korea

b. Department of Materials Science and Engineering, Korea Advanced Institute of Science and Technology (KAIST), 291 Daehak-ro, Yuseong-gu, Daejeon 34141, Republic of Korea

c. Energy Storage Research Center, Korea Institute of Science and Technology (KIST), 14 Gil 5 Hwarang-ro, Seongbuk-gu, Seoul, 02792 Republic of Korea

d. Department of Materials Science and Engineering, College of Engineering, Seoul National University, 1 Gwanak-ro, Gwanak-gu, Seoul 08826, Republic of Korea

e. School of Transdisciplinary Innovations, Seoul National University, 1 Gwanak-ro, Gwanak-gu, Seoul 08826, Republic of Korea

f. Research Institute of Advanced Materials, Seoul National University, 1 Gwanak-ro, Gwanak-gu, Seoul 08826, Republic of Korea

* Corresponding to: dseo@kaist.ac.kr and naecard@snu.ac.kr

‡ These authors contributed equally to this work.

Reviewer #1(Remarks to the Author):

Comments to the Author

This manuscript reported a tetragonal FeF₃ electrochemically derived from LiF/FeF₂ nanocomposite that alleviates iron fluoride cathode's voltage hysteresis and improves its cycling stability. Using a combination of detailed DFT calculations and experimental characterizations, the authors reveal the structure evolutions of tetragonal FeF₃ and rhombohedral FeF₃, as well as the relationship between composition homogeneity and voltage polarization during the phase conversion process. Unfortunately, the reaction mechanisms of metastable tetragonal FeF₃ and rhombohedral FeF₃ have been reported in the previous literatures (*Chem. Mater.*, 2021, 33, 868-880.; *Nat. Mater.*, 2021, 20, 841–850.), and this work provides a supplement. Regarding the pristine materials, the mixtures of LiF and redox-active Fe-species (e.g., Fe⁰, FeF₂, FeO) are obtained by mechanical ball milling, and it is a common method for preparing pre-lithiated iron fluorides. Besides, the optimized electrochemical performance (e.g., capacity and lifespan) is not impressive, even when compared to the iron fluoride cathodes confined with single-electron transfer. Overall, this manuscript cannot meet the requirements of novelty and impressive scientific discoveries for publication on *Nature Communications*. Here are a few comments that will hopefully help improve the manuscript.

Author reply:

We sincerely appreciate the reviewer's insightful comments and the opportunity to clarify the novelty and significance of our work. While previous studies (*Chem. Mater.* **33**, 868-880 (2021); *Nat. Mater.* **20**, 841–850, (2021)) have explored the reaction mechanisms of metastable tetragonal FeF₃ (T-FeF₃) and rhombohedral FeF₃ (R-FeF₃), respectively, our study introduces “guided phase transition strategy” that directly addresses voltage hysteresis and structural reversibility which are critical challenges in iron fluoride cathodes. In addition, we focus on systematically demonstrating how T-FeF₃, electrochemically derived from a LiF-FeF₂ nanocomposite, retains structural integrity during charge and discharge involving phase transitions.

Furthermore, while mechanical ball milling is commonly used for preparing pre-lithiated iron fluorides as reviewer commented, our approach in here is fundamentally distinct in purpose. Rather than pre-lithiation, we strategically utilize structurally analogous precursors (FeF₂ which belongs to tetragonal crystal system) to facilitate the formation of T-FeF₃ via electrochemical splitting of LiF.

Supplementary Fig. S29. (a, b) Comparison of capacity retention and post-cycle discharge capacity for iron fluoride materials composited with carbon and FeF₃ (rhombohedral structure) and other Fe-based conversion-reaction materials (see Supplementary Table S5).

In terms of electrochemical performance, although T-FeF₃ does not surpass all previous reports, it exhibits excellent structural reversibility and long-term cycling stability (Supplementary Fig. S29). Additionally, deep discharge experiments (Fig. 5a-d) confirm that T-FeF₃ maintains its structure even under extreme cycling conditions, a key advantage over conventional iron fluoride cathodes. These findings can contribute to advancing phase transition control strategies for iron fluoride-based electrodes and provide valuable insights for the development of next-generation conversion cathodes.

Fig. 5: Superior reversibility of T-FeF₃ even under deep discharge. **a-c,** The cycle ability and differential analysis of voltage profile of T-FeF₃ and R-FeF₃ with repeated changing discharge cut-off voltage (2-1.5-2V). **d,** The XRD pattern of charged states for T-FeF₃(T) and R-FeF₃(R) at each point in the cycle measured at the changing cut-off voltage. Point 1 (10th cycle), point 2 (20th cycle), and point 3 (40th cycle).

We have carefully addressed each of the reviewer's valuable comments and made necessary modifications to improve the clarity of our manuscript. We hope these revisions satisfactorily address the concerns raised and that our study is now suitable for publication.

Comments

1. *Why can't LiF/FeF₂ formed by electrochemical lithiation form the tetragonal FeF₃ after recharging? Which one has the better electric/ionic contact, LiF/FeF₂ obtained by mechanical ball milling or LiF/FeF₂ formed by electrochemical lithiation of conventional FeF₃?*

Author reply:

We sincerely appreciate your insightful question. We understand your comment as: "Does LiF and FeF₂ formed electrochemically from R-FeF₃ exhibit poorer electric/ionic contact compared to LiF-FeF₂ composite obtained by mechanical ball milling, thereby preventing the formation of T-FeF₃ upon recharging?"

Our *ex-situ* XRD analysis (Supplementary Fig. S7) reveals that, during the initial discharge cycle, R-FeF₃ primarily undergoes Li insertion within its rhombohedral structure rather than a full conversion reaction into LiF and FeF₂. However, as seen in Supplementary Fig. S21c, the diffraction peaks corresponding to the tetragonal phase gradually increase over successive cycles. This suggests that the amount of LiF and FeF₂ formed during the early discharge cycles is extremely scarce, therefore, the electrochemically formed LiF and FeF₂ in R-FeF₃ inevitably forms poor connections and do not efficiently reform into T-FeF₃ upon recharging due to compositional inhomogeneity, and the necessity for long-range diffusion to facilitate structural reorganization.

In particular, LiF is an electronic and ionic insulator, meaning that if sufficient interfacial contact with FeF₂ is not established during charging, LiF splitting becomes inefficient, ultimately hindering the reformation of T-FeF₃. In contrast, LiF-FeF₂ synthesized via mechanical ball milling features a homogeneously distributed nanoscale composite, where the interfacial contact between FeF₂ and LiF is significantly enhanced, facilitating electron and ion transport. This improved connectivity allows LiF decomposition to occur more readily, promoting the structural reversibility necessary for effective T-FeF₃ formation.

Supplementary Fig. S7. *Ex-situ* XRD patterns of LiF-FeF₂ and R-FeF₃ electrodes at pristine and 10th charged/discharged states.

Supplementary Fig. S21. (c) *ex-situ* XRD patterns for the 10th, 50th, and 100th charge states of R-FeF₃.

Therefore, mechanically milled LiF-FeF₂ provides superior electronic/ionic contact compared to LiF-FeF₂ formed via electrochemical lithiation, making it more favorable for reversible T-FeF₃ formation during charge-discharge cycles.

Based on the reviewer's comments, the contents of the manuscript have been modified as follows:

Original text (Page 12)

R-FeF₃ undergoes a structural change from a corner-sharing FeF₆ group structure (rcp) to an edge-sharing tetragonal phase (tcp) during charging and discharge^[19,21,56,60] It is understood that the significant structural difference between the two structures causes an irreversible phase transition, which leads to compositional inhomogeneity²¹ (Supplementary Fig. S24 and Supplementary Note 7) Additionally, when LiF and Fe metal of the discharge state revert, the fcc(LiF)–tcp transition is preferred over the fcc(LiF)-rcp transition. This leads to promotion of the irreversible FeF₂ phase formation and worsens the compositional inhomogeneity compared to the initial state^[21]. Moreover, the compositional inhomogeneity can lead to localized variations in the reaction kinetics, with slower kinetics in inhomogeneous regions resulting in greater hysteresis^[19,20]. Therefore, despite proposed strategies such as nanosizing and compositing with conductive materials to kinetically suppress the voltage hysteresis, unresolved hysteresis is still observed^[19,21,26–31]. In contrast, T-FeF₃ can easily transform into FeF₂ through metal migration within the same anion framework, involving relatively shorter diffusion compared to the rcp–tcp transition. Therefore, unlike R-FeF₃, T-FeF₃ shows high reversibility due to the structural similarity to the phase after the conversion reaction, resulting in mitigated compositional inhomogeneity and voltage hysteresis. In this context, evading the phase-displacement reaction accompanying long-distance diffusion can not only mitigate compositional inhomogeneity and voltage hysteresis but also ensure electrochemical reversibility in the conversion reaction.

Revised text (Page 13)

R-FeF₃ undergoes a structural change from a corner-sharing FeF₆ group structure (rcp) to an edge-sharing tetragonal phase (tcp) during charging and discharge^[19,21,56,60] It is understood that the significant structural difference between the two structures causes an irreversible phase transition, which leads to compositional inhomogeneity²¹ (Supplementary Fig. S25 and Supplementary Note 9). **This inherent structural mismatch and poor interfacial contact between electrochemically formed LiF and Fe species further hinder the reformation of the tetragonal FeF₃ phase upon recharging.** Additionally, when LiF and Fe metal of the discharge state revert, the fcc(LiF)–tcp transition is preferred over the fcc(LiF)-rcp transition. This leads to promotion of the irreversible FeF₂ phase formation and worsens the compositional inhomogeneity compared to the initial state^[21]. Moreover, the compositional inhomogeneity can lead to localized variations in the reaction kinetics, with slower kinetics in inhomogeneous regions resulting in greater hysteresis^[19,20]. Therefore, despite proposed strategies such as nanosizing and compositing with conductive materials to kinetically suppress the voltage hysteresis, unresolved hysteresis is still observed^[19,21,26–31]. In contrast, T-FeF₃ can easily transform into FeF₂ through metal migration within the same anion framework, involving relatively shorter diffusion compared to the rcp–tcp transition. Therefore, unlike R-FeF₃, T-FeF₃ shows high reversibility due to the structural similarity to the phase after the conversion reaction, resulting in mitigated compositional inhomogeneity and voltage hysteresis. In this context, evading the phase-displacement reaction

accompanying long-distance diffusion can not only mitigate compositional inhomogeneity and voltage hysteresis but also ensure electrochemical reversibility in the conversion reaction.

Comments

2. *Considering that LiFeF₃ is thermodynamically metastable, why does further lithiation of Li_{0.5}FeF₃ not result in the phase separation for the case of rhombohedral FeF₃ in Figure 2e? The authors proposed that no LiF is extruded from R-FeF₃ even when one lithium-ion is inserted (Figure 6b). Can the authors verify this through solid physical characterization?*

Author reply:

Thank you for the careful review of our manuscript, which helped to improve the clarity of our paper. As your valuable insight pointed out, we agree that the explanation for the minor phase change from R-FeF₃ to LiF/FeF₂ that occurs during the discharge process is insufficient.

The absence of significant phase separation during lithiation of R-FeF₃ can be attributed to kinetic constraints within the rhombohedral structural flexibility, and thermodynamic stabilization. A recent study (*ACS Appl. Energy Mater.* **7**, 15, 6437-6446 (2024)) reports that at room temperature, R-FeF₃ undergoes lithiation without major phase separation due to high activation energy barriers for phase transformation, structural adaptability within the cubic lattice, and a relatively low thermodynamic driving force for decomposition. However, at higher temperatures, increased ion mobility and lower activation barriers facilitate phase separation, leading to the formation of FeF₂ and LiF. Therefore, at room temperature, kinetic limitations, structural adaptability, and low thermodynamic driving forces can stabilize LiFeF₃ as a single-phase material, delaying phase separation.

Our results confirm that the lithiation of R-FeF₃ proceeds primarily *via* Li insertion into the rhombohedral structure without significant phase separation, as confirmed by ex-situ XRD analysis (Supplementary Fig. S7). However, we confirm the phase transition to LiF/FeF₂ on the short-length scale, as we fully agree with the reviewer's concerns. TEM analysis of R-FeF₃ after the 10th discharge cycle (Fig. R1) reveals the minor presence of FeF₂ and LiF phases, indicating that some degree of phase separation does occur during lithiation, albeit to a limited extent.

Supplementary Fig. S7. *Ex-situ* XRD patterns of LiF-FeF₂ and R-FeF₃ electrodes at pristine and 10th charged/discharged states.

Fig. R1. (a) TEM images at discharge state of R-FeF₃. (b) Azimuthal integration of FFT pattern for overall images. (c, d) FFT patterns at discharge state of R-FeF₃. The green and white boxes indicate areas where the LiF and FeF₂ are predominantly present, respectively.

However, as shown in Supplementary Fig. S21c, the diffraction peaks corresponding to LiF and FeF₂ become progressively more pronounced with continued cycling, supporting the gradual growth of the separated phases. This observation suggests that while initial lithiation mainly proceeds through insertion, repeated cycling promotes partial conversion and localized phase separation, validating the reviewer's point.

Supplementary Fig. S21. (c) *ex-situ* XRD patterns for the 10th, 50th, and 100th charge states of R-FeF₃.

Based on the reviewer's insightful suggestions, we have added TEM analysis and revised the text. Furthermore, we have revised Fig. 6b to incorporate LiF extrusion in the illustration, ensuring that our depiction aligns more accurately with our findings. We appreciate the reviewer's detailed observations, which have helped us improve the clarity and accuracy of our manuscript. In response to the reviewer's comment, we inserted further descriptions and carefully revised Fig. 6 as follows.

Original text (Page 13)

By comparing the reaction mechanism between R-FeF₃^[21] and T-FeF₃, both materials commonly show intercalation and conversion reactions (Fig. 6b). Although the product of the conversion reaction (until 2 V) is the same as FeF₂ in both cases, the reversibility is much better for T-FeF₃ than R-FeF₃ due to the similarity of the structure. R-FeF₃ undergoes a structural change from a corner-sharing FeF₆ group structure (rcp) to an edge-sharing tetragonal phase (tcp) during charging and discharge^[19,21,56,60] It is understood that the significant structural difference between the two structures causes an irreversible phase transition, which leads to compositional inhomogeneity²¹ (Supplementary Fig. S24 and Supplementary Note 7).

Revised text (Page 13)

By comparing the reaction mechanism between R-FeF₃^[21] and T-FeF₃, both materials commonly show intercalation and conversion reactions (Fig. 6b). Although the product of the

conversion reaction (until 2 V) is the same as FeF_2 and LiF in both cases (Fig. 2a and Supplementary Fig. S40), the reversibility is much better for T- FeF_3 than R- FeF_3 due to the structural similarity with FeF_2 . This difference is further validated by STEM-EELS analysis performed after 100 cycles. Since it was measured at 100 mA g^{-1} , both T- FeF_3 and R- FeF_3 show a mixed $\text{Fe}^{2+}/\text{Fe}^{3+}$ state in the charged state. However, R- FeF_3 displays a significantly greater presence of reduced Fe^{2+} regions and lower Fe^{3+} intensity even in the most oxidized areas, indicating more pronounced compositional inhomogeneity compared to T- FeF_3 (Supplementary Fig. S41). R- FeF_3 undergoes a structural change from a corner-sharing FeF_6 group structure (rcp) to an edge-sharing tetragonal phase (tcp) during charging and discharge^[19,21,56,60] It is understood that the significant structural difference between the two structures causes an irreversible phase transition, which leads to compositional inhomogeneity²¹ (Supplementary Fig. S25 and Supplementary Note 9).

Added supplementary Figure

Supplementary Fig. S40. (a) TEM images at discharge state of R- FeF_3 . **(b)** Azimuthal integration of FFT pattern for overall images. **(c, d)** FFT patterns at discharge state of R- FeF_3 . The green and white boxes indicate areas where the LiF and FeF_2 are predominantly present, respectively.

Original Figure

Fig. 6: Reaction mechanism of T-FeF₃ and comparison with R-FeF₃. **a.** Reaction mechanism of T-FeF₃ derived from experimental and DFT calculation results. **b.** Comparison of reaction mechanisms between R-FeF₃ and T-FeF₃.

Revised Figure

Fig. 6: Reaction mechanism of T-FeF₃ and comparison with R-FeF₃. a. Crystal structures of the LiF-FeF₂ nanocomposite, T-FeF₃, and the intermediate phase derived from experimental data and DFT calculations. b. Comparison of reaction mechanisms between R-FeF₃ [21] and T-FeF₃.

Comments

3. In Figure 3a-c, why are the voltage hysteresis and chemical inhomogeneity during the charge state more serious than those during the discharge state for both T-FeF₃ and R-FeF₃?

Author reply:

Thank you for your insightful question regarding voltage hysteresis. As the reviewer pointed out, both T-FeF₃ and R-FeF₃ exhibit greater voltage hysteresis during the charge process compared to discharge, and this can be attributed to the following reasons.

In the case of R-FeF₃, the discharge process involves a structural transition from rhombohedral to tetragonal FeF₂, which requires long-range diffusion of ions and leads to partial irreversible changes. Additionally, the small amounts of FeF₂ and LiF that form during discharge remain inactive materials due to poor contact (Fig. R1 and Supplementary Fig. S21). Such poor contact will exacerbate the compositional inhomogeneity during the charging process, thereby increasing voltage hysteresis.

Fig. R1. (a) TEM images at discharge state of R-FeF₃. (b) Azimuthal integration of FFT pattern for overall images. (c, d) FFT patterns at discharge state of R-FeF₃. The green and white boxes indicate areas where the LiF and FeF₂ are predominantly present, respectively

Supplementary Fig. S21. (c) *ex-situ* XRD patterns for the 10th, 50th, and 100th charge states.

For T-FeF₃, the discharge process involves Li-ion insertion followed by a conversion reaction into LiF/FeF₂ (Fig.6a), resulting in a more uniform contact state compared to the LiF/FeF₂ formed from R-FeF₃ in the discharged state. Moreover, since FeF₂, the discharge product, shares structural similarity with T-FeF₃, the reconversion reaction during charging occurs more smoothly.

Fig. 6a. Reaction mechanism of T-FeF₃ derived from experimental and DFT calculation results.

However, charging T-FeF₃ requires the separation of LiF, a highly stable material, which necessitates a high-voltage environment. As shown in Supplementary Fig. S14, LiF splitting is inefficient under low-voltage conditions (3.4V-2.0V), hindering the reconversion reaction and leading to capacity loss. This is further evidenced by the 111th charging voltage profile, where a large overpotential is observed when the upper cut-off voltage is restored from 3.4 V to 4.8 V after low-voltage cycling. This behavior indicates that the accumulation of LiF significantly increases voltage hysteresis during the charging process. Notably, LiF splitting contributes more to voltage hysteresis during discharge than during charging, as previously reported in the literature (*Nat. Energy*, **2**, 16208 (2017); *Adv. Funct. Mater.* **31**, 2009133 (2021); *J. Power Sources*, **329**, 406 (2016); *J. Power Sources*, **354**, 34 (2017); *ECS Meet. Abstr.* **58**, 87 (2014)).

Supplementary Fig. S14. (b, d) Voltage profile and cycle performance of T-FeF₃ in the order of WV-LV-WV range. Arrows have been added to indicate the progression of voltage profiles across different cycles.

Based on these results, we conclude that the higher voltage hysteresis during charging compared to discharge in both T-FeF₃ and R-FeF₃ is due to LiF splitting in T-FeF₃ and phase transition induced by long-range diffusion in R-FeF₃, respectively.

As the reviewer's comments, the original content and figures have been revised as follows:

Original text (Page 7-8)

To evaluate the effect of maintaining the structural integrity during intercalation and conversion reaction for T-FeF₃ on voltage hysteresis and compositional inhomogeneity, we first compared the voltage hysteresis in T-FeF₃ and R-FeF₃ with galvanostatic intermittent titration technique (GITT) analysis (Fig. 3a). All the analyses were performed after 10 cycles to ensure the evolution of T-FeF₃ from LiF-FeF₂. Fig. 3b shows the difference in relaxed voltage during charge and discharge corresponding to thermodynamic hysteresis, and Fig. 3c presents the extent of voltage change during relaxation corresponding to kinetic hysteresis. T-FeF₃ exhibited a smaller thermodynamic hysteresis compared to R-FeF₃, which is prominent at the end of charge or discharge. In addition, the difference in kinetic hysteresis between T-FeF₃ and R-FeF₃ is relatively insignificant during both charge and discharge states. This is due to the improved reaction rate and mass transfer in both cases using carbon composites with nano-sized particles. This finding implies that the alleviation of voltage hysteresis in T-FeF₃ compared to R-FeF₃ is due not to kinetics but to thermodynamic origins^[21].

Revised text (Page 9)

To evaluate the effect of maintaining the structural integrity during intercalation and conversion reaction for T-FeF₃ on voltage hysteresis and compositional inhomogeneity, we first compared the voltage hysteresis in T-FeF₃ and R-FeF₃ with galvanostatic intermittent titration technique (GITT) analysis (Fig. 3a). All the analyses were performed after 10 cycles to ensure the evolution of T-FeF₃ from LiF-FeF₂. Fig. 3b shows the difference in relaxed voltage during charge and discharge corresponding to thermodynamic hysteresis, and Fig. 3c presents the extent of voltage change during relaxation corresponding to kinetic hysteresis. T-FeF₃ exhibited a smaller thermodynamic hysteresis compared to R-FeF₃, which is prominent at the end of charge or discharge. Both T-FeF₃ and R-FeF₃ exhibit larger voltage hysteresis during the charging process than during discharge. For T-FeF₃, this is mainly attributed to LiF splitting that occurs during charging (Supplementary Note 7), while in the case of R-FeF₃, the increased hysteresis likely results from phase transitions involving long-range diffusion. However, this difference in kinetic hysteresis between T-FeF₃ and R-FeF₃ is relatively insignificant during both charge and discharge states. This is due to the improved reaction rate and mass transfer in both cases using carbon composites with nano-sized particles. This finding implies that the alleviation of voltage hysteresis in T-FeF₃ compared to R-FeF₃ is due not to kinetics but to thermodynamic origins^[21] (Supplementary Fig. S12 and Supplementary Note 5).

Original supplementary Figure

Supplementary Fig. S14. Voltage profile and cycle performance of T-FeF₃ in the order of (a, c) WV-UV-WV range and (b, d) WV-LV-WV range.

Supplementary Note 5. Evaluation of the Reversibility of T-FeF₃

Supplementary Fig. S14 demonstrates the reversibility across different voltage ranges after the initial 10 cycles required to form T-FeF₃. At the 100th cycle, the wide voltage (WV, Supplementary Fig. S1c) and upper voltage ranges (UV) show excellent capacity retention of 99 % and 93 %, respectively, while the lower voltage range (LV) exhibits relatively poor capacity retention of 69 %. The excellent reversibility in the UV and the rapid capacity decay in the LV suggest that the reaction mechanism of T-FeF₃ proceeds through Li⁺ insertion and conversion reaction.

During the charging process, the phase fraction changes gradually through points 4 and 5, unlike the stepwise changes observed during discharge (Fig. 2c). The formation of T-FeF₃ from LiF and FeF₂ requires the splitting of LiF:

LiF splitting requires a high-voltage environment^{8–12}. Insufficient LiF splitting in the LV may lead to incomplete formation of T-FeF₃ and accumulation of inactive LiF, resulting in capacity decay. To verify insufficient LiF splitting in the LV, reversibility was checked in the UV and LV, followed by a cycle in the WV (Supplementary Fig. S14). The electrochemical profile and capacity in UV remained almost identical to the 10th cycle when re-measured in the WV. In contrast, the LV showed the absence of the 4 V plateau and lower capacity compared to the 10th cycle. However, with continued cycling in the WV range, the evolution of the 4 V plateau and an increase in capacity were observed. The absence of the 4 V plateau and lower capacity in the LV suggest failure to form T-FeF₃ due to insufficient LiF splitting,

and the evolution of the 4 V plateau and capacity increase in the WV with additional cycling indicate the successful formation of T-FeF₃ under high-voltage operation. These characteristics of electrochemical behavior across various voltage ranges provide further evidence that T-FeF₃ undergoes both insertion and conversion reaction mechanisms.

Revised supplementary Figure

Supplementary Fig. S17. Voltage profile and cycle performance of T-FeF₃ in the order of (a, c) WV-UV-WV range and (b, d) WV-LV-WV range. Arrows have been added to indicate the progression of voltage profiles across different cycles.

Supplementary Note 7. Evaluation of the Reversibility of T-FeF₃

Supplementary Fig. S14 demonstrates the reversibility across different voltage ranges after the initial 10 cycles required to form T-FeF₃. At the 100th cycle, the wide voltage (WV, Supplementary Fig. S1c) and upper voltage ranges (UV) show excellent capacity retention of 99 % and 93 %, respectively, while the lower voltage range (LV) exhibits relatively poor capacity retention of 69 %. The excellent reversibility in the UV and the rapid capacity decay in the LV suggest that the reaction mechanism of T-FeF₃ proceeds through Li⁺ insertion and conversion reaction.

During the charging process, the phase fraction changes gradually through points 4 and 5, unlike the stepwise changes observed during discharge (Fig. 2c). The formation of T-FeF₃ from LiF and FeF₂ requires the splitting of LiF:

LiF splitting requires a high-voltage environment⁸⁻¹². Insufficient LiF splitting in the LV may lead to incomplete formation of T-FeF₃ and accumulation of inactive LiF, resulting in capacity decay. To verify insufficient LiF splitting in the LV, reversibility was checked in the UV and LV, followed by a cycle in the WV (Supplementary Fig. S17). The electrochemical profile and capacity in UV remained almost identical to the 10th cycle when re-measured in the WV. In contrast, for the 111th voltage profile in the LV region, a large overpotential was observed when the charged cut-off voltage was restored from 3.4 V to 4.8 V. This behavior, coupled with the absence of the 4 V plateau and reduced capacity, suggests a failure to form T-FeF₃ due to insufficient LiF splitting. However, as cycling continued in the WV range, the gradual evolution of the 4 V plateau and the concurrent increase in capacity clearly indicate successful T-FeF₃ formation under high-voltage operation. These characteristics of electrochemical behavior across various voltage ranges provide further evidence that T-FeF₃ undergoes both insertion and conversion reaction mechanisms.

Furthermore, this result highlights the necessity of a higher charge cut-off voltage for the effective formation of T-FeF₃. This is supported by a comparative analysis of electrochemical behavior at 4.8 V and 4.5 V. As shown in Supplementary Fig. S18a, the sample cycled at 4.8 V exhibited a well-defined plateau and significantly higher capacity, while the sample cycled at 4.5 V showed a much less pronounced 4 V feature, suggesting incomplete LiF splitting and limited activation of T-FeF₃. In contrast, R-FeF₃ exhibited little difference in capacity depending on the charge cut-off voltage, but consistently showed lower capacity retention compared to T-FeF₃, regardless of the voltage range (Supplementary Fig. S18b and c). This further emphasizes the structural irreversibility of R-FeF₃ and its inferior electrochemical reversibility. These findings collectively reinforce that 4.5 V is insufficient to fully promote the reconversion reaction, and that a 4.8 V cut-off is essential to separating LiF required for reversible T-FeF₃ formation.

Comments

4. *Could the authors give a clearer explanation of the statement "the alleviation of voltage hysteresis in T-FeF₃ compared to R-FeF₃ is due not to kinetics but to thermodynamic origins"? Won't the kinetic barriers (i.e., electron/ion diffusion) contribute to the compositional inhomogeneity of reaction products or the proposed "thermodynamic hysteresis" in this manuscript?*

Author reply:

We sincerely appreciate the reviewer's insightful comment regarding the relationship between kinetic barriers and the proposed thermodynamic origin of voltage hysteresis. To further verify our claim that the alleviation of voltage hysteresis in T-FeF₃ is primarily governed by thermodynamic factors rather than kinetic limitations, we conducted additional systematic experiments focusing on the crystallinity and interfacial contact of the LiF-FeF₂ nanocomposite precursor.

By reducing the ball-milling time, we intentionally increased the crystal size of the LiF-FeF₂ composite, thereby worsening interfacial contact and potentially increasing kinetic barriers. Specifically, this study prepared the LiF-FeF₂ nanocomposite through a two-step ball-milling process: LiF/FeF₂ mixing (500 rpm, 48 hours) followed by carbon mixing (500 rpm, 12 hours). We fixed the carbon mixing step and content to maintain consistent conductivity and varied only the first ball-milling duration (0 hours, 12 hours, and 48 hours) to modulate the crystallinity size intentionally.

Fig. R2. (a) XRD patterns of LiF-FeF₂ nanocomposites prepared with different mixing times. (b) Full width at half maximum (FWHM) of the (110)_{FeF₂} and (200)_{LiF} diffraction peaks as a function of mixing time. (c) Electrochemical charge-discharge profiles of the composites prepared with varying mixing times, along with their differential voltage (dQ/dV) analysis shown on the right.

As shown in Fig. R2a and b, decreasing the ball-milling time led to increased crystallinity, evidenced by the reduced full width at half maximum (FWHM) of the LiF and FeF₂ diffraction peaks. These variations led to pronounced differences in electrochemical performance. The electrochemical profile (Fig. R2c) shows that higher crystallinity (shorter milling time) resulted in lower discharge capacity, which is thought to be due to the reduced active interfacial contact between LiF and FeF₂.

Fig. R2. (d) GITT profiles at the 10th cycle. The cells were allowed to relax for 3 h after every 11.2 mAh g⁻¹ (corresponding to 0.05 e⁻ per formula unit) of discharge/charge at a current density of 20 mA g⁻¹. (e) Voltage changes after 3 h relaxation (ΔV_{rest}) measured at different lithiation states during discharge and charge for each mixing condition.

Despite this capacity loss due to reduced milling time, the GITT results showed that the rest potential change during charging (ΔV_{rest}), which often reflects kinetic polarization, exhibited only a slight variation (Fig. R2d and e). Specifically, ΔV_{rest} increased marginally at end of charge (region of LiF spitting) as milling time decreased. However, even under the shortest milling condition (0 h), its magnitude remained considerably smaller than that of R-FeF₃. This further demonstrates that while kinetic polarization may contribute to transient overpotential, it does not dominate the overall voltage hysteresis. These results suggest that the influence of electron/ion diffusion-related kinetic barriers is very small in the current system, as it consists of sufficiently small nanosized particles and an abundant conductive carbon composite.

Fig. R2. (c) Electrochemical charge-discharge profiles of the composites prepared with varying mixing times, along with their differential voltage (dQ/dV) analysis shown on the right.

The capacity variation observed across different milling times (Fig. R2c) suggests that the degree of compositional inhomogeneity, particularly the amount of electrochemically formed T-FeF₃, varies depending on the extent of interfacial contact between LiF and FeF₂. Given that kinetic polarization (ΔV_{rest}) remains minimal even under poor interfacial conditions, it is reasonable to attribute these compositional differences not to kinetic limitations such as electron or ion diffusion, but to thermodynamic factors that govern phase evolution and equilibrium.

Fig. R2. (f) Voltage gap ($\Delta V_{\text{relax}} = V_{\text{gap}} = V_{\text{relax,charge}} - V_{\text{relax,discharge}}$) measured after 3 h relaxation at equivalent lithiation states for each mixing condition.

This interpretation is further supported by the V_{relax} data in Fig. R2f. While the overall profiles and absolute values of V_{relax} remained largely consistent across different ball-milling times, clear variations were observed at specific states of charge (SOC). These differences are most likely attributable to variations in the amount of electrochemically formed T-FeF₃ from the LiF-FeF₂, rather than any alteration in the underlying reaction mechanism. In contrast, R-FeF₃, which undergoes a sluggish and partially irreversible phase transition involving long-range cation migration, consistently exhibited a significantly larger ΔV_{relax} than all T-FeF₃ samples. This result suggests that the intrinsic thermodynamic reaction pathway, rather than kinetics, governs the observed hysteresis behavior.

By slightly changing the crystallinity while maintaining the same carbon content and nanoscale particle size, we demonstrate that kinetic factors such as electron/ion transport have minimal influence on the voltage hysteresis in our nanocomposite system. This strongly supports our conclusion that the reduction in voltage hysteresis observed in T-FeF₃ compared to R-FeF₃, which involves irreversible phase transitions and structural instability, is not due to kinetic origin, but rather arises from the intrinsic thermodynamic nature of T-FeF₃, which exhibits high structural reversibility.

Based on the reviewer's comments, we added Supplementary Information (Supplementary Fig. R2 and Supplementary Note 5) and revised the manuscript as follows.

Original text (Page 7-8)

To evaluate the effect of maintaining the structural integrity during intercalation and conversion reaction for T-FeF₃ on voltage hysteresis and compositional inhomogeneity, we first compared the voltage hysteresis in T-FeF₃ and R-FeF₃ with galvanostatic intermittent titration technique (GITT) analysis (Fig. 3a). All the analyses were performed after 10 cycles to ensure the evolution of T-FeF₃ from LiF-FeF₂. Fig. 3b shows the difference in relaxed voltage during charge and discharge corresponding to thermodynamic hysteresis, and Fig.

3C presents the extent of voltage change during relaxation corresponding to kinetic hysteresis. T-FeF₃ exhibited a smaller thermodynamic hysteresis compared to R-FeF₃, which is prominent at the end of charge or discharge. In addition, the difference in kinetic hysteresis between T-FeF₃ and R-FeF₃ is relatively insignificant during both charge and discharge states. This is due to the improved reaction rate and mass transfer in both cases using carbon composites with nano-sized particles. This finding implies that the alleviation of voltage hysteresis in T-FeF₃ compared to R-FeF₃ is due not to kinetics but to thermodynamic origins^[21].

Revised text (Page 9)

To evaluate the effect of maintaining the structural integrity during intercalation and conversion reaction for T-FeF₃ on voltage hysteresis and compositional inhomogeneity, we first compared the voltage hysteresis in T-FeF₃ and R-FeF₃ with galvanostatic intermittent titration technique (GITT) analysis (Fig. 3a). All the analyses were performed after 10 cycles to ensure the evolution of T-FeF₃ from LiF-FeF₂. Fig. 3b shows the difference in relaxed voltage during charge and discharge corresponding to thermodynamic hysteresis, and Fig. 3c presents the extent of voltage change during relaxation corresponding to kinetic hysteresis. T-FeF₃ exhibited a smaller thermodynamic hysteresis compared to R-FeF₃, which is prominent at the end of charge or discharge. Both T-FeF₃ and R-FeF₃ exhibit larger voltage hysteresis during the charging process than during discharge. For T-FeF₃, this is mainly attributed to LiF splitting that occurs during charging (Supplementary Note 7), while in the case of R-FeF₃, the increased hysteresis likely results from phase transitions involving long-range diffusion. However, this difference in kinetic hysteresis between T-FeF₃ and R-FeF₃ is relatively insignificant during both charge and discharge states. This is due to the improved reaction rate and mass transfer in both cases using carbon composites with nano-sized particles. This finding implies that the alleviation of voltage hysteresis in T-FeF₃ compared to R-FeF₃ is due not to kinetics but to thermodynamic origins^[21] (Supplementary Note 5).

Added supplementary Figure

Supplementary Fig. S12. (a) XRD patterns of LiF-FeF₂ nanocomposites prepared with different mixing times. **(b)** Full width at half maximum (FWHM) of the (110)_{FeF₂} and (200)_{LiF} diffraction peaks as a function of mixing time. **(c)** Electrochemical charge-discharge profiles of the composites prepared with varying mixing times, along with their differential voltage (dQ/dV) analysis shown on the right. **(d)** GITT profiles at the 10th cycle. The cells were allowed to relax for 3 h after every 11.2 mAh g⁻¹ (corresponding to 0.05 e⁻ per formula unit) of discharge/charge at a current density of 20 mA g⁻¹. **(e)** Voltage gap (V_{gap} = V_{relax,charge} - V_{relax,discharge}) measured after 3 h relaxation at equivalent lithiation states for each mixing condition. **(f)** Voltage changes after 3 h relaxation (ΔV_{rest}) measured at different lithiation states during discharge and charge for each mixing condition.

Supplementary Note 5. Role of LiF-FeF₂ Interfacial Integrity in Phase Evolution and Electrochemical Performance of T-FeF₃.

The interfacial contact between LiF and FeF₂ in the LiF-FeF₂ nanocomposite plays a critical role in facilitating the guided phase transition toward T-FeF₃ and determining the overall electrochemical performance. To systematically investigate the impact of this interfacial contact, we examined two key factors: the purity of the LiF precursor, which influences the chemical integrity of the composite, and the ball-milling time, which affects particle size and mixing uniformity.

The critical role of LiF-FeF₂ interfacial contact in forming T-FeF₃ was further supported by investigating the effect of LiF precursor purity. To evaluate this, we prepared LiF-FeF₂

composites using LiF with purities of 99.99%, 99.85%, and 97%. XRD analysis revealed no significant differences in crystal structure or phase composition among the samples, indicating that variations in purity did not cause noticeable changes in the bulk structure (Supplementary Fig. S11a). However, electrochemical measurements revealed that lowering the LiF purity led to a slight but consistent reduction in capacity. This capacity loss is likely due to the presence of impurities, which may disrupt the LiF-FeF₂ interface, reduce effective contact, or introduce electrochemically inactive phases that interfere with smooth T-FeF₃ formation (Supplementary Fig. S11b).

To further confirm the importance of interfacial contact, we investigated the effect of crystallinity by systematically varying the ball-milling time. By adjusting only the initial LiF-FeF₂ mixing duration (48 h, 12 h, and 0 h) while maintaining the carbon mixing step constant at 12 h (500 rpm), we were able to isolate the impact of interfacial contact and crystallinity on electrochemical behavior. XRD analysis revealed that reducing the milling time resulted in narrower diffraction peaks and a noticeable decrease in the full width at half maximum (FWHM), indicating increased crystallinity and larger particle sizes (Supplementary Fig. S12a and b). This microstructural change reduced the interfacial contact area between LiF and FeF₂, which is essential for promoting the phase transition to T-FeF₃. Electrochemical testing further confirmed that insufficient LiF-FeF₂ contact deteriorates capacity but has limited impact on voltage hysteresis behavior. As milling time decreased, charge–discharge profiles (Supplementary Fig. S11c) showed reduced capacity due to poorer interfacial contact. This capacity degradation is attributed to the limited formation of electrochemically active T-FeF₃, resulting in compositional inhomogeneity. However, GITT (Galvanostatic Intermittent Titration Technique) analysis revealed that the difference in relaxed voltages between charge and discharge (V_{gap}), representing thermodynamic hysteresis, remained nearly constant across all milling times and significantly smaller than that of R-FeF₃ (Supplementary Fig. S12d–e). The rest potential change (ΔV_{rest}), which reflects kinetic polarization, slightly increased at the end of charge as milling time decreased. Nevertheless, even in the shortest milling condition (0 h), ΔV_{rest} was still considerably smaller than in R-FeF₃ (Supplementary Fig. S12f), indicating that kinetic effects such as electron/ion diffusion barriers were minimal. This is likely because our nanocomposite system consists of sufficiently nanosized particles and a highly conductive carbon matrix, minimizing the impact of transport-related kinetic limitations.

In summary, the interfacial contact between LiF and FeF₂ plays a crucial role in the formation and electrochemical activation of T-FeF₃. In our nanocomposite system consisting of sufficiently small particles and an abundant conductive carbon matrix, changes in interfacial contact significantly affect capacity but have minimal impact on voltage hysteresis. Notably, T-FeF₃ exhibits consistently low hysteresis even under conditions of degraded interfacial contact, reinforcing the conclusion that the hysteresis difference between T-FeF₃ and R-FeF₃ arises primarily from their inherent thermodynamic reaction pathways, rather than from kinetic limitations such as electron or ion transport resistance.

Comments

5. In Figure 5a-c, why does R-FeF₃ suffer a more obvious overcharge problem with a lower coulombic efficiency than T-FeF₃?

Author reply:

We appreciate the reviewer's meticulous comments. We attribute the overcharging behavior of R-FeF₃ to the limited reconversion reaction caused by compositional inhomogeneity.

To clarify our explanation, we classify the following cycling regions in Fig. 5a. Region 1 corresponds to cycles 1–10, operating within the 1-electron transfer voltage range (4.8–2V). Region 2 includes cycles 11–20, where the system undergoes deep discharge (4.8–1.5V). Finally, region 3 consists of cycles 21–40, returning to the 1-electron transfer voltage range (4.8–2V).

Fig. 5a-c. The cycle ability and differential analysis of voltage profile of T-FeF₃ and R-FeF₃ with repeated changing discharge cut-off voltage (2-1.5-2V).

In region 1, the coulombic efficiency (CE) of T-FeF₃ (91.4%) is slightly higher than that of R-FeF₃ (90.6%), which can be attributed to their distinct charging mechanisms. Specifically, during the charging process of T-FeF₃, LiF splitting occurs, requiring a higher voltage to overcome the stability of LiF. This overpotential suppresses overcharging, leading to an improved coulombic efficiency compared to R-FeF₃.

In region 2, the CE difference becomes more pronounced, with T-FeF₃ at 98.9% and R-FeF₃ at 93.98%. Similarly, in region 3, T-FeF₃ maintains a CE of 94.2%, while R-FeF₃ drops significantly to 85.2%, indicating greater overcharge losses in R-FeF₃. This behavior can be explained by Fig. 5d, which presents *ex-situ* XRD patterns of the charged state. In Fig. 5d, "R" and "T" represent R-FeF₃ and T-FeF₃, respectively, while the numbers correspond to the cycle points in Fig. 5a.

Fig. 5d. The XRD pattern of charged states for T-FeF₃(T) and R-FeF₃(R) at each point in the cycle measured at the changing cut-off voltage. Point 1 (10th cycle), point 2 (20th cycle), and point 3 (40th cycle).

The XRD patterns of R2 and R3 exhibit broader and more complex diffraction peaks compared to T2 and T3, suggesting that R-FeF₃ exhibits greater compositional inhomogeneity. At 1.5V deep discharge, both R-FeF₃ and T-FeF₃ undergo conversion reactions to Fe and LiF, requiring a subsequent reconversion process during charging (Supplementary Fig. S28).

Supplementary Fig. S28. Ex-situ XRD patterns of LiF-FeF₂ and R-FeF₃ electrodes at

charged/discharged states in the 4.8-1.5 V range.

Unfortunately, in the case of R-FeF₃, its greater compositional inhomogeneity compared to T-FeF₃ leads to poor contact formation, hindering efficient reconversion. As a result, R-FeF₃ exhibits compositionally inhomogeneous charged products due to insufficient LiF separation during the charging process. This reduces the LiF separation which induces overvoltage, leading to overcharging and ultimately resulting in a more significant decrease in coulombic efficiency

Therefore, we conclude that the more pronounced overcharge issue and lower coulombic efficiency in R-FeF₃ are attributed to greater compositional inhomogeneity, which disrupts the smooth progression of the reconversion process and leads to increased charging inefficiencies.

Comments

6. *Most of the electrochemical studies in this manuscript limit the redox reactions to single-electron transfer for iron fluoride, which causes a low reversible capacity < 200 mAh g⁻¹. Can the guided tetragonal structure have a significant effect on the high-capacity FeF₃ relying on three-electron transfer reactions?*

Author reply:

Thanks for the reviewer's valuable comment. The induced tetragonal structure of T-FeF₃ is particularly effective in stabilizing the intermediate phase during the single electron transfer, so that the structural guiding effect of the tetragonal framework is limited as the reaction proceeds to the complete conversion to Fe and LiF, involving multiple electron transfers.

Supplementary Fig. S28. *Ex-situ* XRD patterns of LiF-FeF₂ and R-FeF₃ electrodes at charged/discharged states in the 4.8-1.5 V range.

In more detail, both T-FeF₃ and R-FeF₃ undergo conversion reactions, generating Fe and LiF phases when transferring more electrons beyond a single-electron reaction. This phenomenon has been clearly demonstrated by *ex-situ* XRD analysis (Supplementary Fig. S28).

Fig. R3. (a) Cycling performance of LiF-FeF₂ (purple) and FeF₃ (yellow) evaluated in the voltage range of 4.8–1.5 V is compared with the cycling performance of LiF-FeF₂ (blue) evaluated in the voltage range of 4.8–1.5 V after electrochemically forming T-FeF₃. Electrochemical formation of T-FeF₃ was performed for 10 cycles at 4.8–2.0 V. (b) Cycling performance comparison of the three cases in the same voltage range of 4.8–1.5 V

The full conversion to Fe and LiF necessitates extensive long-range diffusion, leading to compositional inhomogeneity, disruption of LiF-FeF₂ contacts, and consequently progressive capacity degradation. Our experimental results support this observation, showing that both T-FeF₃ and R-FeF₃ exhibit similar capacity retention when cycled at a cutoff voltage of 1.5 V (~2-electron transfer), as illustrated in Fig. R3.

Fig. 5e. The cycle ability of T-FeF₃ with repeated changing discharge cut-off voltage (2-1.2-2V).

Furthermore, cycling at deeper lithiation (1.2 V, ~3-electron transfer) induces greater irreversible capacity loss, as shown in Fig. 5e, which indicates a reduced guiding effect due to deeper conversions to metallic Fe.

Supplementary Fig. S33. Comparison of specific capacities of T-FeF₃ and R-FeF₃ with repeated changes in depth of discharge.

Nevertheless, as demonstrated by Supplementary Fig. S33, T-FeF₃ exhibits the unique ability to achieve excellent capacity retention upon effective reactivation, even after extensive lithiation cycles. This implies that with the development and implementation of effective

reactivation strategies, T-FeF₃ holds significant potential to outperform R-FeF₃ in terms of capacity retention.

Based on the reviewer's comments, the text has been revised and a figure has been added as follows.

Original text (Page 10)

The superior reversibility of T-FeF₃ is maintained even at deep discharge of the conversion reaction to LiF and Fe metal. Fig. 5a shows the capacity retention when the low cut-off voltage range is continuously varied back and forth from 2 to 1.5 V. The initial 10 cycles were preceded within a 4.8–2.0 V voltage range (point 1, 10th cycle) to form T-FeF₃. Then, the voltage range was changed to 4.8–1.5 V for 10 cycles to induce drastic structural evolution, forming the Fe metal phase (point 2, 20th cycle). Subsequently, the voltage range was recovered to 4.8–2.0 V for another 20 cycles (point 3, 40th cycle) to validate the reversibility. The characteristic 4 V redox feature of T-FeF₃ was absent in the differential curve, and the electrochemical profile is quite analogous to R-FeF₃ at point 2 (Fig. 5b). During deep discharge, Fe metal conversion occurs^[19,21,31,56,57] (Supplementary Fig. S26-28 and Supplementary Note 8) which involves long-range diffusion and could exacerbate the compositional inhomogeneity^[19,21]. Despite these harsh conditions, interestingly, the feature of T-FeF₃ is reversibly observed with the recovery of the 4 V redox at point 3, with an electrochemical profile analogous to that at point 1 displayed. This feature is repeatedly observed during cycling with swinging of the cut-off voltage (Supplementary Fig. S29). In contrast, R-FeF₃ did not exhibit the 4 V redox feature at any points (Fig. 5c and Supplementary Fig. S30). This finding indicates that T-FeF₃ can be reversibly recovered even after undergoing a conversion reaction involving severe structural evolution. Furthermore, note that the lithiated state after deep discharge appears to be same as LiF and Fe metal for both T-FeF₃ and R-FeF₃ given the similarity of the electrochemical profile at point 2; however, it could in fact be different.

Revised text (Page 11)

The superior reversibility of T-FeF₃ is maintained even at deep discharge of the conversion reaction to LiF and Fe metal. Fig. 5a shows the capacity retention when the low cut-off voltage range is continuously varied back and forth from 2 to 1.5 V. The initial 10 cycles were preceded within a 4.8–2.0 V voltage range (point 1, 10th cycle) to form T-FeF₃. Then, the voltage range was changed to 4.8–1.5 V for 10 cycles to induce drastic structural evolution, forming the Fe metal phase (point 2, 20th cycle). Subsequently, the voltage range was recovered to 4.8–2.0 V for another 20 cycles (point 3, 40th cycle) to validate the reversibility. The characteristic 4 V redox feature of T-FeF₃ was absent in the differential curve, and the electrochemical profile is quite analogous to R-FeF₃ at point 2 (Fig. 5b). During deep discharge, Fe metal conversion occurs^[19,21,31,56,57] (Supplementary Fig. S30-32 and Supplementary Note 10) which involves long-range diffusion and could exacerbate the

compositional inhomogeneity^[19,21]. As shown in Supplementary Fig. S33, when cycling under deep discharge conditions involving the conversion reaction to LiF and Fe, both R-FeF₃ and T-FeF₃ commonly experience capacity degradation. Interestingly, despite these harsh conditions (deep discharge), T-FeF₃ exhibits a reversible recovery of its characteristic 4 V redox process at Point 3. Consequently, the electrochemical profile of Point 3 closely resembles that observed at Point 1. This feature is repeatedly observed during cycling with periodically altering cut-off voltage (Supplementary Fig. S34). In contrast, R-FeF₃ did not exhibit the 4 V redox feature at any point (Fig. 5c and Supplementary Fig. S35). This finding indicates that T-FeF₃ can be reversibly recovered even after undergoing a conversion reaction involving severe structural evolution. Furthermore, note that the lithiated state after deep discharge appears to be same as LiF and Fe metal for both T-FeF₃ and R-FeF₃ given the similarity of the electrochemical profile at point 2; however, it could in fact be different.

Added supplementary Figure

Supplementary Fig. S33. (a) Cycling performance of LiF-FeF₂ (purple) and FeF₃ (yellow) evaluated in the voltage range of 4.8–1.5 V is compared with the cycling performance of LiF-FeF₂ (blue) evaluated in the voltage range of 4.8–1.5 V after electrochemically forming T-FeF₃. Electrochemical formation of T-FeF₃ was performed for 10 cycles at 4.8–2.0 V. (b) Cycling performance comparison of the three cases in the same voltage range of 4.8–1.5 V.

Reviewer #2(Remarks to the Author):

Comments to the Author

In the present article, the authors prepared tetragonal FeF₃ (T-FeF₃) with the aim of reducing reaction inhomogeneity and stabilizing the structure. This material maintains structural stability during the reaction process and exhibits lower voltage hysteresis. The authors explored the reaction mechanism through a series of characterizations and DFT calculations, and the investigation of the reaction mechanism is really impressive. However, there are several questions that need to be addressed before the paper can be published.

Author reply:

We are grateful for your valuable and constructive comments. In response, we conducted additional experiments and thoroughly revised the manuscript to address the points you raised, thereby enhancing both its clarity and depth. We genuinely hope that the revised manuscript is now ready to be published in *Nature Communications*.

Comments

1. For example, in the battery cycling tests, the authors selected a voltage range of 4.8–2.0 V. Generally speaking, voltage tests above 4.5 V are not particularly common in lithium-ion batteries. The authors are kindly asked to explain why the upper limit of the voltage range in this study was chosen as 4.8 V. Furthermore, since the electrolyte used in the experiments is a common standard electrolyte, could the electrolyte introduce any other issues during the testing process?

Author reply:

We sincerely appreciate the reviewer's thoughtful and constructive comments regarding the choice of the charge voltage limit and the potential influence of the standard electrolyte. To substantiate our selection of the 4.8 V cut-off and evaluate the role of electrolyte, we performed additional electrochemical tests under varied voltage conditions and in alternative electrolyte systems.

In this study, we selected a charge cut-off voltage of 4.8 V to promote effective LiF splitting, which is essential for electrochemically forming T-FeF₃ from the pristine LiF-FeF₂ nanocomposite. Given the high chemical stability of LiF, decomposition requires a sufficiently high voltage. Without this, LiF splitting is incomplete, which impedes the formation of T-FeF₃.

Fig. R4. (a, b) Electrochemical profiles of T-FeF₃ and R-FeF₃ at the 10th cycle measured at 50 mA g⁻¹ under different charge cut-off voltages (4.5 V and 4.8 V).

To validate the necessity of the 4.8 V cut-off, we conducted comparative electrochemical experiments at 4.5 V and 4.8 V using both T-FeF₃ and R-FeF₃ electrodes. As shown in Fig. R4a and b, the T-FeF₃ sample cycled at 4.8 V exhibited a clearly developed 4 V plateau, which is characteristic of the T-FeF₃ phase. In contrast, the 4 V plateau and capacity were significantly diminished when the cut-off voltage was limited to 4.5 V. Meanwhile, R-FeF₃ showed relatively minor dependence on cut-off voltage in terms of capacity.

Interestingly, as shown in Fig. R4c, even though LiF-FeF₂ cycled at 4.5 V initially formed only a small amount of T-FeF₃, it still maintained superior capacity retention compared to R-FeF₃, which exhibited significant capacity fading. This further supports the importance of T-FeF₃'s structural reversibility which is made possible through effective LiF splitting while also highlighting the irreversible behavior of R-FeF₃.

These results are consistent with prior literature emphasizing the need for high-voltage operation to enable LiF splitting (*Nat. Energy*, **2**, 16208 (2017); *Adv. Funct. Mater.* **31**, 2009133 (2021); *J. Power Sources*, **329**, 406 (2016); *J. Power Sources*, **354**, 34 (2017); *ECS Meet. Abstr.* **58**, 87 (2014)). We therefore conclude that using a 4.8 V cut-off is essential to fully utilize the LiF-FeF₂ composite and guide the system toward the reversible T-FeF₃ phase.

Fig. R4c. Cycling stability of T-FeF₃ and R-FeF₃ at 50 mA g⁻¹ under different charge cut-off voltages (4.5 V and 4.8 V).

Regarding the electrolyte, we agree with the reviewer's concern that using a standard electrolyte (1 M LiPF₆ in EC/DMC, 1:1 v/v) could potentially introduce other issues during testing. In particular, LiPF₆ decomposition at high voltages may generate F⁻ species, which could participate in electrode reactions. Indeed, a previous study (*Chem. Mater.* **30**, 5362–5372 (2018)) reported that electrolyte decomposition under high voltage can lead to F- adsorption on Mn-based cathodes and promote mixed-phase formation.

To determine whether the F source driving FeF₂ activation originated from the electrolyte or the LiF within the composite, we conducted a controlled experiment in a fluorine-free environment. Specifically, we used polyacrylonitrile (PAN) as the binder and 1 M LiClO₄ in EC/DMC as the electrolyte. As presented in Supplementary Fig. S6, even in the absence of any fluorine-containing electrolyte or binder, the gradual development of the characteristic 4 V plateau was still observed. This strongly indicates that T-FeF₃ formation is driven by the intrinsic LiF in the composite, not by fluorine released from the electrolyte. While we acknowledge that minor contributions from electrolyte decomposition at high voltage cannot be completely ruled out, these results confirm that LiF is the dominant F source responsible for T-FeF₃ formation in our system.

Supplementary Fig. S6. Electrochemical profile of LiF-FeF₂ nanocomposite without additional fluorine sources other than LiF.

Based on the reviewer's valuable feedback, we have revised the manuscript to clarify the rationale behind the voltage selection and included additional electrochemical data and comparative figures to strengthen our conclusions. We hope this sufficiently addresses the reviewer's concerns.

Original Supplementary Note (Page 16)

Supplementary Note 5. Evaluation of the Reversibility of T-FeF₃

Supplementary Fig. S14 demonstrates the reversibility across different voltage ranges after the initial 10 cycles required to form T-FeF₃. At the 100th cycle, the wide voltage (WV, Supplementary Fig. S1c) and upper voltage ranges (UV) show excellent capacity retention of 99 % and 93 %, respectively, while the lower voltage range (LV) exhibits relatively poor capacity retention of 69 %. The excellent reversibility in the UV and the rapid capacity decay in the LV suggest that the reaction mechanism of T-FeF₃ proceeds through Li⁺ insertion and conversion reaction.

During the charging process, the phase fraction changes gradually through points 4 and 5, unlike the stepwise changes observed during discharge (Fig. 2c). The formation of T-FeF₃ from LiF and FeF₂ requires the splitting of LiF:

LiF splitting requires a high-voltage environment⁸⁻¹². Insufficient LiF splitting in the LV may lead to incomplete formation of T-FeF₃ and accumulation of inactive LiF, resulting in capacity decay. To verify insufficient LiF splitting in the LV, reversibility was checked in the UV and LV, followed by a cycle in the WV (Supplementary Fig. S14). The electrochemical profile and capacity in UV remained almost identical to the 10th cycle when re-measured in the WV. In contrast, the LV showed the absence of the 4 V plateau and lower capacity compared to the 10th cycle. However, with continued cycling in the WV range, the evolution of the 4 V plateau and an increase in capacity were observed. The absence of the 4 V plateau and lower capacity in the LV suggest failure to form T-FeF₃ due to insufficient LiF splitting, and the evolution of the 4 V plateau and capacity increase in the WV with additional cycling indicate the successful formation of T-FeF₃ under high-voltage operation. These characteristics of electrochemical behavior across various voltage ranges provide further evidence that T-FeF₃ undergoes both insertion and conversion reaction mechanisms.

Revised Supplementary Note

Supplementary Note 7. Evaluation of the Reversibility of T-FeF₃

Supplementary Fig. S14 demonstrates the reversibility across different voltage ranges after the initial 10 cycles required to form T-FeF₃. At the 100th cycle, the wide voltage (WV,

Supplementary Fig. S1c) and upper voltage ranges (UV) show excellent capacity retention of 99 % and 93 %, respectively, while the lower voltage range (LV) exhibits relatively poor capacity retention of 69 %. The excellent reversibility in the UV and the rapid capacity decay in the LV suggest that the reaction mechanism of T-FeF₃ proceeds through Li⁺ insertion and conversion reaction.

During the charging process, the phase fraction changes gradually through points 4 and 5, unlike the stepwise changes observed during discharge (Fig. 2c). The formation of T-FeF₃ from LiF and FeF₂ requires the splitting of LiF:

LiF splitting requires a high-voltage environment⁸⁻¹². Insufficient LiF splitting in the LV may lead to incomplete formation of T-FeF₃ and accumulation of inactive LiF, resulting in capacity decay. To verify insufficient LiF splitting in the LV, reversibility was checked in the UV and LV, followed by a cycle in the WV (Supplementary Fig. S17). The electrochemical profile and capacity in UV remained almost identical to the 10th cycle when re-measured in the WV. In contrast, for the 111th voltage profile in the LV region, a large overpotential was observed when the charged cut-off voltage was restored from 3.4 V to 4.8 V. This behavior, coupled with the absence of the 4 V plateau and reduced capacity, suggests a failure to form T-FeF₃ due to insufficient LiF splitting. However, as cycling continued in the WV range, the gradual evolution of the 4 V plateau and the concurrent increase in capacity clearly indicate successful T-FeF₃ formation under high-voltage operation. These characteristics of electrochemical behavior across various voltage ranges provide further evidence that T-FeF₃ undergoes both insertion and conversion reaction mechanisms.

Furthermore, this result highlights the necessity of a higher charge cut-off voltage for the effective formation of T-FeF₃. This is supported by a comparative analysis of electrochemical behavior at 4.8 V and 4.5 V. As shown in Supplementary Fig. S18a, the sample cycled at 4.8 V exhibited a well-defined plateau and significantly higher capacity, while the sample cycled at 4.5 V showed a much less pronounced 4 V feature, suggesting incomplete LiF splitting and limited activation of T-FeF₃. In contrast, R-FeF₃ exhibited little difference in capacity depending on the charge cut-off voltage, but consistently showed lower capacity retention compared to T-FeF₃, regardless of the voltage range (Supplementary Fig. S18b and c). This further emphasizes the structural irreversibility of R-FeF₃ and its inferior electrochemical reversibility. These findings collectively reinforce that 4.5 V is insufficient to fully promote the reconversion reaction, and that a 4.8 V cut-off is essential to separating LiF required for reversible T-FeF₃ formation.

Added supplementary Figure

Comments

2. Please add the corresponding electrochemical performance of this work to Supplementary Table S5 for the convenience of comparison by the readers.

Author reply:

We are appreciated for the reviewer's meticulous comments. Based on the reviewer's comments, the supplementary Table S5 was modified as follows.

Original supplementary information

Supplementary Table S1. Comparison of previously reported iron fluoride materials composited with carbon.

N o.	re f.	sample	Theoreti cal	Initial discharge	Cycli ng number	Curre nt (mA g ⁻¹)	Post-cycle discharge	capac ity retentio n	Volta ge range
------	-------	--------	--------------	-------------------	-----------------	--------------------------------	----------------------	----------------------	----------------

		Capacit y (mAh g ⁻¹)	Capac ity (mAh g ⁻¹)			capac ity (mAh g ⁻¹)			
				650		360	55.40 %		
1	(1 0)	FeF ₃ /C (70:25)	712 (3Li)	620	30	71.2	200	32.30 %	1.0- 4.5V
				650			200	30.80 %	
				645			360	55.80 %	
2	(1 1)	FeF ₃ FeF ₃ /AB(85 :15)	712 (3Li)	210.51 346.25	40	50	54.42 161.5 8	25.90 46.70 %	1.0- 4.0V
3	(1 2)	FeF ₃ -H- rGO FeF ₃	712 (3Li)	690 520	110	100	200 90	29.00 17.30 %	1.0- 4.5V
4	(1 3)	FeF ₃ /AB (70:25)	712 (3Li)	510	15	71.2	180	35.30 %	1.0- 4.5V
5	(1 4)	FeF ₃ /C- SWNT FeF ₃	712 (3Li)	300 200	100	300	190 60	63.33 30.00 %	1.0- 4.5V 1.0- 4.0V
6	(1 5)	FeF ₃ /C	475 (2Li)	170	53	20	89	52.40 %	1.5- 4.5V
7	(1 6)	bare FeF ₃	475 (2Li)	598	170	100	80	13.40 %	1.5- 4.5V
8	(1 7)	FeF ₃ /r- GO1.7 FeF ₃	237 (1Li) 475 (2Li) 237 (1Li) 475 (2Li)	195 414 114 314	50	23.7 71.2 23.7 71.2	170 274 50 35	87.18 66.18 43.86 11.15 %	2.0- 4.5 V 1.5- 4.5 V 2.0- 4.5 V 1.5- 4.5 V
9	(1 8)	FeF ₃	712 (3Li)	710	12	14.3	140	19.72 %	1.5- 4.5 V
10	(1 9)	FeF ₃ /AB (85:15)	237 (1Li)	129.3 119 105.1	100	474	237 101.9 83.6 71.2	78.80 70.30 67.70 %	2.0V- 4.5V
11	(2 0)	FeF ₃ /AB (70:25) FeF ₃ /AB- HT	237 (1Li)	210 200	25 50	10 10	100 168	47.60 84.00 %	2.0V- 4.5V
12	(2 1)	FeF ₃ /C composite	237 (1Li)	170	100	250	127	74.71 %	2.0- 4.5V
		FeF ₃		40			30	75.00 %	
13	(2 2)	FCO (FeF ₃ @carbon nanocomposite) FCN (FeF ₃ @N- doped carbon	237 (1Li)	120 140	200	474	45 100	37.50 71.43 %	2.0- 4.5V

Revised supplementary information

Supplementary Table S2. Comparison of previously reported iron fluoride materials composited with carbon.

No.	ref.	Sample	Theoretical	Initial	Cycling	Current	Post-	capacit	Volta	
			Capacity	Capacity			cycle			ity
			(mAh g ⁻¹)	(mAh g ⁻¹)	number	(mA g ⁻¹)	discharg	retention	range	
							e			
							capaci			
			(mAh g ⁻¹)	(mAh g ⁻¹)			ty (mAh			
							g ⁻¹)			
This work		T-FeF ₃	224 (1Li)	205.11 178.49	114 300	20 50	182.68 128.85	89.06 72.19 67.08 50.92	2.0- 4.8V	
		R-FeF ₃	237 (1Li)	185.19 164.74	114 300	20 50	124.23 83.89			
				650			360	55.40		
				620			200	32.30		
1	(20)	FeF ₃ /C (70:25)	712 (3Li)	650	30	71.2	200	30.80	1.0- 4.5V	
				645			360	55.80		
2	(21)	FeF ₃ FeF ₃ /AB(85:15)	712 (3Li)	210.51 346.25	40	50	54.42 161.58	25.90 46.70	1.0- 4.0V	
3	(22)	FeF ₃ -H-rGO FeF ₃	712 (3Li)	690 520	110	100	200 90	29.00 17.30	1.0- 4.5V	
4	(23)	FeF ₃ /AB (70:25)	712 (3Li)	510	15	71.2	180	35.30	1.0- 4.5V	
5	(24)	FeF ₃ /C- SWNT FeF ₃	712 (3Li)	300 200	100	300	190 60	63.33 30.00	1.0- 4.5V 4.0V	
6	(25)	FeF ₃ /C	475 (2Li)	170	53	20	89	52.40	1.5- 4.5V	
7	(26)	bare FeF ₃	475 (2Li)	598	170	100	80	13.40	1.5- 4.5V	
8	(27)	FeF ₃ /r-GO1.7 FeF ₃	237 (1Li)	195	50	23.7	170	87.18	2.0- 4.5 V	
			475 (2Li)	414			71.2	66.18	1.5- 4.5 V	
			237 (1Li)	114			23.7	50	43.86	2.0- 4.5 V
			475 (2Li)	314			71.2	35	11.15	1.5- 4.5 V
9	(28)	FeF ₃	712 (3Li)	710	12	14.3	140	19.72	1.5- 4.5 V	
10	(29)	FeF ₃ /AB (85:15)		129.3	100	474	101.9	78.80	2.0V- 4.5V	
			237 (1Li)	119			83.6	70.30		
				105.1			118.5	67.70		

11	(30)	FeF ₃ /AB (70:25)	237 (1Li)	210	25	10	100	47.60 %	2.0V-4.5V
		FeF ₃ /AB-HT		200	50	10	168	84.00 %	
12	(31)	FeF ₃ /C composite	237 (1Li)	170	100	250	127	74.71 %	2.0-4.5V
		FeF ₃		40			30	75.00 %	
13	(32)	FCO (FeF ₃ @carbon nanocomposite)	237 (1Li)	120	200	474	45	37.50 %	2.0-4.5V
		FCN (FeF ₃ @N-doped carbon nanocomposite)		140			100	71.43 %	

Comments

3. The authors mentioned in the introduction the advantages of FeF₃ as a typical conversion-type cathode material, such as high energy density and low cost. The theoretical capacity of FeF₃ as a cathode material for lithium-ion batteries is 712 mAh/g. However, the actual discharge capacity of the material prepared by the authors under low current density testing (e.g., 20 mA/g in the rate capability test) is much lower than the theoretical capacity of FeF₃. Please explain the reason for this discrepancy.

Author reply:

We would like to thank the reviewer for the careful review of our manuscript. In conclusion, the discharge capacity reported in our study is lower than the theoretical capacity of FeF₃ (712 mAh g⁻¹) because our work intentionally focuses on the single-electron reaction (FeF₃ + Li⁺ + e⁻ → LiFeF₃), rather than the full three-electron conversion reaction that leads to metallic Fe and LiF formation. This strategic focus is aimed at improving structural reversibility and mitigating voltage hysteresis, which are critical challenges in FeF₃ cathode materials.

Supplementary Fig. S30a. The 10th charge/discharge profile and differential analysis of voltage profile for LiF-FeF₂ in various voltage ranges. Blue and pink are voltage profiles in the 4.8-2.0 V and 4.8-1.5 V voltage ranges, respectively.

As shown in Supplementary Fig. S30a, at 20 mA g⁻¹, the measured discharge capacities were 189.5 mAh g⁻¹ at a 2.0 V cut-off and 436.2 mAh g⁻¹ at a 1.5 V cut-off. These values correspond to the insertion of approximately 0.85 Li⁺ and 0.97 Li⁺, respectively, relative to the theoretical capacity of the LiF-FeF₂ system. Consistently, these capacities are comparable to those of R-FeF₃ under the same conditions, as shown in Fig. 1c and Fig. R3a.

Fig. 1c. Electrochemical profile of LiF-FeF₂ nanocomposite at 25 °C and 20 mA g⁻¹ current density. The right is the differential analysis of the voltage profile.

Fig. R3a. Cycling performance of LiF-FeF₂ (purple) and FeF₃ (yellow) evaluated in the voltage range of 4.8-1.5 V is compared with the cycling performance of LiF-FeF₂ (blue) evaluated in the voltage range of 4.8-1.5 V after electrochemically forming T-FeF₃. Electrochemical formation of T-FeF₃ was performed for 10 cycles at 4.8-2.0 V.

Our main focus on the single-electron reaction is driven by the need to overcome the intrinsic limitations of R-FeF₃, which suffers from poor capacity retention and large voltage hysteresis even within the insertion regime. These issues are primarily caused by irreversible phase transitions, structural collapse, and compositional inhomogeneity during cycling (*Nat. Mater.* **20**, 841 (2021); *J. Am. Chem. Soc.* **138**, 2838 (2016)).

By designing T-FeF₃, our strategy enables the reversible utilization of FeF₂, which is typically an irreversible by-product in R-FeF₃, while enhancing structural stability and ensuring a stable reaction pathway. As a result, T-FeF₃ shows significantly improved capacity retention and reduced voltage hysteresis compared to R-FeF₃ (Fig. 3a-b and 4a).

Fig. 3. a, Galvanostatic intermittent titration technique (GITT) profiles of T-FeF₃ and R-FeF₃. The cells were allowed to relax for 3 h after every 11.2 mAh g⁻¹ (corresponding to 0.05 e⁻/formula unit) discharging/charging at 20 mA g⁻¹. **b**, Voltage difference ($V_{\text{gap}} = V_{\text{relax, charge}} - V_{\text{relax, discharge}}$) between charge and discharge steps after the 3 h relaxation at the same state of lithiation of T-FeF₃ and R-FeF₃.

Fig. 4a. Cycle stability at 50 mA g⁻¹ of T-FeF₃ and FeF₃.

Moreover, T-FeF₃ demonstrates excellent structural reversibility, even after deep conversion to Fe/LiF. This is confirmed by deep discharge experiments (Fig. 5). Even after extreme cycling conditions, T-FeF₃ restores its original structure, unlike R-FeF₃. Ex-situ XRD analysis also shows that after deep discharge, T-FeF₃ maintains a reversible pattern when cycled within 4.8V–2.0V, while R-FeF₃ exhibits broad and complex diffraction peaks, indicating severe structural degradation (Fig. 5d). These results emphasize the superior structural reversibility of T-FeF₃ compared to R-FeF₃.

In summary, the lower capacity observed in our study compared to the theoretical 712 mAh g⁻¹ is due to our intentional design focusing on the one-electron reaction, which enables improved structural reversibility and reduced voltage hysteresis, rather than pursuing the full three-electron conversion reaction.

Fig. 5. a-c, The cycle ability and differential analysis of voltage profile of T-FeF₃ and R-FeF₃ with repeated changing discharge cut-off voltage (2-1.5-2V). **d,** The XRD pattern of charged states for T-FeF₃(T) and R-FeF₃(R) at each point in the cycle measured at the changing cut-off voltage. Point 1 (10th cycle), point 2 (20th cycle), and point 3 (40th cycle).

Comments

4. *In Supplementary Table S5, different studies have used different voltage ranges for the cycling tests. For FeF₃, varying voltage ranges imply significant differences in the dominant reaction mechanism during the electrochemical process (1Li-3Li). One of the effects of this difference is the variation in cycling stability. Given the significant differences in testing conditions, is it rigorous to compare the performance under these conditions?*

Author reply:

Thank you for your comments on the comparison of the cycle stability of iron fluoride cathode materials. Our comparison of cycle stability may be confusing as the reviewer feared. Therefore, it seems important to compare the cycle stability under conditions where the same electrons are transferred. Based on the reviewer's comments, Figure 4e and Figure S25 have been revised as follows:

Original Figure

Fig. 4: Electrochemical performance of T-FeF₃. **a.** Cycle stability at 50 mA g⁻¹ of T-FeF₃ and FeF₃. **b,c,** Electrochemical profile of T-FeF₃ and R-FeF₃ at various cycles. **d,** Rate performance of LiF-FeF₂ and FeF₃. **e,** Comparison of capacity retention of T-FeF₃ and iron fluoride materials mixed with carbon.

Revised Figure

Fig. 4: Electrochemical performance of T-FeF₃. **a.** Cycle stability at 50 mA g⁻¹ of T-FeF₃ and FeF₃. **b,c,** Electrochemical profile of T-FeF₃ and R-FeF₃ at various cycles. **d,** Rate performance of LiF-FeF₂ and FeF₃. **e,** Comparison of capacity retention of T-FeF₃ and iron fluoride materials mixed with carbon. **The electrochemical stability of iron fluoride materials was evaluated in the 1 electron transfer range (Discharge cutoff voltage ~2V).**

Original supplementary Figure

Supplementary Fig. S25. Comparison of post-cycle discharge capacity of iron fluoride materials composited with carbon and T-FeF₃. (See Supplementary Table S5.)

Revised supplementary Figure

Supplementary Fig. S29. (a, b) Comparison of capacity retention and post-cycle discharge capacity for iron fluoride materials composited with carbon and FeF₃ (rhombohedral structure) and other Fe-based conversion-reaction materials (see Supplementary Table S5).

Comments

5. *What is the theoretical capacity of the metastable FeF₃ prepared by the authors within the voltage range used in the tests mentioned in the paper?*

Author reply:

We sincerely appreciate the reviewer's careful evaluation of our manuscript. The theoretical capacity of the metastable T-FeF₃ prepared in our study is 223.75 mAh g⁻¹. This corresponds to the theoretical capacity in the 4.8 V–2.0 V range, where a single-electron transfer occurs. This value is derived based on the fact that the initial precursor material used to

electrochemically induce T-FeF₃ is a LiF-FeF₂ nanocomposite. Given this composition (LiF-FeF₂), the molar mass of the active material is 119.78 g mol⁻¹, and one electron is involved in the redox reaction, leading to a theoretical capacity of 223.75 mAh g⁻¹.

The manuscript was revised with reference to the reviewer's comments.

Original text (Page 5)

First, the voltage profile of the LiF-FeF₂ nanocomposite was examined at room temperature within a voltage range of 4.8-2.0 V. Fig. 1c shows the charge-discharge profiles of the LiF-FeF₂ for the initial 10 cycles, compared with the 10th cycle profile of rhombohedral FeF₃ (s.g. *R-3c*, R-FeF₃), prepared by ball-milling with carbon to mitigate kinetic limitations (Supplementary Fig. S3). During cycling, the 4 V plateau in the LiF-FeF₂ nanocomposite, indicated by the red-shaded area, gradually evolved, with the average discharge voltage increasing from 3.02 V (1st cycle) to 3.15 V (10th cycle). In contrast, such electrochemical features were absent in R-FeF₃, as clearly seen in the dQ/dV analysis. Given the reversible redox reaction of iron involving fluorination which is confirmed by X-ray absorption spectroscopy (Supplementary Fig. S4a and S5), it is notable that the redox reaction around the 4 V in the LiF-FeF₂ nanocomposite represents the highest voltage for Fe²⁺/Fe³⁺ redox couple compared to LiFeSO₄F with triplite (3.9 V) and tavorite structure (3.6 V)^[45-47]. The origin of high redox potential will be discussed in later, but it implies structural evolution during electrochemical cycling, distinct from R-FeF₃.

Revised text (Page 5)

First, the voltage profile of the LiF-FeF₂ nanocomposite was examined at room temperature within a voltage range of 4.8-2.0 V. Fig. 1c shows the charge-discharge profiles of the LiF-FeF₂ for the initial 10 cycles, compared with the 10th cycle profile of rhombohedral FeF₃ (s.g. *R-3c*, R-FeF₃), prepared by ball-milling with carbon to mitigate kinetic limitations (Supplementary Fig. S3). During cycling, the 4 V plateau in the LiF-FeF₂ nanocomposite, indicated by the red-shaded area, gradually evolved, with the average discharge voltage increasing from 3.02 V (1st cycle) to 3.15 V (10th cycle). **By the 10th cycle, the discharge capacity is 189.5 mAh g⁻¹, corresponding to the insertion of 0.85 Li⁺ (theoretical capacity of 223.75 mAh g⁻¹ for single-electron transfer).** In contrast, such electrochemical features were absent in R-FeF₃, as clearly seen in the dQ/dV analysis. Given the reversible redox reaction of iron involving fluorination which is confirmed by X-ray absorption spectroscopy (Supplementary Fig. S4a and S5), it is notable that the redox reaction around the 4 V in the LiF-FeF₂ nanocomposite represents the highest voltage for Fe²⁺/Fe³⁺ redox couple compared to LiFeSO₄F with triplite (3.9 V) and tavorite structure (3.6 V)^[45-47]. The origin of high redox potential will be discussed in later, but it implies structural evolution during electrochemical cycling, distinct from R-FeF₃.

Comments

6. The GITT test can also be used to assess the diffusion level of lithium ions. Is there any difference in the lithium ions diffusion levels between the different materials in the paper?

Author reply:

We are really grateful to the reviewer for the valuable suggestions. After careful consideration, however, unfortunately, assessing lithium-ion diffusion levels through GITT analysis seems to be challenging. To accurately assess lithium-ion diffusion coefficients using the Galvanostatic Intermittent Titration Technique (GITT), certain key conditions is limited to the diffusion-limited reactions, such as layered (LCO, NMC) or spinel (LMO) materials, having large particle size (e.g. over 10 μm), where phase transitions are minimal (*J. Electrochem. Soc.* **168**, 120504 (2021); *J. Electrochem. Soc.* **168**, 120503 (2021)). In these materials, lithium-ion insertion/extraction occurs uniformly within the electrode and diffusion length is long enough to consider the diffusion as the primary rate-determining step instead of charge transfer reaction.

Therefore, GITT has limitations when applied to materials that undergo multi-phase transitions or include conversion reactions. For example, T- FeF_3 undergoes not only lithium insertion but also a conversion reaction to LiF and FeF_2 (Fig. 6a). In such cases, the observed voltage changes in GITT measurements are strongly influenced by thermodynamic fluctuations from phase transitions and interfacial reactions, rather than lithium-ion diffusion itself. As a result, the calculated diffusion coefficient may not accurately reflect the actual diffusion behavior of the material.

Fig. 6a. Reaction mechanism of T- FeF_3 derived from experimental and DFT calculation results.

Additionally, T- FeF_3 consists of nanoscale particles (Fig. R5, ~ 100 nm), which can lead to interfacial reactions at the electrode-electrolyte interface being the dominant rate-determining process. In nano-sized materials, the diffusion distance is short, meaning interfacial reactions rather than bulk diffusion may play a greater role, making GITT-based diffusion coefficient calculations less reliable (*J. Electrochem. Soc.* **168**, 120504 (2021); *J. Electrochem. Soc.* **168**,

120503 (2021)).

Fig. R5. Morphology of pristine LiF-FeF₂ nanocomposite.

In conclusion, although we measured GITT, it seems to be unsuitable to estimate lithium-ion diffusion coefficient because it may be difficult to obtain accurate values for materials in which multiple phase transformations occur or conversion reactions occur even in the nanoscale.

Comments

7. *If long-cycle tests with a voltage range similar to that in Supplementary Table S5 are conducted, the research results will be more convincing.*

Author reply:

Thank you for the reviewer's valuable suggestions. In line with the reviewer's recommendation, we conducted long-cycle tests within the two-electron reaction range (447.51 mAh g⁻¹, based on LiF-FeF₂) as shown in Fig. R3. When cycling within a 4.8–1.5 V voltage range, corresponding to the two-electron reaction, both R-FeF₃ and T-FeF₃ exhibited similar capacity retention. Furthermore, we also observed comparable capacity retention in two cases: (1) directly cycling LiF-FeF₂ within the 4.8–1.5 V range, and (2) cycling within the same voltage range after electrochemical formation of T-FeF₃. Analogous capacity degradation of all the cases seems to be attributed to the conversion reaction to LiF and Fe occurring beyond the single-electron reaction range (Supplementary Fig. S28). Because this conversion reaction inevitably involves long-range diffusion, compositional inhomogeneity arises, hindering effective contact between LiF and FeF₂. Therefore, these results indicate that T-FeF₃ also

showed capacity decay like R-FeF₃ when it undergoes the conversion reaction to LiF and Fe metal phases.

Fig. R3. (a) Cycling performance of LiF-FeF₂ (purple) and FeF₃ (yellow) evaluated in the voltage range of 4.8–1.5 V is compared with the cycling performance of LiF-FeF₂ (blue) evaluated in the voltage range of 4.8–1.5 V after electrochemically forming T-FeF₃. Electrochemical formation of T-FeF₃ was performed for 10 cycles at 4.8–2.0 V. (b) Cycling performance comparison of the three cases in the same voltage range of 4.8–1.5 V

Supplementary Fig. S28. *Ex-situ* XRD patterns of LiF-FeF₂ and R-FeF₃ electrodes at charged/discharged states in the 4.8-1.5 V range.

Although the T-FeF₃ shows analogous capacity retention to R-FeF₃ within wide voltage range (4.8-1.5 V), it can exhibit much improved capacity retention compared to R-FeF₃ when the voltage range is recovered to one electron transfer reaction (4.8-2.0 V) despite the history

of wide voltage range operation. As shown in Supplementary Fig. S32, even when cycling initially in the two-electron reaction range, subsequent cycling in the one-electron reaction range (4.8–2.0V) facilitates the reformation of the characteristic 4V plateau of T-FeF₃ (Supplementary Fig. S32d). This behavior is a key distinction between T-FeF₃ and R-FeF₃. As shown in Supplementary Fig. S33, R-FeF₃ experiences a faster capacity fade than T-FeF₃ even after subsequent cycling in the one-electron range where insertion dominates.

Supplementary Fig. S32. (a) Specific capacity of T-FeF₃ with repeated changed discharge cut-off voltage starting at 1.5 V discharge cut-off voltage. (b) Electrochemical profile and (d) differential analysis of voltage profile at 30th, 60th, and 90th (voltage range: 4.8–2.0 V). (c) The charge/discharge profile and (e) differential analysis of voltage profile at 10th, 40th, 70th, and 100th cycle (voltage range: 4.8–1.5 V).

Supplementary Fig. S33. Comparison of specific capacities of T-FeF₃ and R-FeF₃ with repeated changes in depth of discharge.

These results suggest that T-FeF₃ is a more reversible structure than R-FeF₃. Furthermore, if the tetragonal structure can be reactivated even after undergoing long-range diffusion-driven conversion reactions beyond the single-electron range in T-FeF₃, it indicates that it can achieve superior capacity retention compared to R-FeF₃.

Based on the author's comments, the manuscript has been revised as follows and a figure has been added.

Original text (Page 10)

The superior reversibility of T-FeF₃ is maintained even at deep discharge of the conversion reaction to LiF and Fe metal. Fig. 5a shows the capacity retention when the low cut-off voltage range is continuously varied back and forth from 2 to 1.5 V. The initial 10 cycles were preceded within a 4.8–2.0 V voltage range (point 1, 10th cycle) to form T-FeF₃. Then, the voltage range was changed to 4.8–1.5 V for 10 cycles to induce drastic structural evolution, forming the Fe metal phase (point 2, 20th cycle). Subsequently, the voltage range was recovered to 4.8–2.0 V for another 20 cycles (point 3, 40th cycle) to validate the reversibility. The characteristic 4 V redox feature of T-FeF₃ was absent in the differential curve, and the electrochemical profile is quite analogous to R-FeF₃ at point 2 (Fig. 5b). During deep discharge, Fe metal conversion occurs^[19,21,31,56,57] (Supplementary Fig. S26-28 and Supplementary Note 8 which involves long-range diffusion and could exacerbate the compositional inhomogeneity^[19,21]). Despite these harsh conditions, interestingly, the feature of T-FeF₃ is reversibly observed with the recovery of the 4 V redox at point 3, with an electrochemical profile analogous to that at point 1 displayed. This feature is repeatedly observed during cycling with swinging of the cut-off voltage (Supplementary Fig. S29). In contrast, R-FeF₃ did not exhibit the 4 V redox feature at any points (Fig. 5c and Supplementary Fig. S30). This finding indicates that T-FeF₃ can be reversibly recovered even after undergoing a conversion reaction involving severe structural evolution. Furthermore, note that the lithiated state after deep discharge appears to be same as LiF and Fe metal for both T-FeF₃ and R-FeF₃ given the similarity of the electrochemical profile at point 2; however, it could in fact be different.

Revised text (Page 11)

The superior reversibility of T-FeF₃ is maintained even at deep discharge of the conversion reaction to LiF and Fe metal. Fig. 5a shows the capacity retention when the low cut-off voltage range is continuously varied back and forth from 2 to 1.5 V. The initial 10 cycles were preceded within a 4.8–2.0 V voltage range (point 1, 10th cycle) to form T-FeF₃. Then, the voltage range was changed to 4.8–1.5 V for 10 cycles to induce drastic structural evolution, forming the Fe metal phase (point 2, 20th cycle). Subsequently, the voltage range was recovered to 4.8–2.0 V for another 20 cycles (point 3, 40th cycle) to validate the reversibility. The characteristic 4 V redox feature of T-FeF₃ was absent in the differential curve, and the electrochemical profile is quite analogous to R-FeF₃ at point 2 (Fig. 5b). During deep discharge, Fe metal conversion occurs^[19,21,31,56,57] (Supplementary Fig. S30-32 and Supplementary Note 10) which involves long-range diffusion and could exacerbate the compositional inhomogeneity^[19,21]. As shown in Supplementary Fig. S33, when cycling under deep discharge conditions involving the conversion reaction to LiF and Fe, both R-FeF₃ and T-FeF₃ commonly experience capacity degradation. Interestingly, despite these harsh conditions (deep discharge), T-FeF₃ exhibits a reversible recovery of its characteristic 4 V redox process at Point 3. Consequently, the electrochemical profile of Point 3 closely resembles that observed at Point 1. This feature is repeatedly observed during cycling with periodically altering cut-off voltage (Supplementary Fig. S34). In contrast, R-FeF₃ did not exhibit the 4 V redox feature at any point (Fig. 5c and Supplementary Fig. S35). This finding indicates that T-FeF₃ can be reversibly recovered even after undergoing a conversion reaction involving severe structural evolution. Furthermore, note that the lithiated state after deep discharge appears to be same as LiF and Fe metal for both T-FeF₃ and R-FeF₃ given the similarity of the electrochemical profile at point 2; however, it could in fact be different.

Added supplementary Figure

Supplementary Fig. S33. (a) Cycling performance of LiF-FeF₂ (purple) and FeF₃ (yellow)

evaluated in the voltage range of 4.8–1.5 V is compared with the cycling performance of LiF-FeF₂ (blue) evaluated in the voltage range of 4.8–1.5 V after electrochemically forming T-FeF₃. Electrochemical formation of T-FeF₃ was performed for 10 cycles at 4.8–2.0 V. (b) Cycling performance comparison of the three cases in the same voltage range of 4.8–1.5 V.

Comments

8. Why the cycle stability of T-FeF₃ is 72% after 300 cycles? It is related to the size of FeF₂?

Author reply:

Thank you for your insightful question regarding the cycling retention of T-FeF₃. We understand that the reviewer is inquiring about why the cycling retention of T-FeF₃ decreases to 72% after 300 cycles, despite its structural reversibility. The observed capacity fading in T-FeF₃ is primarily attributed to its conversion reaction mechanism. As shown in Fig. 6a, T-FeF₃ undergoes both Li⁺ insertion and conversion reactions, which contribute to its electrochemical behavior.

Fig. 6a. Reaction mechanism of T-FeF₃ derived from experimental and DFT calculation results.

In terms of crystal structure, T-FeF₃ exhibits a similar atomic arrangement to FeF₂, allowing for a relatively reversible conversion reaction to LiF and FeF₂. This structural similarity enhances its ability to maintain phase integrity after conversion, setting it apart from R-FeF₃, which undergoes irreversible phase transitions. However, over prolonged cycling, the reversibility of this conversion reaction gradually deteriorates, which can be attributed to several key factors:

LiF is chemically highly stable and electronically insulating, making its decomposition and reintegration into FeF₃ during reconversion challenging. If LiF does not maintain uniform contact with FeF₂, the reconversion process becomes inhomogeneous, reducing the overall reversibility. As cycling progresses, the interface between LiF and FeF₂ becomes increasingly inhomogeneous, leading to the formation of electrically disconnected inactive regions (dead

zones). This inhibits FeF_2 oxidation, resulting in continuous capacity loss.

This phenomenon can be observed in Fig. 4c, where the electrochemical profile of T-FeF_3 shows no new plateau or redox peak formation during long-term cycling. However, the characteristic 4 V plateau of T-FeF_3 gradually decreases, suggesting that the formation of T-FeF_3 gradually decreases during long-term cycling.

Fig. 4c. Electrochemical profile of T-FeF_3 and at various cycles.

Given these considerations, we believe that the conversion reaction plays a more dominant role in capacity fading than particle size effects. While structural characteristics of T-FeF_3 enable a relatively reversible conversion reaction to LiF and FeF_2 , long-term cycling introduces electronic insulation from LiF , structural stress accumulation, and potential dissolution effects, all of which contribute to the observed decline in capacity retention. Therefore, we are currently continuing follow-up additional research to improve the cycle stability of the FeF_3 cathode by overcoming those causes.

Comments

9. *If the time of ball-milling would affect the size distribution of FeF_2 and LiF and the final electrochemical performance?*

Author reply:

We thank the reviewer for the insightful question regarding the influence of ball-milling time on the particle size distribution of FeF_2 and LiF and its correlation with the resulting electrochemical performance. To address this, we conducted additional experiments to systematically investigate how the ball-milling condition affects the electrochemical behavior of the LiF-FeF_2 nanocomposite.

In our original preparation, the LiF-FeF₂ nanocomposite was synthesized by ball milling LiF and FeF₂ at 500 rpm for 48 hours, followed by an additional 500 rpm for 12 hours after adding carbon. To isolate the effect of the initial mixing stage, we varied only the initial ball-milling time for LiF and FeF₂ (0 h, 12 h, and 48 h), while keeping the carbon-mixing step constant (12 h at 500 rpm) to ensure uniform electronic conductivity across all samples.

Fig. R2. (a) XRD patterns of LiF-FeF₂ nanocomposites prepared with different mixing times. (b) Full width at half maximum (FWHM) of the (110)_{FeF₂} and (200)_{LiF} diffraction peaks as a function of mixing time.

XRD analysis (Fig. R2a) showed that decreasing the milling time resulted in sharper diffraction peaks. The full width at half maximum (FWHM) of the (110)_{FeF₂} and (200)_{LiF} peaks increased with shorter milling time, indicating larger crystallite sizes (Fig. R2b). This confirms that particle size distribution is indeed influenced by the ball-milling duration, with shorter milling resulting in larger FeF₂ and LiF domains.

Fig. R2. (c) Electrochemical charge-discharge profiles of the composites prepared with varying mixing times, along with their differential voltage (dQ/dV) analysis shown on the right.

Electrochemical charge-discharge profiles (Fig. R2c) demonstrated that samples with larger crystallites (shorter milling) exhibited significantly lower reversible capacities, despite the same carbon content. This result indicates that the reduction in ball-milling time leads to increased crystallinity of FeF₂ and LiF, which in turn reduces interfacial contact between the LiF and FeF₂ particles, thereby suppressing the electrochemical activation of T-FeF₃ and resulting in incomplete phase transformation.

Fig. R2. (d) GITT profiles at the 10th cycle. The cells were allowed to relax for 3 h after every 11.2 mAh g⁻¹ (corresponding to 0.05 e⁻ per formula unit) of discharge/charge at a current density of 20 mA g⁻¹. (e) Voltage gap ($\Delta V_{\text{relax}} = V_{\text{gap}} = V_{\text{relax,charge}} - V_{\text{relax,discharge}}$) measured after 3 h relaxation at equivalent lithiation states for each mixing condition.

However, while the ball-milling time clearly impacts the particle size distribution and resulting capacity, its effect on voltage hysteresis behavior was found to be minimal. Galvanostatic intermittent titration technique (GITT) results revealed that the voltage gap between charge and discharge (ΔV_{relax}), representing thermodynamic hysteresis, remained nearly unchanged across all ball-milling conditions and was significantly smaller than that of R-FeF₃ (Fig. R2d and e).

Fig. R2. (f) Voltage changes after 3 h relaxation (ΔV_{rest}) measured at different lithiation states during discharge and charge for each mixing condition.

At the rest potential change (ΔV_{rest}) which reflects kinetic polarization during relaxation, a slight increase in ΔV_{rest} was observed near the end of charge in samples with shorter milling times (Fig. R2f). However, even in the shortest milling condition (0 h), ΔV_{rest} remained much lower than that of R-FeF₃. This observation indicates that although the particle size increases with shorter milling, the presence of sufficient conductive carbon and overall nanoscale morphology mitigates the impact of kinetic limitations such as ion or electron diffusion.

From these findings, we conclude that the ball-milling time clearly affects the particle size distribution of FeF₂ and LiF, which in turn controls the interfacial contact and extent of T-FeF₃ formation, thus strongly influencing electrochemical capacity. However, since all samples remain within the nanoscale regime and are well-integrated with carbon, the influence of particle size on voltage hysteresis is limited.

To address the reviewer's comments, we have added supplementary notes and supplementary figures and revised the manuscript as follows:

Original text (Page 6)

Moreover, for the pair distribution function (PDF) analysis, the 10th charge state was more consistent with T-FeF₃ than with R-FeF₃ (Supplementary Fig. S9). Conclusively, Rietveld refinement of the XRD pattern of the electrode in the 10th charged state (Supplementary Fig. S10 and Supplementary Note 4) revealed an excellent match with the T-FeF₃ phase including residual LiF and FeF₂. These results confirm that the LiF-FeF₂ nanocomposite successfully leads to the gradual formation of T-FeF₃ while maintaining structural similarity to the mother structure (FeF₂). Notably, tetragonal FeF₃ derived from LiF-FeF₂ nanocomposites offers practical advantages in full-cell manufacturing^[43] compared to the previously reported tetragonal FeF₃ phase is formed *via* the delithiation of Li_{0.5}FeF₃, particularly due to the safety concerns and chemical instability associated with metallic lithium and lithium-containing anode electrodes^[4].

Revised text (Page 6)

Moreover, for the pair distribution function (PDF) analysis, the 10th charge state was more consistent with T-FeF₃ than with R-FeF₃ (Supplementary Fig. S9). Conclusively, Rietveld refinement of the XRD pattern of the electrode in the 10th charged state (Supplementary Fig. S10 and Supplementary Note 4) revealed an excellent match with the T-FeF₃ phase including residual LiF and FeF₂. These results confirm that the LiF-FeF₂ nanocomposite successfully leads to the gradual formation of T-FeF₃ while maintaining structural similarity to the mother structure (FeF₂). **The efficient formation of T-FeF₃ is closely governed by the interfacial contact between LiF and FeF₂, which facilitates the guided phase transition during cycling (Supplementary Note 5).** Notably, tetragonal FeF₃ derived from LiF-FeF₂ nanocomposites offers practical advantages in full-cell manufacturing^[43] compared to the previously reported tetragonal FeF₃ phase is formed *via* the delithiation of Li_{0.5}FeF₃, particularly due to the safety concerns and chemical instability associated with metallic lithium and lithium-containing anode electrodes^[4].

Added supplementary Figure

Supplementary Fig. S12. 2. (a) XRD patterns of LiF-FeF₂ nanocomposites prepared with different mixing times. **(b)** Full width at half maximum (FWHM) of the (110)_{FeF₂} and (200)_{LiF} diffraction peaks as a function of mixing time. **(c)** Electrochemical charge-discharge profiles

of the composites prepared with varying mixing times, along with their differential voltage (dQ/dV) analysis shown on the right. **(d)** GITT profiles at the 10th cycle. The cells were allowed to relax for 3 h after every 11.2 mAh g⁻¹ (corresponding to 0.05 e⁻ per formula unit) of discharge/charge at a current density of 20 mA g⁻¹. **(e)** Voltage gap ($V_{\text{gap}} = V_{\text{relax,charge}} - V_{\text{relax,discharge}}$) measured after 3 h relaxation at equivalent lithiation states for each mixing condition. **(f)** Voltage changes after 3 h relaxation (ΔV_{rest}) measured at different lithiation states during discharge and charge for each mixing condition.

Supplementary Note 5. Role of LiF-FeF₂ Interfacial Integrity in Phase Evolution and Electrochemical Performance of T-FeF₃.

The interfacial contact between LiF and FeF₂ in the LiF-FeF₂ nanocomposite plays a critical role in facilitating the guided phase transition toward T-FeF₃ and determining the overall electrochemical performance. To systematically investigate the impact of this interfacial contact, we examined two key factors: the purity of the LiF precursor, which influences the chemical integrity of the composite, and the ball-milling time, which affects particle size and mixing uniformity.

The critical role of LiF-FeF₂ interfacial contact in forming T-FeF₃ was further supported by investigating the effect of LiF precursor purity. To evaluate this, we prepared LiF-FeF₂ composites using LiF with purities of 99.99%, 99.85%, and 97%. XRD analysis revealed no significant differences in crystal structure or phase composition among the samples, indicating that variations in purity did not cause noticeable changes in the bulk structure (Supplementary Fig. S11a). However, electrochemical measurements revealed that lowering the LiF purity led to a slight but consistent reduction in capacity. This capacity loss is likely due to the presence of impurities, which may disrupt the LiF-FeF₂ interface, reduce effective contact, or introduce electrochemically inactive phases that interfere with smooth T-FeF₃ formation (Supplementary Fig. S11b).

To further confirm the importance of interfacial contact, we investigated the effect of crystallinity by systematically varying the ball-milling time. By adjusting only the initial LiF-FeF₂ mixing duration (48 h, 12 h, and 0 h) while maintaining the carbon mixing step constant at 12 h (500 rpm), we were able to isolate the impact of interfacial contact and crystallinity on electrochemical behavior. XRD analysis revealed that reducing the milling time resulted in narrower diffraction peaks and a noticeable decrease in the full width at half maximum (FWHM), indicating increased crystallinity and larger particle sizes (Supplementary Fig. S12a and b). This microstructural change reduced the interfacial contact area between LiF and FeF₂, which is essential for promoting the phase transition to T-FeF₃. Electrochemical testing further confirmed that insufficient LiF-FeF₂ contact deteriorates capacity but has limited impact on voltage hysteresis behavior. As milling time decreased, charge-discharge profiles (Supplementary Fig. S11c) showed reduced capacity due to poorer interfacial contact. This capacity degradation is attributed to the limited formation of electrochemically active T-FeF₃, resulting in compositional inhomogeneity. However, GITT (Galvanostatic Intermittent Titration Technique) analysis revealed that the difference in relaxed voltages between charge and discharge (V_{gap}), representing thermodynamic

hysteresis, remained nearly constant across all milling times and significantly smaller than that of R-FeF₃ (Supplementary Fig. S12d–e). The rest potential change (ΔV_{rest}), which reflects kinetic polarization, slightly increased at the end of charge as milling time decreased. Nevertheless, even in the shortest milling condition (0 h), ΔV_{rest} was still considerably smaller than in R-FeF₃ (Supplementary Fig. S12f), indicating that kinetic effects such as electron/ion diffusion barriers were minimal. This is likely because our nanocomposite system consists of sufficiently nanosized particles and a highly conductive carbon matrix, minimizing the impact of transport-related kinetic limitations.

In summary, the interfacial contact between LiF and FeF₂ plays a crucial role in the formation and electrochemical activation of T-FeF₃. In our nanocomposite system consisting of sufficiently small particles and an abundant conductive carbon matrix, changes in interfacial contact significantly affect capacity but have minimal impact on voltage hysteresis. Notably, T-FeF₃ exhibits consistently low hysteresis even under conditions of degraded interfacial contact, reinforcing the conclusion that the hysteresis difference between T-FeF₃ and R-FeF₃ arises primarily from their inherent thermodynamic reaction pathways, rather than from kinetic limitations such as electron or ion transport resistance.

Comments

10. *ex-XRD of Fig. 2a, is there any description for this figure?*

Author reply:

We carefully appreciate the reviewer's thorough comments. We recognize that our explanation of Fig. 2a was insufficient. Based on the reviewer's comment, we have revised the manuscript as follows.

Original text (Page 6)

Reversible and sequential intercalation and conversion reaction were confirmed for T-FeF₃ during charge and discharge. First, to investigate the reaction mechanism of T-FeF₃, *ex-situ* XRD was performed across various voltage ranges (Fig. 2a). The voltage regions, including each redox reaction around 4 V and 3 V, were categorized as the wide voltage range (WV, 4.8-2.0 V), the upper voltage range (UV, 4.8-3.4 V), and the lower voltage range (LV, 3.4-2.0 V). Fig. 2b shows the voltage profile and state of charge used for structural analysis, and Fig. 2c displays the phase fractions at each state obtained *via* Rietveld refinement (Supplementary Fig. S11).

Revised text (Page 6)

Reversible and sequential intercalation and conversion reaction were confirmed for T-FeF₃ during charge and discharge. First, to investigate the reaction mechanism of T-FeF₃, *ex-situ* XRD was performed across various voltage ranges. The voltage regions, including each

redox reaction around 4 V and 3 V, were categorized as the wide voltage range (WV, 4.8-2.0 V), the upper voltage range (UV, 4.8-3.4 V), and the lower voltage range (LV, 3.4-2.0 V). Fig. 2a shows the ex-situ XRD patterns during the charged and discharged process, including the charged state at 4.8 V (red), half-discharged state at 3.4 V (light red), discharged state at 2 V (purple), and recharged state at 4.8 V (blue). Additionally, Fig. 2b shows the voltage profile and state of charge used for structural analysis, and Fig. 2c displays the phase fractions at each state obtained *via* Rietveld refinement (Supplementary Fig. S13).

Comments

11. “only a change (0.03 Å, 0.3% increase) in the $c/3$ lattice parameter of the tetragonal phase (Li_xFeF_y , $x < 0.5$, and $0.5 < y < 3$). It was observed without a change in the phase fraction or occupancy” However, from XRD analysis, this phenomenon can not be observed. The peaks at 34.8 and 40o are also hard to observe, even after 10 cycles.

Author reply:

We sincerely appreciate the reviewer’s meticulous comments regarding the *ex-situ* XRD analysis. As demonstrated in Fig. R6, the gradual lattice change of T-FeF₃ during lithiation is clearly observed in the ex-situ XRD patterns through distinct peak shifts, confirming that the structural evolution is detectable.

Fig. R6. a, Ex-situ XRD patterns of T-FeF₃ for (1) charge to 4.8 V, (2) half-discharge to 3.4 V, and (3) discharge to 2 V. **b**, XRD patterns ($\lambda = 1.5406 \text{ \AA}$) for the main $P4_2/mnm$ structure with the largest phase fraction at each state. (Refer to Fig. 2a.)

During discharge, a gradual shift of the XRD peaks toward lower angles is observed when comparing Pattern 1 (charged state) and Pattern 2 (half-discharged state) in Fig. R6a. Notably,

the (003), (113), (213), and (303) peaks of T-FeF₃, corresponding to 34.8°, 40°, 52°, and 66.7°, respectively, exhibit clear shifts (Fig. R6b), indicative of c-axis lattice expansion. Among these, the (003), (113), and (303) peaks shift more prominently toward lower angles, resulting in peak broadening due to overlap with FeF₂ or LiF peaks present in the same state. This lattice expansion is further confirmed by Rietveld refinement, which quantitatively demonstrates the increase in the c-axis parameter (Supplementary Fig. S12). These results confirm that the observed XRD pattern changes arise from lattice parameter variation rather than changes in phase fraction or site occupancy.

Supplementary Fig. S12. Comparison of refined crystallographic parameters at each voltage. (a) Lattice a parameter, (b) lattice c parameter, and (c) volume.

Additionally, although T-FeF₃ and FeF₂ share the same space group (*P4₂/mnm*), their distinct lattice parameters produce differentiable diffraction peaks (Supplementary Fig. S12d). These peaks evolve and become more prominent as T-FeF₃ forms electrochemically over cycling (Fig. 1d).

Supplementary Fig. S12d. Crystal structure of T-FeF₃ and FeF₂. Since T-FeF₃ is similar to a three-fold stacking of the FeF₂ anion framework, the c lattice parameter and volume were expressed as Å/3 and Å³/3 to compare with FeF₂.

Fig. 1d, *Ex-situ* XRD patterns of LiF-FeF₂ electrodes at charged/discharged states after the 1st, 5th, and 10th cycles.

Based on the reviewer's suggestion, we have revised the manuscript for clarity and added Fig. R6 to explicitly demonstrate these peak shifts.

Original text (Page 6)

Reversible and sequential intercalation and conversion reaction were confirmed for T-FeF₃ during charge and discharge. First, to investigate the reaction mechanism of T-FeF₃, *ex-situ* XRD was performed across various voltage ranges (Fig. 2a). The voltage regions, including each redox reaction around 4 V and 3 V, were categorized as the wide voltage range (WV, 4.8-2.0 V), the upper voltage range (UV, 4.8-3.4 V), and the lower voltage range (LV, 3.4-2.0 V). Fig. 2b shows the voltage profile and state of charge used for structural analysis, and Fig. 2c displays the phase fractions at each state obtained *via* Rietveld refinement (Supplementary Fig. S11). At points 1 and 2 (within the UV range), only a change (0.03 Å, 0.3% increase) in the $c/3$ lattice parameter of the tetragonal phase (Li_xFeF_y, $x < 0.5$, and $0.5 < y < 3$) was observed without a change in the phase fraction or occupancy, suggesting Li⁺ insertion into the host structure of T-FeF₃ in the UV region (Supplementary Fig. S12 and Fig. 2c). As shown in Supplementary Fig. S12, during further discharge from point 2 to point 3 (within the LV range), the a and $c/3$ lattice parameters of the tetragonal phase increased by 0.8% and 1.6%, respectively, and become similar to those of FeF₂. Furthermore, the ratio of Fe to F significantly decreased from 1:2.96 (point 2) to 1:2.59 (point 3). At point 3, the similarity of the lattice parameters and Fe–F ratio of the tetragonal phase and FeF₂ as well as the phase increase of LiF and FeF₂ with the consumption of the tetragonal phase indicates that the conversion reaction of the tetragonal phase to FeF₂ occurs near 3 V.

Revised text (Page 6-7)

Reversible and sequential intercalation and conversion reaction were confirmed for T-FeF₃

during charge and discharge. First, to investigate the reaction mechanism of T-FeF₃, *ex-situ* XRD was performed across various voltage ranges. The voltage regions, including each redox reaction around 4 V and 3 V, were categorized as the wide voltage range (WV, 4.8-2.0 V), the upper voltage range (UV, 4.8-3.4 V), and the lower voltage range (LV, 3.4-2.0 V). Fig. 2a shows the *ex-situ* XRD patterns during the charged and discharged process, including the charged state at 4.8 V (red), half-discharged state at 3.4 V (light red), discharged state at 2 V (purple), and recharged state at 4.8 V (blue). Additionally, Fig. 2b shows the voltage profile and state of charge used for structural analysis, and Fig. 2c displays the phase fractions at each state obtained *via* Rietveld refinement (Supplementary Fig. S13). As shown in Fig. 2a and Supplementary Fig. S14, the *ex-situ* XRD data of T-FeF₃ indicate that the diffraction pattern largely retains the diffraction patterns of *P4₂/mnm* structure throughout the charge-discharge process, while certain peaks exhibit shifts and new peaks gradually emerge (Supplementary Note 6). At points 1 and 2 (within the UV range), only a small expansion (0.03 Å, a 0.3% increase) in the *c*/3 lattice parameter of the tetragonal phase (Li_xFeF_y, $x < 0.5$, and $0.5 < y < 3$) was observed, with no noticeable change in phase fraction or occupancy. This gradual shift in the XRD peaks, without a significant change in phase fraction, suggests that the structural evolution in this region is primarily driven by lattice parameter changes rather than a phase transformation, indicating that Li⁺ insertion occurs within the host structure of T-FeF₃ in the UV region rather than triggering a phase transition (Supplementary Figure S15 and Fig. 2c). However, as shown in Supplementary Fig. S15, during further discharge from point 2 to point 3 (within the LV range), the *a* and *c*/3 lattice parameters of the tetragonal phase increased by 0.8% and 1.6%, respectively, and become similar to those of FeF₂. Furthermore, the ratio of Fe to F significantly decreased from 1:2.96 (point 2) to 1:2.59 (point 3). At point 3, the similarity of the lattice parameters and Fe–F ratio of the tetragonal phase and FeF₂ as well as the phase increase of LiF and FeF₂ with the consumption of the tetragonal phase indicates that the conversion reaction of the tetragonal phase to FeF₂ occurs near 3 V.

Added supplementary Figure

Supplementary Fig. S14. (a) Crystal structure of T-FeF₃ and FeF₂. Since T-FeF₃ is similar to a three-fold stacking of the FeF₂ anion framework. (b) Ex-situ XRD patterns of T-FeF₃ for (1) charge to 4.8 V, (2) half-discharge to 3.4 V, and (3) discharge to 2 V. (c) XRD patterns ($\lambda = 1.5406 \text{ \AA}$) for the main $P4_2/mnm$ structure with the largest phase fraction at each state. (See Fig. 2a-c.)

Supplementary Note 6. XRD Pattern Changes During Discharge of T-FeF₃.

Although T-FeF₃ and FeF₂ share the same space group ($P4_2/mnm$), their unit cell parameters differ, leading to distinct diffraction patterns for T-FeF₃ compared to FeF₂ (Supplementary Fig. S14a and 14b). These peaks become increasingly pronounced as T-FeF₃ is electrochemically formed during the initial cycles, as shown in Fig. 1d. During discharge, a gradual shift of the XRD peaks toward lower angles is observed when comparing patterns of point 1 (charged state) and point 2 (half-discharged state) in Supplementary Fig. S14c. Specifically, shifts in the (003), (113), (213), and (303) peaks of T-FeF₃, corresponding to 34.8°, 40°, 52°, and 66.7°, are clearly identifiable. Among these, the (003), (113), and (303) peaks shift to lower angles, leading to peak broadening due to overlap with FeF₂ or LiF peaks present in the same state. At point 3 (discharged state), only FeF₂ and LiF peaks are sharply observed (Supplementary Fig. S14b).

Original supplementary Figure

Supplementary Fig. S12. Comparison of refined crystallographic parameters at each voltage. **(a)** Lattice a parameter, **(b)** lattice c parameter, and **(c)** volume. **(d)** Crystal structure of T-FeF₃ and FeF₂. Since T-FeF₃ is similar to a three-fold stacking of the FeF₂ anion framework, the c lattice parameter and volume were expressed as Å/3 and Å³/3 to compare with FeF₂. (See Supplementary Table S3 and S4.)

Revised supplementary Figure

Supplementary Fig. S15. Comparison of refined crystallographic parameters at each voltage. **(a)** Lattice a parameter, **(b)** lattice c parameter, and **(c)** volume. **The c lattice**

parameter and volume of T-FeF₃ were expressed as Å³/3 and Å³/3 to compare with FeF₂. (See Supplementary Tables S3 and S4.)

Comments

12. The scale bar in Fig. S22 Figure S23 should be given.

Author reply:

Thank you for your comments. Based on the reviewer's comments, we have revised the figure caption as follows:

Original supplementary Figure

Supplementary Fig. S22. (a,b) STEM-EELS images of T-FeF₃ and R-FeF₃ at 10th charged

state for the energy distribution of the Fe L_3 -edge peak. **(c,d)** Fe L_3 -edge peak energy and the L_3/L_2 ratio of T-FeF₃ and R-FeF₃ at 10th charged state.

Revised supplementary Figure

Supplementary Fig. S26. **(a,b)** STEM-EELS images of T-FeF₃ and R-FeF₃ at 10th charged state for the energy distribution of the Fe L_3 -edge peak. **(c,d)** Fe L_3 -edge peak energy and the L_3/L_2 ratio of T-FeF₃ and R-FeF₃ at 10th charged state. The line represents the range of values measured multiple times, while the square indicates their respective average values. The scale bar in all images represents 20 nm.

Original supplementary Figure

Supplementary Fig. S24. (a,b) STEM-EELS images of T-FeF₃ and R-FeF₃ at 10th discharged state for the energy distribution of the Fe L₃-edge peak. (c,d) Fe L₃-edge peak energy and the L₃/L₂ ratio of T-FeF₃ and R-FeF₃ at 10th discharged state.

Revised supplementary Figure

Comments

13. For R-FeF₃ and T-FeF₃ samples, EELS should be provided after 300 cycles to support the conclusion in Fig. 6b.

Author reply:

We sincerely thank the reviewer for the thoughtful suggestion to include EELS analysis

after long-term cycling to support the conclusions in Fig. 6b. We fully agree that compositional analysis after extended cycling is critical to validating the structural stability and redox homogeneity of R-FeF₃ and T-FeF₃.

Due to the time constraints associated with the revision process, completing 300 cycles at a low current density (20 mA g⁻¹) was not feasible within the allowed period. To obtain meaningful comparative insight under practical conditions, we instead conducted 100 cycles at an elevated current density of 100 mA g⁻¹, followed by EELS analysis of both T-FeF₃ and R-FeF₃ samples at the charged state.

Fig. R7a. Cycle stability at 100 mA g⁻¹ of T-FeF₃ and R-FeF₃.

As shown in Fig. R7a, T-FeF₃ maintains excellent capacity retention (87%) even at 100 mA g⁻¹, whereas R-FeF₃ exhibits capacity fading (70%), confirming that this accelerated protocol still provides a suitable platform to differentiate the long-term stability of the two materials.

Fig. R7b-c. STEM-EELS images of T-FeF₃ and R-FeF₃ at charged state after 100th cycle for the energy distribution of the Fe *L*₃-edge peak. The charge state of T-FeF₃ (TC) and R-FeF₃ (RC). The most oxidized and most reduced regions within the particle are marked 1 and 2, respectively. The scale bar in all images represents 50 nm.

STEM-EELS mapping (Fig. R7b and c) revealed that both materials show some degree of Fe²⁺/Fe³⁺ mixed regions—likely due to the faster cycling rate. However, R-FeF₃ displayed a more pronounced Fe²⁺-rich domain, particularly in localized areas, while T-FeF₃ showed a more homogeneous oxidation state. Notably, even in the most oxidized region (Region 1), T-FeF₃ retained a higher proportion of Fe³⁺ compared to R-FeF₃, suggesting better oxidation uniformity.

Fig. R7d. EELS spectra of Fe $L_{3,2}$ -edge peak energy of TC and RC.

Fig. R7d further confirms this difference through Fe $L_{3,2}$ -edge spectra, showing that R- FeF_3 accumulates reduced Fe species more readily than T- FeF_3 . This behavior is consistent with its lower structural reversibility and more severe compositional inhomogeneity.

While the faster cycling rate may have increased the overall inhomogeneity, the observed difference between R- FeF_3 and T- FeF_3 still supports our original conclusion that T- FeF_3 maintains better compositional stability during cycling. These additional EELS results have been included in the revised manuscript to address the reviewer's comment.

We hope this clarifies our experimental approach and sufficiently addresses the reviewer's concern.

Original text (Page 12)

By comparing the reaction mechanism between R- FeF_3 ^[21] and T- FeF_3 , both materials commonly show intercalation and conversion reactions (Fig. 6b). Although the product of the conversion reaction (until 2 V) is the same as FeF_2 in both cases, the reversibility is much better for T- FeF_3 than R- FeF_3 due to the similarity of the structure. R- FeF_3 undergoes a structural change from a corner-sharing FeF_6 group structure (rcp) to an edge-sharing tetragonal phase (tcp) during charging and discharge^[19,21,56,60] It is understood that the significant structural difference between the two structures causes an irreversible phase transition, which leads to compositional inhomogeneity²¹ (Supplementary Fig. S24 and Supplementary Note 7)

Revised text (Page 13)

By comparing the reaction mechanism between R- FeF_3 ^[21] and T- FeF_3 , both materials commonly show intercalation and conversion reactions (Fig. 6b). **Although the product of the conversion reaction (until 2 V) is the same as FeF_2 and LiF in both cases (Fig. 2a and**

Supplementary Fig. S40), the reversibility is much better for T-FeF₃ than R-FeF₃ due to the structural similarity with FeF₂. This difference is further validated by STEM–EELS analysis performed after 100 cycles. Since it was measured at 100 mA g⁻¹, both T-FeF₃ and R-FeF₃ show a mixed Fe²⁺/Fe³⁺ state in the charged state. However, R-FeF₃ displays a significantly greater presence of reduced Fe²⁺ regions and lower Fe³⁺ intensity even in the most oxidized areas, indicating more pronounced compositional inhomogeneity compared to T-FeF₃ (Supplementary Fig. S41). R-FeF₃ undergoes a structural change from a corner-sharing FeF₆ group structure (rcp) to an edge-sharing tetragonal phase (tcp) during charging and discharge^[19,21,56,60] It is understood that the significant structural difference between the two structures causes an irreversible phase transition, which leads to compositional inhomogeneity²¹ (Supplementary Fig. S25 and Supplementary Note 9).

Added supplementary Figure

Comments

14. Please add the figure number here.” (Supplementary Fig. S33). Fig. shows the capacity retention under this over-deep discharge condition.”

Author reply:

We are very grateful for your thorough comments on our manuscript. As the reviewer's comments, we have revised our manuscript as follows:

Original text (Page 11)

Therefore, the superior reversibility of T-FeF₃ may be attributed to the seed of remaining tetragonal phase even in the discharged state. To confirm this assumption, electrochemical evaluation was conducted in an over-deep discharge voltage range (4.8–1.2 V) to remove the seeds of the tetragonal phase (Supplementary Fig. S33). Fig. shows the capacity retention under this over-deep discharge condition. At point 3 in Fig. 5e, following the over-deep discharge, significant capacity decay and failure to clearly recover the 4 V redox peak were observed (Fig. 5f-h and Supplementary Fig. S34). These results imply that T-FeF₃ can achieve reversible structural recovery if the phase seed remains, whereas irreversible structural changes occur in the case of R-FeF₃ regardless of the presence of the phase seed due to the irreversible reactions accompanying the reaction mechanism.

Revised text (Page 11-12)

Therefore, the superior reversibility of T-FeF₃ may be attributed to the seed of remaining tetragonal phase even in the discharged state. To confirm this assumption, electrochemical evaluation was conducted in an over-deep discharge voltage range (4.8–1.2 V) to remove the seeds of the tetragonal phase (Supplementary Fig. S38). Fig. 5e shows the capacity retention under this over-deep discharge condition. At point 3 in Fig. 5e, following the over-deep discharge, significant capacity decay and failure to clearly recover the 4 V redox peak were observed (Fig. 5f-h and Supplementary Fig. S39). These results imply that T-FeF₃ can achieve reversible structural recovery if the phase seed remains, whereas irreversible structural changes occur in the case of R-FeF₃ regardless of the presence of the phase seed due to the irreversible reactions accompanying the reaction mechanism.

Reviewer #3(Remarks to the Author):

Comments to the Author

The manuscript presents a significant advancement in lithium-ion battery technology by introducing a guided phase transition approach for iron fluoride cathode materials. The transformation of a LiF-FeF₂ nanocomposite into metastable tetragonal FeF₃ (T-FeF₃) demonstrates reduced voltage hysteresis and enhanced structural reversibility. This is a substantial contribution to the field, as it addresses persistent issues in conversion-reaction-based cathode materials, including compositional inhomogeneity and poor cycle stability. The experimental data, supported by detailed DFT calculations, convincingly demonstrate the superiority of T-FeF₃ over rhombohedral FeF₃ (R-FeF₃) in terms of electrochemical performance and structural stability.

This work is of high significance to the field of energy storage and conversion. The proposed T-FeF₃ material, derived from a nanocomposite strategy, offers a novel and effective solution to mitigate voltage hysteresis and structural degradation, problems widely reported in conversion-reaction materials. The findings align with and extend prior studies, such as those focusing on Na_{0.7}CoO₂ and LiF-MnO systems, by introducing a new paradigm in the design of cathode materials. The comparison to the established literature is thorough, and the study builds on previous insights by providing a thermodynamic and kinetic framework for improving reversibility and cycling stability. The references cited are appropriate and comprehensive.

The conclusions drawn are well-supported by the experimental and computational results. The authors demonstrate the formation and stability of T-FeF₃ through XRD, TEM, and EXAFS analyses, as well as the material's enhanced performance through electrochemical testing. The DFT calculations further strengthen the argument, providing insights into the thermodynamic stability and reaction pathways of T-FeF₃ compared to R-FeF₃. However, additional details on the reproducibility of the synthesis method and its scalability could strengthen the conclusions further.

The methodology is sound and meets the expected standards for research in this domain. The combination of experimental techniques and theoretical modeling is comprehensive and provides a holistic understanding of the material properties and performance. The methods for nanocomposite synthesis, electrochemical testing, and structural characterization are described in sufficient detail to allow replication. The use of advanced tools like STEM-EELS and GITT for analyzing compositional homogeneity and hysteresis adds rigor to the study.

The data analysis is robust, and the interpretations are logical and well-aligned with the results. The study effectively correlates structural properties with electrochemical performance, providing a clear narrative for the advantages of T-FeF₃. However, the manuscript could benefit from additional discussion on potential limitations, such as the impact of prolonged cycling at high current densities or the effect of varying the nanocomposite composition.

Author reply:

We are grateful for the reviewer's valuable and constructive comments and especially appreciate the positive evaluation of the message we intend to convey through our study. In the revision, we conducted additional experiments that can support and reinforce our conclusions. We sincerely hope that this revision relieves the reviewer's concerns.

Comments

1. *While the manuscript is comprehensive, additional long-term cycling data at varying current densities and a comparison with other conversion-reaction cathode materials could provide further validation*

Author reply:

We sincerely appreciate the valuable comments. As the reviewer pointed out, comparing the cycling performance of T-FeF₃ with other conversion-reaction materials is crucial for objectively evaluating its performance and enhancing its generalizability.

In general, cathode materials based on conversion reactions offer high theoretical capacities. However, they often suffer from significant volume expansion and electrochemical irreversibility during charge and discharge cycles, leading to structural degradation, a decline in electrical conductivity, and ultimately, poor cycling stability.

To further validate the robustness of our system, we conducted long-term cycling tests of T-FeF₃ at various current densities (Fig. R8). The results demonstrated that T-FeF₃ maintains stable cycling performance with 68% capacity retention at 100 cycles even at 1000 mA g⁻¹ (approximately 4.5C). These findings underscore the advantages of T-FeF₃ in terms of both cycling stability and structural integrity, even under demanding cycling conditions.

Fig. R8. Cycle stability of T-FeF₃ at various current densities.

Additionally, we agree with the reviewer that comparing T-FeF₃ with other conversion-reaction cathode materials is crucial for generalizing its superior performance. Among conversion-reaction materials, Fe-based fluorides (e.g., FeF₃, FeF₂) exhibit superior cycling stability due to the inherently high reversibility of Fe redox reactions. This stability helps them retain structural integrity better than other transition-metal-based materials, leading to enhanced cycling performance over prolonged charge-discharge cycles (*Chem. - Eur. J.* **24**, 7177 (2018); *Nano Res.* **10**, 4232 (2017); *Nat. Commun.*, **6**, 6668 (2015); *J. Power Sources*, 521, 230935 (2022); *Small*, **15**, 1804670 (2019); *Nat. Commun.* **9**, 2324 (2018); *Adv. Funct. Mater.* **27**, 1702783 (2017)).

Accordingly, in response to the reviewer's suggestion, we expanded our comparison to include not only Fe-based fluorides but also iron oxyfluoride materials reported in previous studies, providing a broader context to further validate our results (Supplementary Fig. S25).

Supplementary Fig. S25. Comparison of post-cycle discharge capacity of iron fluoride materials composited with carbon and T-FeF₃. (See Supplementary Table S5.)

Based on the reviewer's insightful comments, the figure has been revised as follows:

Original supplementary Figure

Supplementary Fig. S24. (a,b) Electrochemical profile of T-FeF₃ and R-FeF₃ corresponding charge/discharge profiles at various current densities.

Revised supplementary Figure

Supplementary Fig. S28. (a,b) Electrochemical profile of T-FeF₃ and R-FeF₃ corresponding charge/discharge profiles at various current densities. **(c) Cycle stability of T-FeF₃ at various current densities.**

Original supplementary Figure

Supplementary Fig. S25. Comparison of post-cycle discharge capacity of iron fluoride materials composited with carbon and T-FeF₃. (See Supplementary Table S5.)

Revised supplementary Figure

Supplementary Fig. S29. (a, b) Comparison of capacity retention and post-cycle discharge capacity for iron fluoride materials composited with carbon and FeF₃ (rhombohedral structure) and other Fe-based conversion-reaction materials (see Supplementary Table S5).

Original supplementary information

Supplementary Table S3. Comparison of previously reported iron fluoride materials composited with carbon.

No.	ref.	sample	Theoretical	Initial discharge	Cycling number	Current (mA g ⁻¹)	Post-cycle discharge	capacity retention	Voltage range
			Capacity (mAh g ⁻¹)	Capacity (mAh g ⁻¹)			capacity (mAh g ⁻¹)		
1	(10)	FeF ₃ /C (70:25)	712 (3Li)	650	30	71.2	360	55.40%	1.0-4.5V
				620			200		
				650			200		

				645			360	55.80%	
2	(11)	FeF ₃	712 (3Li)	210.51	40	50	54.42	25.90%	1.0-4.0V
		FeF ₃ /AB(85:15)		346.25			161.58	46.70%	
3	(12)	FeF ₃ -H-rGO	712 (3Li)	690	110	100	200	29.00%	1.0-4.5V
		FeF ₃		520			90	17.30%	
4	(13)	FeF ₃ /AB (70:25)	712 (3Li)	510	15	71.2	180	35.30%	1.0-4.5V
5	(14)	FeF ₃ /C-SWNT	712 (3Li)	300	100	300	190	63.33%	1.0-4.5V
		FeF ₃		200			60	30.00%	1.0-4.0V
6	(15)	FeF ₃ /C	475 (2Li)	170	53	20	89	52.40%	1.5-4.5V
7	(16)	bare FeF ₃	475 (2Li)	598	170	100	80	13.40%	1.5-4.5V
8	(17)	FeF ₃ /r-GO1.7	237 (1Li)	195	50	23.7	170	87.18%	2.0-4.5 V
			475 (2Li)	414		71.2	274	66.18%	1.5-4.5 V
			237 (1Li)	114		23.7	50	43.86%	2.0-4.5 V
			475 (2Li)	314		71.2	35	11.15%	1.5-4.5 V
9	(18)	FeF ₃	712 (3Li)	710	12	14.3	140	19.72%	1.5-4.5 V
10	(19)	FeF ₃ /AB (85:15)		129.3		237	101.9	78.80%	2.0V-4.5V
			237 (1Li)	119	100	474	83.6	70.30%	
				105.1		1185	71.2	67.70%	
11	(20)	FeF ₃ /AB (70:25)	237 (1Li)	210	25	10	100	47.60%	2.0V-4.5V
		FeF ₃ /AB-HT		200	50	10	168	84.00%	
12	(21)	FeF ₃ /C composite	237 (1Li)	170	100	250	127	74.71%	2.0-4.5V
		FeF ₃		40			30	75.00%	
13	(22)	FCO (FeF ₃ @carbon nanocomposite)	237 (1Li)	120	200	474	45	37.50%	2.0-4.5V
		FCN (FeF ₃ @N-doped carbon nanocomposite)		140			100	71.43%	

Revised supplementary information

Supplementary Table S4. Comparison of previously reported iron fluoride materials composited with carbon and Fe-based conversion reaction materials.

No.	ref.	Sample	Theoretical	Initial discharge	Cycling number	Current (mA g ⁻¹)	Post-cycle discharge	capacity retention	Voltage range
			Capacity (mAh g ⁻¹)	Capacity (mAh g ⁻¹)			capacity (mAh g ⁻¹)		
This work		T-FeF ₃	224 (1Li)	205.11	114	20	182.68	89.06%	2.0-4.8V
				178.49	300	50	128.85	72.19%	

		R-FeF ₃	237 (1Li)	185.19 164.74	114 300	20 50	124.23 83.89	67.08% 50.92%	
				650			360	55.40%	
1	(20)	FeF ₃ /C (70:25)	712 (3Li)	620	30	71.2	200	32.30%	1.0-4.5V
				650			200	30.80%	
				645			360	55.80%	
2	(21)	FeF ₃	712 (3Li)	210.51	40	50	54.42	25.90%	1.0-4.0V
		FeF ₃ /AB(85:15)		346.25			161.58	46.70%	
3	(22)	FeF ₃ -H-rGO	712 (3Li)	690	110	100	200	29.00%	1.0-4.5V
		FeF ₃		520			90	17.30%	
4	(23)	FeF ₃ /AB (70:25)	712 (3Li)	510	15	71.2	180	35.30%	1.0-4.5V
5	(24)	FeF ₃ /C-SWNT	712 (3Li)	300	100	300	190	63.33%	1.0-4.5V
		FeF ₃		200			60	30.00%	1.0-4.0V
6	(25)	FeF ₃ /C	475 (2Li)	170	53	20	89	52.40%	1.5-4.5V
7	(26)	bare FeF ₃	475 (2Li)	598	170	100	80	13.40%	1.5-4.5V
8	(27)	FeF ₃ /r-GO1.7	237 (1Li)	195		23.7	170	87.18%	2.0-4.5 V
			475 (2Li)	414	50	71.2	274	66.18%	1.5-4.5 V
		FeF ₃	237 (1Li)	114		23.7	50	43.86%	2.0-4.5 V
			475 (2Li)	314		71.2	35	11.15%	1.5-4.5 V
9	(28)	FeF ₃	712 (3Li)	710	12	14.3	140	19.72%	1.5-4.5 V
10	(29)	FeF ₃ /AB (85:15)	237 (1Li)	129.3 119	100	237 474	101.9 83.6	78.80% 70.30%	2.0V-4.5V
				105.1		1185	71.2	67.70%	
11	(30)	FeF ₃ /AB (70:25)	237 (1Li)	210	25	10	100	47.60%	2.0V-4.5V
		FeF ₃ /AB-HT		200	50	10	168	84.00%	
12	(31)	FeF ₃ /C composite	237 (1Li)	170	100	250	127	74.71%	2.0-4.5V

13	(32)	FCO (FeF ₃ @carbon nanocomposite)	237 (1Li)	120	200	474	45	37.50%	2.0- 4.5V
		FCN (FeF ₃ @N- doped carbon nanocomposite)		140			100	71.43%	
14	(25)	C/FeOF/FeF ₃		438.3	50	20	120	27.38%	1.5- 4.5V
15	(33)	LiF-FeO	274 (1Li)	140	100	50	160	114.29%	1.5- 4.8V
16	(34)	FeOF	885.1 (3Li)	550	100	70	300	54.55%	1.2- 4.0V
17	(35)	FeOF	885.1 (3Li)	300	100	50	33	11.00%	1.0- 3.8V
18	(36)	FeOF	885.1 (3Li)	345.1	100	100	250.3	72.53%	1.3- 4.0V
19	(37)	FeOF	885.1 (3Li)	520	50	50	300	57.69%	1.0- 4.0V
20	(38)	FeOF@CN	885.1 (3Li)	194	50	100	104	53.61%	1.2- 4.0V
21	(39)	bare FeOF	885.1 (3Li)	600	100	100	20.6	3.43%	1.2- 4.0V

Added references

REFERENCES

20. Senoh, H. et al. Degradation Mechanism of Conversion-Type Iron Trifluoride: Toward Improvement of Cycle Performance. *ACS Appl. Mater. Interfaces* **11**, 30959–30967 (2019).
21. Tang, M. et al. High-Temperature Electrochemical Performance of FeF₃/C Nanocomposite as a Cathode Material for Lithium-Ion Batteries. *J. Mater. Eng. Perform.* **27**, 624–629 (2018).
22. Zhao, X., Hayner, C. M., Kung, M. C. & Kung, H. H. Photothermal-assisted fabrication of iron fluoride–graphene composite paper cathodes for high-energy lithium-ion batteries. *Chem. Commun.* **48**, 9909–9911 (2012).
23. Tawa, S., Yamamoto, T., Matsumoto, K. & Hagiwara, R. Iron(III) fluoride synthesized by a fluorolysis method and its electrochemical properties as a positive electrode material for lithium secondary batteries. *J. Fluor. Chem.* **184**, 75–81 (2016).
24. Chen, T. et al. Liquid phase exfoliation of nonlayered non-van der Waals iron trifluoride (FeF₃) into 2D-platelets for high-capacity lithium storing cathodes. *FlatChem* **33**, 100360 (2022).
25. Li, W. et al. The facile in situ preparation and characterization of C/FeOF/FeF₃ nanocomposites as LIB cathode materials. *Ionics* **24**, 1561–1569 (2018).
26. Li, L., Zhu, J., Xu, M., Jiang, J. & Li, C. M. In Situ Engineering Toward Core Regions: A Smart Way to Make Applicable FeF₃@Carbon Nanoreactor Cathodes for Li-Ion Batteries. *ACS Appl. Mater. Interfaces* **9**, 17992–18000 (2017).

27. Jung, H., Song, H., Kim, T., Lee, J. K. & Kim, J. FeF₃ microspheres anchored on reduced graphene oxide as a high performance cathode material for lithium ion batteries. *J. Alloys Compd.* **647**, 750–755 (2015).
28. Martha, S. K. et al. Electrode architectures for high capacity multivalent conversion compounds: Iron (ii and iii) fluoride. *RSC Adv.* **4**, 6730–6737 (2014).
29. Liu, L. et al. Excellent cycle performance of Co-doped FeF₃/C nanocomposite cathode material for lithium-ion batteries. *J. Mater. Chem.* **22**, 17539–17550 (2012).
30. Yabuuchi, N. et al. Effect of heat-treatment process on FeF₃ nanocomposite electrodes for rechargeable Li batteries. *J. Mater. Chem.* **21**, 10035–10041 (2011).
31. Lee, J. & Kang, B. Novel and scalable solid-state synthesis of a nanocrystalline FeF₃/C composite and its excellent electrochemical performance. *Chem. Commun.* **52**, 9414–9417 (2016).
32. Li, J. et al. Improved Electrochemical Performance of FeF₃ by Inlaying in a Nitrogen-Doped Carbon Matrix. *ChemElectroChem* **6**, 5203–5210 (2019).
33. Jung, S. K. et al. New Iron-Based Intercalation Host for Lithium-Ion Batteries. *Chem. Mater.* **30**, 1956–1964 (2018).
34. Fan, X. et al. High energy-density and reversibility of iron fluoride cathode enabled via an intercalation-extrusion reaction. *Nat. Commun.* **9**, 2324 (2018).
35. Zhou, H. et al. Phosphorus-Doped FeOF Nanoparticle-Based Cathodes for Lithium Storage. *ACS Appl. Nano Mater.* **5**, 13444–13454 (2022).
36. Zhai, J., Lei, Z., Sun, K. & Zhu, S. MXene enabled binder-free FeOF cathode with high volumetric and gravimetric capacities for flexible lithium ion batteries. *Electrochim. Acta* **423**, 140595 (2022).
37. Lin, Y. et al. Boosting the intercalation reaction of FeOF-based cathode toward highly reversible lithium storage. *Nano Energy* **128**, 109944 (2024).
38. Li, W. et al. FeOF/TiO₂Hetero-Nanostructures for High-Areal-Capacity Fluoride Cathodes. *ACS Appl. Mater. Interfaces* **12**, 33803–33809 (2020).
39. Xiao, S. et al. Versatile metal fluorides in ion battery application. *J. Mater. Chem. A* **12**, 20783–20802 (2024).

Comments

2. Additionally, an exploration of the impact of impurities or variations in the precursor materials would strengthen the generalizability of the findings.

Author reply:

Thank you for your valuable comments. In order to examine the effect of precursor material purity that you pointed out, we conducted additional comparisons using LiF with 99.85% and 97% purity, in addition to the 99.99% LiF which is originally used in the main text. When looking at the XRD patterns (Fig. R9a), no significant differences were observed among the samples. However, in the charge–discharge profiles of the 10th cycle where T-FeF₃ is electrochemically formed (Fig. R9b), the characteristic 4 V plateau of T-FeF₃ appeared relatively less distinct in the 97% purity LiF sample (black curve). Although the difference is subtle, we believe it clearly indicates how crucial contact between LiF and FeF₂ is for T-FeF₃

formation. Consequently, we confirmed that even slight differences in precursor purity can influence electrode performance, emphasizing the need to carefully manage precursor purity to enhance reproducibility and reliability.

Fig.R9. (a) XRD patterns of LiF-FeF₂ nanocomposites prepared using LiF of different purities. (d) Electrochemical profile and differential analysis of voltage profile at 10th cycle.

Original text (Page 6)

This result is attributed to the shorter Fe–Fe bond distance of T-FeF₃ (3.16 and 3.69 Å) with its edge-sharing framework, compared to R-FeF₃ (~3.7 Å), which only has a corner-sharing framework of iron octahedra (Supplementary Fig. S4b and Supplementary Note 3). Moreover, for the pair distribution function (PDF) analysis, the 10th charge state was more consistent with T-FeF₃ than with R-FeF₃ (Supplementary Fig. S9). Conclusively, Rietveld refinement of the XRD pattern of the electrode in the 10th charged state (Supplementary Fig. S10 and Supplementary Note 4) revealed an excellent match with the T-FeF₃ phase including residual LiF and FeF₂. These results confirm that the LiF–FeF₂ nanocomposite successfully leads to the gradual formation of T-FeF₃ while maintaining structural similarity to the mother structure (FeF₂). Notably, tetragonal FeF₃ derived from LiF-FeF₂ nanocomposites offers practical advantages in full-cell manufacturing^[43] compared to the previously reported tetragonal FeF₃ phase is formed *via* the delithiation of Li_{0.5}FeF₃, particularly due to the safety concerns and chemical instability associated with metallic lithium and lithium-containing anode electrodes^[4].

Revised text (Page 6)

This result is attributed to the shorter Fe–Fe bond distance of T-FeF₃ (3.16 and 3.69 Å) with its edge-sharing framework, compared to R-FeF₃ (~3.7 Å), which only has a corner-sharing framework of iron octahedra (Supplementary Fig. S4b and Supplementary Note 3). Moreover, for the pair distribution function (PDF) analysis, the 10th charge state was more consistent with T-FeF₃ than with R-FeF₃ (Supplementary Fig. S9). Conclusively, Rietveld

refinement of the XRD pattern of the electrode in the 10th charged state (Supplementary Fig. S10 and Supplementary Note 4) revealed an excellent match with the T-FeF₃ phase including residual LiF and FeF₂. These results confirm that the LiF-FeF₂ nanocomposite successfully leads to the gradual formation of T-FeF₃ while maintaining structural similarity to the mother structure (FeF₂). The efficient formation of T-FeF₃ is closely governed by the interfacial contact between LiF and FeF₂, which facilitates the guided phase transition during cycling (Supplementary Note 5). Notably, tetragonal FeF₃ derived from LiF-FeF₂ nanocomposites offers practical advantages in full-cell manufacturing^[43] compared to the previously reported tetragonal FeF₃ phase is formed *via* the delithiation of Li_{0.5}FeF₃, particularly due to the safety concerns and chemical instability associated with metallic lithium and lithium-containing anode electrodes^[4].

Added supplementary Figure and Note

Supplementary Fig. S11. (a) XRD patterns of LiF-FeF₂ nanocomposites prepared using LiF of different purities. (d) Electrochemical profile and differential analysis of voltage profile at 10th cycle.

Supplementary Note 5. Role of LiF-FeF₂ Interfacial Integrity in Phase Evolution and Electrochemical Performance of T-FeF₃.

The interfacial contact between LiF and FeF₂ in the LiF-FeF₂ nanocomposite plays a critical role in facilitating the guided phase transition toward T-FeF₃ and determining the overall electrochemical performance. To systematically investigate the impact of this interfacial contact, we examined two key factors: the purity of the LiF precursor, which influences the chemical integrity of the composite, and the ball-milling time, which affects particle size and mixing uniformity.

The critical role of LiF-FeF₂ interfacial contact in forming T-FeF₃ was further supported by investigating the effect of LiF precursor purity. To evaluate this, we prepared LiF-FeF₂ composites using LiF with purities of 99.99%, 99.85%, and 97%. XRD analysis revealed no significant differences in crystal structure or phase composition among the samples,

indicating that variations in purity did not cause noticeable changes in the bulk structure (Supplementary Fig. S11a). However, electrochemical measurements revealed that lowering the LiF purity led to a slight but consistent reduction in capacity. This capacity loss is likely due to the presence of impurities, which may disrupt the LiF-FeF₂ interface, reduce effective contact, or introduce electrochemically inactive phases that interfere with smooth T-FeF₃ formation (Supplementary Fig. S11b).

To further confirm the importance of interfacial contact, we investigated the effect of crystallinity by systematically varying the ball-milling time. By adjusting only the initial LiF-FeF₂ mixing duration (48 h, 12 h, and 0 h) while maintaining the carbon mixing step constant at 12 h (500 rpm), we were able to isolate the impact of interfacial contact and crystallinity on electrochemical behavior. XRD analysis revealed that reducing the milling time resulted in narrower diffraction peaks and a noticeable decrease in the full width at half maximum (FWHM), indicating increased crystallinity and larger particle sizes (Supplementary Fig. S12a and b). This microstructural change reduced the interfacial contact area between LiF and FeF₂, which is essential for promoting the phase transition to T-FeF₃. Electrochemical testing further confirmed that insufficient LiF-FeF₂ contact deteriorates capacity but has limited impact on voltage hysteresis behavior. As milling time decreased, charge–discharge profiles (Supplementary Fig. S11c) showed reduced capacity due to poorer interfacial contact. This capacity degradation is attributed to the limited formation of electrochemically active T-FeF₃, resulting in compositional inhomogeneity. However, GITT (Galvanostatic Intermittent Titration Technique) analysis revealed that the difference in relaxed voltages between charge and discharge (V_{gap}), representing thermodynamic hysteresis, remained nearly constant across all milling times and significantly smaller than that of R-FeF₃ (Supplementary Fig. S12d–e). The rest potential change (ΔV_{rest}), which reflects kinetic polarization, slightly increased at the end of charge as milling time decreased. Nevertheless, even in the shortest milling condition (0 h), ΔV_{rest} was still considerably smaller than in R-FeF₃ (Supplementary Fig. S12f), indicating that kinetic effects such as electron/ion diffusion barriers were minimal. This is likely because our nanocomposite system consists of sufficiently nanosized particles and a highly conductive carbon matrix, minimizing the impact of transport-related kinetic limitations.

In summary, the interfacial contact between LiF and FeF₂ plays a crucial role in the formation and electrochemical activation of T-FeF₃. In our nanocomposite system consisting of sufficiently small particles and an abundant conductive carbon matrix, changes in interfacial contact significantly affect capacity but have minimal impact on voltage hysteresis. Notably, T-FeF₃ exhibits consistently low hysteresis even under conditions of degraded interfacial contact, reinforcing the conclusion that the hysteresis difference between T-FeF₃ and R-FeF₃ arises primarily from their inherent thermodynamic reaction pathways, rather than from kinetic limitations such as electron or ion transport resistance.

Comments

3. *The authors are encouraged to provide more details on the scalability of the synthesis process and the potential industrial applications of T-FeF₃. A clearer*

explanation of the limitations of the R-FeF₃ system in practical applications would help contextualize the advantages of T-FeF₃.

Author reply:

Thank you for your interest in the industrial potential of our findings. The primary advantage of our T-FeF₃ system for commercial applications lies in enhancing the energy efficiency of conversion-type cathode materials by mitigating voltage hysteresis, which is a crucial factor in practical for both high energy density and efficiency of the batteries. Unlike R-FeF₃, which undergoes irreversible phase transitions and significant structural evolution, T-FeF₃ can reversibly transition between insertion and conversion reactions due to its structural similarity to the discharged FeF₂ phase. As a result, the voltage hysteresis is reduced, enabling an increase in energy efficiency from 81% (for R-FeF₃) to approximately 87% for T-FeF₃.

Nevertheless, our current research focuses on a model system that employs nano-scale particles to validate the proposed reaction mechanism, leaving open challenges for implementing T-FeF₃ in large-scale, high energy density electrodes. Further investigations, such as devising chemical synthesis routes to produce T-FeF₃ independently, developing micron-scale particles and optimizing bulk synthesis methods, are needed to enable successful industrial applications of T-FeF₃.

Based on the reviewer's comments, the content of the text has been revised as follows:

Original text (Page 13-14)

Conclusion

We emphasize the potential of tetragonal FeF₃ (T-FeF₃) derived from LiF–FeF₂ nanocomposites to address the issues of compositional inhomogeneity and voltage hysteresis in iron fluoride cathode materials. LiF–FeF₂ nanocomposite successfully guided the phase transition into metastable T-FeF₃ while maintaining the structural framework of FeF₂. Due to the structural integrity, lower voltage hysteresis is achieved for T-FeF₃ under conversion reaction into FeF₂ with mitigated compositional inhomogeneity. This result starkly contrasts with that for R-FeF₃, which inevitably suffers from compositional inhomogeneity induced by irreversible phase transitions into FeF₂. As a result, although T-FeF₃ undergoes sequential insertion and conversion reactions, this material achieved excellent cycle stability, maintaining 72% and 74% of its capacity at 50 and 100 mA g⁻¹, respectively, over 300 cycles. Moreover, the superior reversibility of the T-FeF₃ phase recovery was further validated even after conversion into LiF and Fe metal phases if the seeds of the tetragonal phase remain. Our study suggests that harnessing the conversion reaction that maintains structural integrity can resolve the chronic issues of large voltage hysteresis and low structural reversibility for conversion reaction electrode materials. Furthermore, our approach of nanocomposite to design cathode materials could offer a new direction for developing rechargeable batteries with high energy density using conversion chemistry.

Revised text (Page 14)

Conclusion

We emphasize the potential of tetragonal FeF₃ (T-FeF₃) derived from LiF–FeF₂ nanocomposites to address the issues of compositional inhomogeneity and voltage hysteresis in iron fluoride cathode materials. LiF–FeF₂ nanocomposite successfully guided the phase transition into metastable T-FeF₃ while maintaining the structural framework of FeF₂. Due to the structural integrity, lower voltage hysteresis is achieved for T-FeF₃ under conversion reaction into FeF₂ with mitigated compositional inhomogeneity. This result starkly contrasts with that for R-FeF₃, which inevitably suffers from compositional inhomogeneity induced by irreversible phase transitions into FeF₂. As a result, although T-FeF₃ undergoes sequential insertion and conversion reactions, this material achieved excellent cycle stability, maintaining 72% and 74% of its capacity at 50 and 100 mA g⁻¹, respectively, over 300 cycles. In addition, its energy efficiency improves from 81% for R-FeF₃ to 87% for T-FeF₃. Moreover, the superior reversibility of the T-FeF₃ phase recovery was further validated even after conversion into LiF and Fe metal phases if the seeds of the tetragonal phase remain. Our study suggests that harnessing the conversion reaction that maintains structural integrity can resolve the chronic issues of large voltage hysteresis and low structural reversibility for conversion reaction electrode materials. Furthermore, our approach of nanocomposite to design cathode materials could offer a new direction as a model system for developing rechargeable batteries with high energy density using conversion chemistry. Hereafter, further investigations, such as developing micron-scale particles and optimizing bulk synthesis methods, are needed to enable the practical application of designed materials.

Comments

4. *The figures are well-designed, but the captions could be more detailed to aid standalone interpretation.*

Author reply:

We are appreciated for the reviewer's meticulous comments. The captions were revised as follows based on the reviewer's comments:

Original Figure

Fig. 1: Formation of T-FeF₃ phase guided by LiF-FeF₂ nanocomposite. **a**, Rietveld refinement of the X-ray diffraction (XRD) data ($\lambda = 1.5406 \text{ \AA}$) of the LiF-FeF₂. **b**, High-resolution transmission electron microscope (TEM) image of LiF-FeF₂ in a pristine state. The inset shows the fast Fourier transform (FFT) pattern of LiF-FeF₂. **c**, Electrochemical profile of LiF-FeF₂ nanocomposite at 25 °C and 20 mA g⁻¹ current density. The right is the differential analysis of the voltage profile. **d**, Ex-situ XRD patterns of LiF-FeF₂ electrodes at charged/discharged states after the 1st, 5th, and 10th cycles. **e**, Crystal structures of FeF₂, T-FeF₃ (determined through X-ray diffraction of the 10th charge state), and R-FeF₃. Brown and silver balls indicate Fe and F ions, respectively. **f**, Fourier transformed magnitude (black), imaginary part (blue), and best fit (red) using the T-FeF₃ model for the charged electrode.

Revised Figure

Fig. 1: Formation of T-FeF₃ phase guided by LiF-FeF₂ nanocomposite. **a**, Rietveld refinement of the X-ray diffraction (XRD) data ($\lambda = 1.5406 \text{ \AA}$) of the LiF-FeF₂. **b**, High-resolution transmission electron microscope (TEM) image of LiF-FeF₂ in a pristine state. Each domain is outlined with a dotted line. The inset shows the fast fourier transform (FFT) pattern of LiF-FeF₂. White and green represent FeF₂ and LiF, respectively. **c**, Electrochemical profile of LiF-FeF₂ nanocomposite at 25 °C and 20 mA g⁻¹ current density. Blue depicts the evolving voltage profile of LiF-FeF₂ up to the 10th cycle, while yellow represents the 10th cycle profile of R-FeF₃. The right is the differential analysis of the voltage profile. **d**, Ex-situ XRD patterns of LiF-FeF₂ electrodes at charged/discharged states after the 1st, 5th, and 10th cycles. Red and blue are discharge and charge states, respectively. **e**, Crystal structures of FeF₂, T-FeF₃ (determined through X-ray diffraction of the 10th charge state), and R-FeF₃. Brown and silver balls indicate Fe and F ions, respectively. **f**, Fourier transformed magnitude (black), imaginary part (blue), and best fit (red) using the T-FeF₃

model for the charged electrode.

Original Figure

Fig. 2: Intercalation and conversion reaction of T-FeF₃. **a**, *Ex-situ* XRD patterns of T-FeF₃ nanocomposite at different lithiation states in the WV range. **b**, Voltage profile depending on lithiation state in the WV range. **c**, Phase fraction at different lithiation states determined by XRD Rietveld refinement. **d**, The formation energy of T-FeF₃ and R-FeF₃ at different states of lithiation. **e**, Experimentally measured voltage profile and DFT calculated reaction voltage for T-FeF₃ at different states of lithiation.

Revised Figure

Fig. 2: Intercalation and conversion reaction of T-FeF₃. **a**, *Ex-situ* XRD patterns of T-FeF₃ nanocomposite at different lithiation states in the wide voltage range (WV, 4.8V-2.0V). **b**, Voltage profile depending on lithiation state in the WV range. **c**, Phase fraction at different lithiation states determined by XRD Rietveld refinement. **d**, The formation energy of T-FeF₃ and R-FeF₃ at different states of lithiation. **e**, Experimentally measured voltage profile and DFT calculated reaction voltage for T-FeF₃ at different states of lithiation.

Original Figure

Fig. 3: Mitigated voltage hysteresis and compositional inhomogeneity of T-FeF₃ compared to R-FeF₃. **a**, Galvanostatic intermittent titration technique (GITT) profiles of T-FeF₃ and R-FeF₃. The cells were allowed to relax for 3 h after every 11.2 mAh g⁻¹ (corresponding to 0.05 e⁻/formula unit) discharging/charging at 20 mA g⁻¹. **b**, Voltage difference ($V_{\text{gap}} = V_{\text{relax, charge}} - V_{\text{relax, discharge}}$) between charge and discharge steps after the 3 h relaxation at the same state of lithiation of T-FeF₃ and R-FeF₃. **c**, Voltage changes after the 3 h relaxation at different states of discharge and charge at T-FeF₃ and R-FeF₃. **d-e, g-h**, a scanning TEM (STEM)- electron energy loss spectroscopy (EELS) images of T-FeF₃ and R-FeF₃ in charged state and discharged state for the energy distribution of the Fe L₃-edge peak. The charge state of T-FeF₃ (TC) and R-FeF₃ (RC). The discharge state of T-FeF₃ (TD) and R-FeF₃ (RD). **f,i**, In charge state and discharge state, the left is EELS spectra of Fe L_{3,2}-edge, and the right is Fe L₃-edge peak energy and the L₃/L₂ ratio.

Revised Figure

Original Figure

Fig. 4: Electrochemical performance of T-FeF₃. **a**, Cycle stability at 50 mA g⁻¹ of T-FeF₃ and FeF₃. **b,c**, Electrochemical profile of T-FeF₃ and R-FeF₃ at various cycles. **d**, Rate performance of LiF-FeF₂ and FeF₃. **e**, Comparison of capacity retention of T-FeF₃ and iron fluoride materials mixed with carbon.

Revised Figure

Fig. 4: Electrochemical performance of T- FeF_3 . **a**, Cycle stability at 50 mA g^{-1} of T- FeF_3 and FeF_3 . **b,c**, Electrochemical profile of T- FeF_3 and R- FeF_3 at various cycles. **d**, Rate performance of LiF- FeF_2 and FeF_3 . **e**, Comparison of capacity retention of T- FeF_3 and iron fluoride materials mixed with carbon. The electrochemical stability of iron fluoride materials was evaluated in the 1-electron transfer range (Discharge cutoff voltage $\sim 2\text{V}$).

Original Figure

Response to reviewers

Guided phase transition for mitigating voltage hysteresis of iron fluoride cathode materials in lithium-ion batteries

Hyoj Jo,^{‡1} Minjeong Gong^{‡2}, Se Young Kim³, Dong-Hwa Seo^{*2} & Sung-Kyun Jung^{*1,4,5,6}

Affiliation

¹Institute for Battery Research Innovation, Seoul National University, Seoul, Republic of Korea.

²Department of Materials Science and Engineering, Korea Advanced Institute of Science and Technology (KAIST), Daejeon, Republic of Korea.

³Energy Storage Research Center, Korea Institute of Science and Technology (KIST), Seoul, Republic of Korea.

⁴Department of Materials Science and Engineering, College of Engineering, Seoul National University, Seoul, Republic of Korea.

⁵School of Transdisciplinary Innovations, Seoul National University, Seoul, Republic of Korea.

⁶Research Institute of Advanced Materials, Seoul National University, Seoul, Republic of Korea.

‡ These authors contributed equally: Hyoj Jo, Minjeong Gong

* These authors jointly supervised this work: Dong-Hwa Seo, Sung-Kyun Jung (email: dseo@kaist.ac.kr and naecard@snu.ac.kr)

Reviewer #1(Remarks to the Author):

Comments to the Author

I have carefully reviewed the responses regarding the issues. The authors believe that the “guided phase transition strategy” addresses voltage hysteresis and structural reversibility, which are critical challenges in iron fluoride cathodes. However, according to the response to comment 6, this proposed strategy only exhibits favorable effects on the intercalation reaction of iron fluoride cathodes, while demonstrating limited effectiveness for conversion reactions. Intercalation-type iron fluoride cathodes have been extensively studied (e.g., *Adv. Mater.*, 2019, 31, 1905146; *Nano energy*, 2023, 108, 108181) and have enabled high-rate (up to 100C) and long-life (up to 1000 cycles with a high capacity retention of 84.1%) single-electron transfer reactions, as well as high-loading cathodes (5.3 mg/cm²). Therefore, the electrochemical performance achieved in this study is deemed unsatisfactory.

On the other hand, voltage hysteresis and structural reversibility have not been the key challenges for iron fluorides limited to the intercalation reaction stage, but rather for their conversion reactions to deliver high-capacity and high-energy. From a long-term perspective, activating the conversion reaction could enhance the competitiveness of iron fluorides compared with currently commercial cathodes, especially in terms of capacity and energy density. Unfortunately, the proposed strategy showed a limited effect on the critical conversion reactions. Overall, this work provides a supplement to the study on the intercalation reaction of the T-FeF₃ cathode, but fails to make impressive progress in the development of iron fluoride cathodes. Therefore, I still cannot recommend this manuscript for publication in *Nature Communications*. Additionally, several issues remain unresolved. Please refer to the comments below.

Author reply:

We thank Reviewer 1 for the careful re-examination of our study and for highlighting an important question: Does our “guided phase-transition” strategy offer benefits only within the “single-electron transfer” reaction corresponding to “intercalation” regime, leaving the deeper conversion reactions—where capacity and energy density truly increase—largely unaddressed? We agree that demonstrating relevance to conversion chemistry is essential for long-term impact. To resolve the reviewer's concern, our key findings are summarized below. In addition, we performed additional experiments and analyses to address the reviewer’s comments.

(1) Conversion already occurs inside the so-called intercalation range.

First of all, both polymorphs (T-FeF₃ and R-FeF₃) undergo a conversion reaction ($\text{FeF}_3 \rightarrow \text{LiF} + \text{FeF}_2$) even within the “single-electron transfer” regime (Our study and *Nat. Mater.* **20**, 841-850 (2021)). Therefore, our “guided phase transition” strategy is not limited to “intercalation” reaction but includes the “conversion” reaction even within single-electron

transfer reaction.

(2) Seed-guided reversible structural recovery even after deeper conversion to LiF and Fe metal phases.

Second, the “guided phase transition” strategy is also beneficial for deeper conversion reactions forming LiF and Fe metal phase. Notably, if a small fraction of T-FeF₃ exist even after undergoing conversion reaction including LiF and Fe metal phase formation, it can act as structural seeds that guide the reaction course, leading to reversible phase recovery to T-FeF₃ (Fig. 5a-d). This guided behavior results in markedly improved cycling stability, which is in contrast to conventional R-FeF₃ that fails to achieve reversible structural recovery (Supplementary Fig. S38). Altogether, these findings strongly support that a phase-guided framework can effectively steer reaction pathways, providing a rational design strategy that may be further extended to enhance the reversibility of more complete conversion reactions.

Fig. 5. a-c, The cycle ability and differential analysis of voltage profile of T-FeF₃ and R-FeF₃ with repeated changing discharge cut-off voltage (2-1.5-2V). **d**, The XRD pattern of charged states for T-FeF₃(T) and R-FeF₃(R) at each point in the cycle measured at the changing cut-off voltage. Point 1 (10th cycle), point 2 (20th cycle), and point 3 (40th cycle).

Supplementary Fig. S38. Comparison of specific capacities of T-FeF₃ and R-FeF₃ with repeated changes in depth of discharge.

(3) Complementarity to kinetics-oriented previous approaches

Third, we fully acknowledge the impressive electrochemical performance *via* alleviated kinetic limitations (references from reviewer: e.g., *Adv. Mater.* **31**, 1905146 (2019); *Nano energy*, **108**, 108181 (2023)). Our work targets a different bottleneck “how reaction pathway engineering (T-FeF₃ vs. R-FeF₃) suppresses compositional inhomogeneity and intrinsic voltage hysteresis, thereby enhancing reversibility?”. Additional long-relaxation GITT (48 h) confirm that hysteresis suppression also can arise from reaction-route control (Figure R2 below), not merely faster kinetics. It therefore complements, rather than merely increments, prior kinetics-focused research. Therefore, rather than directly competing with this electrochemical performance, this study provides insights into improving structural reversibility. Thus, the two strategies are complementary and could be combined in future designs.

Together, these results constitute, to our knowledge, the first demonstration that both partial and full conversion reactions in iron-fluoride systems can be rendered structurally reversible through polymorph-guided reaction pathway control. We hope this additional evidence addresses the reviewer’s concern and underscores the broader relevance of our approach for high-energy, low-hysteresis conversion cathodes.

Comments

1. Regarding comment 1, I hope the authors can demonstrate the differences in chemical distribution and electronic/ionic contact between the LiF-FeF₂ composites formed by mechanical ball milling and electrochemical reaction through solid physical characterization.

Author reply:

We thank the reviewer for this valuable comment. In response, we have conducted additional EELS (Electron Energy Loss Spectroscopy) mapping analyses to directly compare the chemical distribution and potential contact quality between LiF and Fe species in the two systems of interest: the mechanically synthesized LiF-FeF₂ nanocomposite (pristine state) and the electrochemically discharged R-FeF₃ electrode.

Fig. R1. (a) Scanning transmission electron microscopy (STEM)–electron energy loss spectroscopy (EELS) maps of the pristine state of the LiF-FeF₂ composite, showing the spatial distribution of the Fe L_{3} -edge and LiF-related F K -edge signals. The Fe L_{3} -edge peak is color-coded to represent oxidation states, with blue indicating the most oxidized regions and pink indicating the most reduced regions. The LiF-related F K -edge signal is shown in green. (b) EELS spectra of the F K -edge and Fe $L_{3,2}$ -edge collected from two regions of Fig. R1a. Region (1), with the highest LiF signal intensity, and region (2), with the lowest LiF signal intensity.

In the pristine LiF-FeF₂ nanocomposite, the LiF-related F K -edge signal (*J. Mater. Chem. A*, **1**, 11629-11640 (2013); *Nat. Commun.* **13**, 6070 (2022); *J. Appl. Electrochem.* **41**, 1295-1299 (2011)) was uniformly distributed and spatially well-connected with Fe²⁺ signals (Fig. R1a and b). This suggests an intimate and homogeneous nanoscale contact between LiF and FeF₂ domains formed through mechanical ball milling. Such contact is favorable for ensuring efficient electronic and ionic percolation throughout the electrode.

Fig. R1. (c) Scanning transmission electron microscopy (STEM)–electron energy loss spectroscopy (EELS) maps of the discharge state of the R-FeF₃, showing the spatial distribution of the Fe *L*₃-edge and LiF-related F *K*-edge signals. The Fe *L*₃-edge peak is color-coded to represent oxidation states, with blue indicating the most oxidized regions and pink indicating the most reduced regions. The LiF-related F *K*-edge signal is shown in green. **(d)** EELS spectra of the F *K*-edge and Fe *L*_{3,2}-edge collected from two regions of Fig. R1a. Region (1), with the highest LiF signal intensity, and region (2), with the lowest LiF signal intensity.

In contrast, for the electrochemically discharged R-FeF₃, the LiF signal was sparse and confined to localized regions (Fig. R1c and d). The formation of T-FeF₃ from LiF requires direct contact with FeF₂. Therefore, a limited presence of LiF may lead to either the formation of an undetectably small amount of T-FeF₃ or the generation of isolated LiF clusters that are not properly connected to FeF₂.

Supplementary Fig. S21. (c) *ex-situ* XRD patterns for the 10th, 50th, and 100th charge states of R-FeF₃.

This interpretation is supported by *ex-situ* XRD analysis of R-FeF₃ at various stages of cycling. As shown in Fig. R2, the LiF and FeF₂ phases initially generated from R-FeF₃ are minimal in quantity and remain spatially and interfacially separated, rendering them electrochemically inactive. However, upon repeated cycling, the accumulation of these inactive phases leads to a gradual formation of the tetragonal phase, likely due to partial alleviation of their spatial isolation (Supplementary Fig. S21c).

Fig. 1d, *Ex-situ* XRD patterns of LiF-FeF₂ electrodes at charged/discharged states after the 1st, 5th, and 10th cycles.

In contrast, T-FeF₃ undergoes a more complete conversion to LiF and FeF₂ during discharge, enabling more effective reversion upon charging due to improved structural connectivity

(Fig. 1d).

This distinct difference in the spatial distribution of LiF shows how mechanical synthesis can provide better chemical integration and interfacial contact, whereas electrochemical conversion tends to produce more disconnected, electronically and ionically isolated composites. These differences affect not only the structural evolution of T-FeF₃ originating from LiF-FeF₂ nanocomposites and R-FeF₃, but also play a critical role in determining its electrochemical reversibility.

Based on the reviewer's suggestion, we have added a figure and revised the manuscript to incorporate these additional findings and discussions.

Original text (Page 13)

By comparing the reaction mechanism between R-FeF₃^[21] and T-FeF₃, both materials commonly show intercalation and conversion reactions (Fig. 6b). Although the product of the conversion reaction (until 2 V) is the same as FeF₂ and LiF in both cases (Fig. 2a and Supplementary Fig. S40), the reversibility is much better for T-FeF₃ than R-FeF₃ due to the structural similarity with FeF₂. This difference is further validated by STEM–EELS analysis performed after 100 cycles. Since it was measured at 100 mA g⁻¹, both T-FeF₃ and R-FeF₃ show a mixed Fe²⁺/Fe³⁺ state in the charged state. However, R-FeF₃ displays a significantly greater presence of reduced Fe²⁺ regions and lower Fe³⁺ intensity even in the most oxidized areas, indicating more pronounced compositional inhomogeneity compared to T-FeF₃ (Supplementary Fig. S41). R-FeF₃ undergoes a structural change from a corner-sharing FeF₆ group structure (rcp) to an edge-sharing tetragonal phase (tcp) during charging and discharge^[19,21,56,60] It is understood that the significant structural difference between the two structures causes an irreversible phase transition, which leads to compositional inhomogeneity²¹ (Supplementary Fig. S25 and Supplementary Note 9)

Revised text (Page 13)

By comparing the reaction mechanism between R-FeF₃^[21] and T-FeF₃, both materials commonly show intercalation and conversion reactions (Fig. 6b). Although the final products of the conversion reaction (down to 2 V) are the same in both cases (Fig. 2a and Supplementary Fig. S41), the FeF₂ and LiF converted from R-FeF₃ are formed in small amounts in isolated regions, resulting in poor interfacial contact (Supplementary Fig. S42). Moreover, from a structural standpoint, T-FeF₃ exhibits much higher reversibility than R-FeF₃ due to its closer structural similarity to FeF₂. This difference is further validated by STEM–EELS analysis performed after 100 cycles. Since it was measured at 100 mA g⁻¹, both T-FeF₃ and R-FeF₃ show a mixed Fe²⁺/Fe³⁺ state in the charged state. However, R-FeF₃ displays a significantly greater presence of reduced Fe²⁺ regions and lower Fe³⁺ intensity even in the most oxidized areas, indicating more pronounced compositional inhomogeneity compared to T-FeF₃ (Supplementary Fig. S43). R-FeF₃ undergoes a structural change from

a corner-sharing FeF_6 group structure (rcp) to an edge-sharing tetragonal phase (tcp) during charging and discharge^[19,21,56,60] It is understood that the significant structural difference between the two structures causes an irreversible phase transition, which leads to compositional inhomogeneity²¹ (Supplementary Fig. S25 and Supplementary Note 9)

Added supplementary Figure

Supplementary Fig. S42. Scanning transmission electron microscopy (STEM)–electron energy loss spectroscopy (EELS) maps of (a) LiF-FeF₂ pristine state and (b) the discharged R-FeF₃, showing the spatial distribution of the Fe $L_{3,2}$ -edge and LiF-related F K -edge signals. The Fe $L_{3,2}$ -edge peak is color-coded to represent oxidation states, with blue indicating the most oxidized regions and pink indicating the most reduced regions. The LiF-related F K -edge signal is shown in green. (b) EELS spectra of the F K -edge and Fe $L_{3,2}$ -edge collected from two regions of Fig. R1a. Region (1), with the highest LiF signal intensity, and region (2), with the lowest LiF signal intensity.

Comments

2. Regarding comment 4, voltage hysteresis within a reversible cycle is inherently a kinetic phenomenon. This reference (*Nat. Chem.*, 2021, 13, 1070.) may help the authors to understand the origin of voltage hysteresis observed during GITT testing.

Author reply:

We sincerely thank the reviewer for the thoughtful comment and for suggesting the relevant reference (*Nat. Chem.*, **13**, 1070 (2021)), which provides valuable insight into the kinetic origin of voltage hysteresis. By referring to the reference and the reviewer’s earlier comment 4, we have refined our interpretation. In our initial manuscript, the phrase “thermodynamic voltage

hysteresis” was meant to denote the voltage gap after relaxation that appears when the reaction pathway itself changes—specifically, the largely irreversible route $\text{R-FeF}_3 \rightarrow \text{LiF} + \text{FeF}_2$ versus the reversible route $\text{T-FeF}_3 \rightarrow \text{LiF} + \text{FeF}_2$. As the reference (*Nat. Chem.*, **13**, 1070 (2021)) makes clear, such hysteresis originates from “slow kinetics” of phase transitions and bond breaking/re-formation, not from the rapid transport-controlled overpotentials (ohmic drop, ion/electron diffusion) that equilibrate within several hours (typically regarded as “kinetic hysteresis”).

Fig. R2a. GITT profiles of T-FeF₃ and R-FeF₃ after the 10th cycle, along with the absolute value of the voltage change rate (dV/dt) in each relaxation region. Cells were allowed to relax for 3 h after every 11.2 mAh g⁻¹ (corresponding to 0.05 e⁻ per formula unit) of charge or discharge at a current density of 20 mA g⁻¹. The time-dependent voltage profile is shown in the inset, with the voltage variation indicated by the dotted line.

To separate these contributions, we first enough lengthened every GITT rest step until the decay rate fell below 0.01 mV s⁻¹ which is generally used in the literature to approximate the quasi-equilibrium voltage (*Nat. Commun.* **11**, 1252 (2020); *J. Electrochem. Soc.* **166**, A3980 (2019); *J. Electrochem. Soc.* **162**, A1374 (2015); *Nat. Commun.* **10**, 585 (2019); *J. Electrochem. Soc.* **144**, 3886 (1997)). Under these near-equilibrium conditions, the relaxed potentials of T-FeF₃ and R-FeF₃ converge, confirming that transport overpotential is essentially identical in both electrodes (Fig.R2a)

Fig. R2. Long-term (48 h) relaxation tests of **(b)** T-FeF₃ and **(c)** R-FeF₃ at 20 mA g⁻¹ after the 10th cycle. The time-dependent voltage profile is shown in the inset. **(d)** Voltage difference ($V_{\text{gap}} = V_{\text{relax, charge}} - V_{\text{relax, discharge}}$) between charge and discharge steps after 48 h relaxation.

We then carried out ultra-long relaxations of 48 h to measure the “kinetic hysteresis” from phase transition and bond breaking/reformation (Figs. R2b and c). The additional rest sharply reduced the remaining gap, showing that the dominant term now arises from the slow, pathway-dependent kinetics described above. Crucially, even after 48 h a finite gap persists and T-FeF₃ still exhibits markedly lower hysteresis than R-FeF₃; because R-FeF₃ forms irreversible LiF and FeF₂ domains, this residual gap cannot be attributed to kinetics alone (Fig. R2d).

In light of these results, we have replaced “*thermodynamic hysteresis*” with the more precise expression “*reaction pathway-dependent kinetic hysteresis arising from phase-transition and bond-breaking barriers.*” This wording clarifies that the voltage gap is kinetic in origin yet strongly controlled by the chosen reaction pathway, thereby aligning our discussion with the mechanistic framework outlined in *Nat. Chem.*, **13**, 1070 (2021). In addition, the specific origin of this kinetic hysteresis may be due to the spatial distribution of LiF (compositional inhomogeneity) that is replied for comment 1.

We greatly appreciate the reviewer’s guidance, which has allowed us to refine the clarity and precision of our interpretation.

Original text (Page 3)

Therefore, strategies such as designing the composites with conductive materials^[26–30] or reducing the particle size to the nano-size level^[19,21,26,31] have been proposed to overcome the limited reaction kinetics due to the low electronic conductivity and sluggish mass transport from long-range diffusion. Despite these efforts, the compositional inhomogeneity and voltage hysteresis have not been completely resolved, which implies that they may also have thermodynamic origins.

Revised text (Page 3)

Therefore, strategies such as designing the composites with conductive materials^{26–30} or reducing the particle size to the nano-size level^{19,21,26,31} have been proposed to overcome the limited reaction kinetics due to the low electronic conductivity and sluggish mass transport from long-range diffusion. Despite these efforts, the compositional inhomogeneity and voltage hysteresis have not been completely resolved, which implies that they may also have **reaction pathway** origins.

Original text (Page 4)

To mitigate the compositional inhomogeneity from a thermodynamic perspective, it is essential to evade phase-displacement reactions accompanying long-range diffusion. In this respect, nanocomposite cathodes composed of lithium compounds and transition-metal compounds have successfully guided reversible reaction routes with minimal diffusion while maintaining the mother structure or anion framework of transition-metal compounds.

Revised text (Page 4)

To mitigate the compositional inhomogeneity from a **reaction pathway** perspective, it is essential to evade phase-displacement reactions accompanying long-range diffusion. In this respect, nanocomposite cathodes composed of lithium compounds and transition-metal compounds have successfully guided reversible reaction routes with minimal diffusion while maintaining the mother structure or anion framework of transition-metal compounds.

Original text (Page 9)

To evaluate the effect of maintaining the structural integrity during intercalation and conversion reaction for T-FeF₃ on voltage hysteresis and compositional inhomogeneity, we first compared the voltage hysteresis in T-FeF₃ and R-FeF₃ with galvanostatic intermittent titration technique (GITT) analysis (Fig. 3a). All the analyses were performed after 10 cycles to ensure the evolution of T-FeF₃ from LiF-FeF₂. Fig. 3b shows the difference in relaxed voltage during charge and discharge corresponding to thermodynamic hysteresis, and Fig. 3c presents the extent of voltage change during relaxation corresponding to kinetic hysteresis. T-FeF₃ exhibited a smaller thermodynamic hysteresis compared to R-FeF₃, which is prominent at the end of charge or discharge. Both T-FeF₃ and R-FeF₃ exhibit larger voltage hysteresis during the charging process than during discharge. For T-FeF₃, this is mainly attributed to LiF splitting that occurs during charging (Supplementary Note 7), while in the case of R-FeF₃, the increased hysteresis likely results from phase transitions involving long-range diffusion. However, this difference in kinetic hysteresis between T-FeF₃ and R-FeF₃ is relatively insignificant during both charge and discharge states. This is due to the improved reaction rate and mass transfer in both cases using carbon composites with nano-sized particles. This finding implies that the alleviation of voltage hysteresis in T-FeF₃ compared

to R-FeF₃ is due not to kinetics but to thermodynamic origins^[21] (Supplementary Fig. S12 and Supplementary Note 5).

Revised text (Page 9)

To evaluate the effect of maintaining the structural integrity during intercalation and conversion reaction for T-FeF₃ on voltage hysteresis and compositional inhomogeneity, we first compared the voltage hysteresis in T-FeF₃ and R-FeF₃ with galvanostatic intermittent titration technique (GITT) analysis (Fig. 3a). Charge/discharge measurements were performed at 11.2 mAh g⁻¹ (corresponding to 0.05 e⁻ per formula unit) with a current of 20 mA g⁻¹, and each relaxation step was maintained for 3 hours until the voltage decay rate (dV/dt) dropped below ~0.01 mV s⁻¹, a criterion commonly used to approximate quasi-equilibrium (Supplementary Fig. S26a). All the analyses were performed after 10 cycles to ensure the evolution of T-FeF₃ from LiF-FeF₂. Fig. 3b shows the voltage gap between the relaxed potentials during charge and discharge, which corresponds to reaction pathway-dependent kinetic hysteresis arising from phase-transition and bond-breaking barriers⁵⁵. Fig. 3c presents the extent of voltage change during relaxation, which reflects conventional kinetic polarization related to ion/electron transport. The reaction pathway-dependent kinetic hysteresis was smaller for T-FeF₃ than for R-FeF₃, and this trend remained consistent even after extended relaxation for 48 hours (Supplementary Fig. S26b–d), suggesting that the major voltage gap originates not from transient transport polarization but from slow structural transformations such as phase transition and bond breaking/reformation. This voltage difference is prominent at the end of charge or discharge. Both T-FeF₃ and R-FeF₃ exhibit larger voltage hysteresis during the charging process than during discharge. For T-FeF₃, this is mainly attributed to LiF splitting that occurs during charging (Supplementary Note 7), while in the case of R-FeF₃, the increased hysteresis likely results from phase transitions involving long-range diffusion. However, this difference in kinetic hysteresis between T-FeF₃ and R-FeF₃ is relatively insignificant during both charge and discharge states. This is due to the improved reaction rate and mass transfer in both cases using carbon composites with nano-sized particles. Taken together, despite similar particle size and carbon content, these results indicate that the reduced hysteresis in T-FeF₃ compared to R-FeF₃ stems from differences in reaction pathways and the reversibility of phase transitions rather than from extrinsic kinetic limitations such as transport resistance²¹ (Supplementary Fig. S12 and Supplementary Note 5).

Original Supplementary Note 5 (Page 13-14)

To further confirm the importance of interfacial contact, we investigated the effect of crystallinity by systematically varying the ball-milling time. By adjusting only the initial LiF-FeF₂ mixing duration (48 h, 12 h, and 0 h) while maintaining the carbon mixing step constant at 12 h (500 rpm), we were able to isolate the impact of interfacial contact and crystallinity on electrochemical behavior. XRD analysis revealed that reducing the milling

time resulted in narrower diffraction peaks and a noticeable decrease in the full width at half maximum (FWHM), indicating increased crystallinity and larger particle sizes (Supplementary Fig. S12a and b). This microstructural change reduced the interfacial contact area between LiF and FeF₂, which is essential for promoting the phase transition to T-FeF₃. Electrochemical testing further confirmed that insufficient LiF-FeF₂ contact deteriorates capacity but has limited impact on voltage hysteresis behavior. As milling time decreased, charge–discharge profiles (Supplementary Fig. S11c) showed reduced capacity due to poorer interfacial contact. This capacity degradation is attributed to the limited formation of electrochemically active T-FeF₃, resulting in compositional inhomogeneity. However, GITT (Galvanostatic Intermittent Titration Technique) analysis revealed that the difference in relaxed voltages between charge and discharge (V_{gap}), representing thermodynamic hysteresis, remained nearly constant across all milling times and significantly smaller than that of R-FeF₃ (Supplementary Fig. S12d–e). The rest potential change (ΔV_{rest}), which reflects kinetic polarization, slightly increased at the end of charge as milling time decreased. Nevertheless, even in the shortest milling condition (0 h), ΔV_{rest} was still considerably smaller than in R-FeF₃ (Supplementary Fig. S12f), indicating that kinetic effects such as electron/ion diffusion barriers were minimal. This is likely because our nanocomposite system consists of sufficiently nanosized particles and a highly conductive carbon matrix, minimizing the impact of transport-related kinetic limitations.

In summary, the interfacial contact between LiF and FeF₂ plays a crucial role in the formation and electrochemical activation of T-FeF₃. In our nanocomposite system consisting of sufficiently small particles and an abundant conductive carbon matrix, changes in interfacial contact significantly affect capacity but have minimal impact on voltage hysteresis. Notably, T-FeF₃ exhibits consistently low hysteresis even under conditions of degraded interfacial contact, reinforcing the conclusion that the hysteresis difference between T-FeF₃ and R-FeF₃ arises primarily from their inherent thermodynamic reaction pathways, rather than from kinetic limitations such as electron or ion transport resistance.

Revised Supplementary Note 5 (Page 13-14)

To further confirm the importance of interfacial contact, we investigated the effect of crystallinity by systematically varying the ball-milling time. By adjusting only the initial LiF-FeF₂ mixing duration (48 h, 12 h, and 0 h) while maintaining the carbon mixing step constant at 12 h (500 rpm), we were able to isolate the impact of interfacial contact and crystallinity on electrochemical behavior. XRD analysis revealed that reducing the milling time resulted in narrower diffraction peaks and a noticeable decrease in the full width at half maximum (FWHM), indicating increased crystallinity and larger particle sizes (Supplementary Fig. S12a and b). This microstructural change reduced the interfacial contact area between LiF and FeF₂, which is essential for promoting the phase transition to T-FeF₃. Electrochemical testing further confirmed that insufficient LiF-FeF₂ contact deteriorates capacity but has limited impact on voltage hysteresis behavior. As milling time decreased, charge–discharge profiles (Supplementary Fig. S11c) showed reduced capacity due to poorer interfacial contact. This capacity degradation is attributed to the limited formation of

electrochemically active T-FeF₃, resulting in compositional inhomogeneity. However, GITT (Galvanostatic Intermittent Titration Technique) analysis revealed that the difference in relaxed voltages between charge and discharge (V_{gap}), representing reaction pathway-dependent kinetic hysteresis arising from phase-transition and bond-breaking barriers, remained nearly constant across all milling times and significantly smaller than that of R-FeF₃ (Supplementary Fig. S12d–e). The rest potential change (ΔV_{rest}), which reflects kinetic polarization, slightly increased at the end of charge as milling time decreased. Nevertheless, even in the shortest milling condition (0 h), ΔV_{rest} was still considerably smaller than in R-FeF₃ (Supplementary Fig. S12f), indicating that kinetic effects such as electron/ion diffusion barriers were minimal. This is likely because our nanocomposite system consists of sufficiently nanosized particles and a highly conductive carbon matrix, minimizing the impact of transport-related kinetic limitations.

In summary, the interfacial contact between LiF and FeF₂ plays a crucial role in the formation and electrochemical activation of T-FeF₃. In our nanocomposite system consisting of sufficiently small particles and an abundant conductive carbon matrix, changes in interfacial contact significantly affect capacity but have minimal impact on voltage hysteresis. Notably, T-FeF₃ exhibits consistently low hysteresis even under conditions of degraded interfacial contact, reinforcing the conclusion that the hysteresis difference between T-FeF₃ and R-FeF₃ arises primarily from their inherent reaction pathway-governed kinetic behavior, associated with differences in phase transition reversibility and bond-breaking energetics, rather than from kinetic limitations such as electron or ion transport resistance.

Added reference

55. Li, B. *et al.* Correlating ligand-to-metal charge transfer with voltage hysteresis in a Li-rich rock-salt compound exhibiting anionic redox. *Nat. Chem.* **13**, 1070–1080 (2021).

Added supplementary Figure

Comments

3. Regarding comment 5, a distinct additional oxidation behavior can be observed around 4.8 V in the differential profile of the charge curve for R-FeF₃ (Fig. 5c). This is likely related to the oxidative decomposition of the electrolyte under high-voltage application, rather than the LiF separation proposed by the authors. This overcharge phenomenon becomes more severe when conversion reactions are involved for both T-FeF₃ and R-FeF₃ (Fig. R3), leading to their low coulombic efficiency. The high charge cutoff voltage generally induces undesired interfacial side reactions and deeper structural changes, thereby reducing the cycling life and coulombic/energy efficiency. Does the formation of T-FeF₃ require such a high charge cutoff voltage of 4.8 V?

Author reply:

We greatly appreciate the reviewer's valuable comments regarding the implications of applying a high charge cut-off voltage. The concern about the side reaction of electrolyte under high voltage is highly relevant, and we welcome the opportunity to clarify the rationale for selecting 4.8 V as the cut-off voltage in our system.

Our decision to use 4.8 V stems from the electrochemical requirements of the LiF-FeF₂ composite system. The transformation of this material into the desired T-FeF₃ phase critically depends on the decomposition of LiF, which is known for its high thermodynamic stability. Lower voltages are insufficient to overcome this barrier, and as such, fail to drive meaningful conversion. To clarify this, we conducted comparative cycling tests at both 4.5 V and 4.8 V cut-offs for electrodes based on T-FeF₃ and R-FeF₃.

Supplementary Fig. S18. (a, b) Electrochemical profiles of T-FeF₃ and R-FeF₃ at the 10th cycle measured at 50 mA g⁻¹ under different charge cut-off voltages (4.5 V and 4.8 V).

Fig. 6b. Comparison of reaction mechanisms between R-FeF₃ and T-FeF₃.

To assess whether the 4.8 V charge cut-off is indeed essential for activating the intended phase transition, we carried out a series of electrochemical tests comparing cut-off voltages of 4.5 V and 4.8 V for both T-FeF₃ and R-FeF₃ electrodes. The T-FeF₃ electrode subjected to 4.8 V charging exhibited a well-defined voltage plateau near 4.0 V during discharge, which is characteristic of the T-FeF₃ phase (Supplementary Fig. S18a and S18b). However, when the cut-off voltage was restricted to 4.5 V, this plateau became notably suppressed, and the associated capacity decreased substantially. This indicates that full formation of the T-FeF₃ phase is hindered under lower voltage conditions. In contrast, R-FeF₃ electrodes displayed comparatively marginal sensitivity to the change in cut-off voltage, maintaining similar capacity profiles under both voltage settings. The distinct different trends in capacity expression between the two materials under different cut-off voltages stem from their fundamentally different electrochemical mechanisms. Unlike R-FeF₃, which primarily undergoes insertion, T-FeF₃ proceeds through a combination of insertion followed by conversion (to LiF and FeF₂) (Fig. 6b). As a result, recharging only up to 4.5 V limits the extent of LiF splitting, thereby suppressing full capacity achievement (Supplementary Fig. S18a). Therefore, achieving sufficient formation of T-FeF₃ necessitates effective LiF dissociation, making a 4.8 V cut-off voltage essential (*Nat. Energy*, **2**, 16208 (2017); *Adv. Funct. Mater.* **31**, 2009133 (2021); *J. Power Sources*, **329**, 406 (2016); *J. Power Sources*, **354**, 34 (2017); *ECS Meet. Abstr.* **58**, 87 (2014)).

Supplementary Fig. S18c. Cycling stability of T-FeF₃ and R-FeF₃ at 50 mA g⁻¹ under different charge cut-off voltages (4.5 V and 4.8 V).

Notably, even at a lower cut-off voltage of 4.5 V, which helps minimize electrolyte decomposition, T-FeF₃ still demonstrated superior capacity retention compared to R-FeF₃ (Supplementary Fig. S18c). This finding highlights not only the more reversible electrochemical behavior of T-FeF₃, but also the inherently irreversible nature of R-FeF₃.

Therefore, the comparison between the 4.5 V and 4.8 V charge cut-off voltages not only

reinforces the necessity of a high voltage to achieve effective LiF splitting but also underscores the crucial role of T-FeF₃ in enabling stable cycling performance, while further highlighting the irreversible nature of R-FeF₃.

Although a high voltage is required to sufficiently form T-FeF₃, concerns regarding electrolyte stability under such high-voltage operation still remain. In particular, LiPF₆ is known to decompose at high potentials, generating fluorine-containing species (F⁻). Moreover, such F⁻ ions may participate in electrode reactions. A previous study reported that F⁻ ions produced by electrolyte decomposition under high-voltage conditions can adsorb onto Mn-based cathodes and promote the formation of mixed phases (*Chem. Mater.* **30**, 5362–5372 (2018)). Based on this, it is important to examine whether, in our study, the formation of T-FeF₃ involves fluoride species originating from side reactions of the electrolyte, as the transformation from FeF₂ to T-FeF₃ fundamentally requires an external fluorine.

Supplementary Fig. S6. Electrochemical profile of LiF-FeF₂ nanocomposite without additional fluorine sources other than LiF.

To examine whether the fluorine involved in the conversion originates from LiF in the composite or from electrolyte degradation, we performed a control study using a fluorine-free electrolyte (1 M LiClO₄ in EC/DMC) and polyacrylonitrile (PAN) binder, eliminating potential external fluorine sources. Despite the environment without F ion source except LiF, the same 4.0 V plateau gradually developed during cycling (Supplementary Fig. S6), strongly indicating that LiF within the composite is the primary fluorine source responsible for T-FeF₃ formation. While we cannot entirely exclude minor effects from electrolyte decomposition, the consistency across systems confirms that the observed transformation is predominantly driven by internal LiF chemistry.

Taken together, our comprehensive electrochemical and control experiments clearly demonstrate that the formation of T-FeF₃ from the LiF-FeF₂ composite necessitates a high charge cut-off voltage of 4.8 V. This voltage is essential to effectively induce the phase transformation to T-FeF₃, which in turn plays a critical role in enabling reversible and stable

cycling behavior. Furthermore, fluorine-free system tests confirm that the fluorine required for T-FeF₃ formation originates from the LiF in the composite, not from electrolyte degradation. Therefore, despite the general concerns associated with high-voltage operation, in our specific system, the use of 4.8 V is both justified and indispensable for realizing the intended phase transition and achieving improved electrochemical performance.